# What Matters When Repurposing Diffusion Models for General Dense Perception Tasks?

**Guangkai Xu**[1]    **Yongtao Ge**[1]    **Mingyu Liu**[1]    **Chengxiang Fan**[1,2]
**Kangyang Xie**[1]    **Zhiyue Zhao**[1]    **Hao Chen**[1]    **Chunhua Shen**[1,2]

[1] Zhejiang University, China    [2] Ant Group

## Abstract

Extensive pre-training with large data is indispensable for downstream geometry and semantic visual perception tasks. Thanks to large-scale text-to-image (T2I) pretraining, recent works show promising results by simply fine-tuning T2I diffusion models for a few dense perception tasks. However, several crucial design decisions in this process still lack comprehensive justification, encompassing the necessity of the multi-step diffusion mechanism, training strategy, inference ensemble strategy, and fine-tuning data quality. In this work, we conduct a thorough investigation into critical factors that affect transfer efficiency and performance when using diffusion priors. Our key findings are: 1) High-quality fine-tuning data is paramount for both semantic and geometry perception tasks. 2) As a special case of the diffusion scheduler by setting its hyper-parameters, the multi-step generation can be simplified to a one-step fine-tuning paradigm without any loss of performance, while significantly speeding up inference. 3) Apart from fine-tuning the diffusion model with only latent space supervision, task-specific supervision can be beneficial to enhance fine-grained details. These observations culminate in the development of **GenPercept**, an effective deterministic one-step fine-tuning paradigm tailored for dense visual perception tasks exploiting diffusion priors. Different from the previous multi-step methods, our paradigm offers a much faster inference speed, and can be seamlessly integrated with customized perception decoders and loss functions for task-specific supervision, which can be critical for improving the fine-grained details of predictions. Comprehensive experiments on a diverse set of dense visual perceptual tasks, including monocular depth estimation, surface normal estimation, image segmentation, and matting, are performed to demonstrate the remarkable adaptability and effectiveness of our proposed method. Code: `https://github.com/aim-uofa/GenPercept`

## 1 Introduction

Recent studies have explored the transferability of text-to-image (T2I) diffusion models to dense visual perception tasks, such as geometry estimation (Ke et al., 2024; Lee et al., 2024; Fu et al., 2024b; Gui et al., 2024; Ye et al., 2024), image segmentation (Van Gansbeke & De Brabandere, 2024; Lee et al., 2024), and inverse rendering (Chen et al., 2024; Kocsis et al., 2024; Zeng et al., 2024). While these works have demonstrated impressive results by repurposing diffusion models for estimating geometric and semantic dense prediction maps, the critical design choices made in transferring diffusion models to other dense perception tasks still lack comprehensive justification. This makes it challenging to determine the optimal strategy for achieving optimal performance.

For example, Ke et al. (2024) align the visual perception process with the denoising process of Stable Diffusion by fine-tuning all U-Net parameters. They highlight the significance of "multi-resolution noise" in the forward diffusion process during training, aiming to obtain clean predictions by gradually removing Gaussian noise. On the other hand, Lee et al. (2024) modify the forward diffusion process by interpolating perception annotations with RGB images instead of using Gaussian noise, and only train the low-rank adaptation (LoRA) (Hu et al., 2022) parameters while keeping the U-Net frozen. To our knowledge, the effective components of these approaches have not been thoroughly investigated, and it is unclear which design choices contribute most to the success.

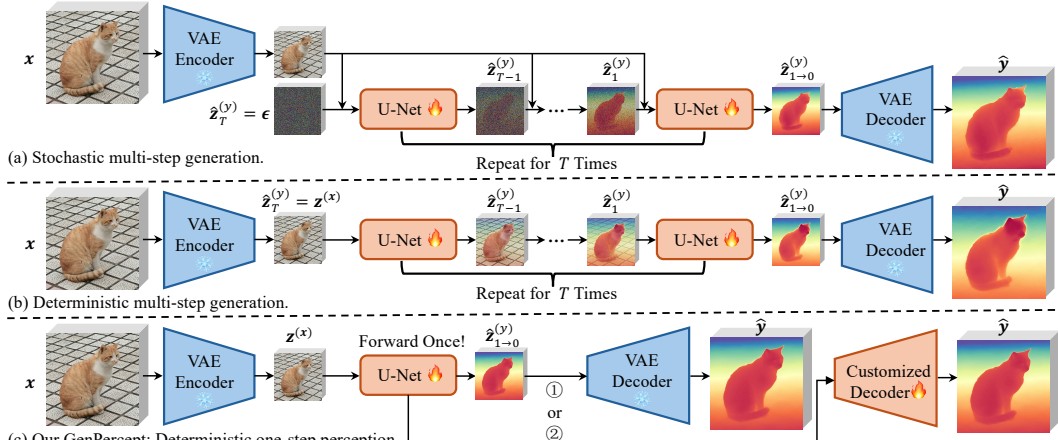

Figure 1: Comparisons of three different pipelines. Our GenPercept enables one-step inference and supports pixel-wise losses and customized decoders to replace the cumbersome VAE decoder. We also extend GenPercept to five dense perception tasks including monocular depth estimation, surface normal estimation, dichotomous image segmentation, semantic segmentation, and image matting.

In this work, we examine the design space of repurposing diffusion models for dense visual perception tasks, and attempt to answer the key question: **What are the important design choices when adapting diffusion models for general dense perception tasks?**

To answer this question, we rethink the importance of both fine-tuning protocols and fine-tuning data. From the perspective of fine-tuning protocols, we categorize recent methods into two main groups: *stochastic multi-step generation* and *deterministic multi-step generation*. We explore several critical design choices, including the diffusion mechanism, key architectural components, training methodologies, and data quality. Our key observations are as follows: **1)** By setting the hyperparameters of the diffusion scheduler to particular values, the multi-step generation can be simplified to a one-step fine-tuning paradigm without any loss of performance. **2)** Strict adherence to traditional diffusion processes appears to be unnecessary. Single-step inference provides similar performance with significantly faster execution. **3)** High-quality synthetic fine-tuning data is crucial for several perception tasks. From the perspective of fine-tuning data quality, we conduct comprehensive dataset ablation studies on both synthetic datasets and real-world datasets.

Based on the aforementioned observations, we propose **GenPercept** (see Fig. 1), a deterministic fine-tuning paradigm featuring a remarkably simple one-step inference pipeline, an optional customized decoder, and an easily adaptable pixel-wise loss. We conduct extensive quantitative and qualitative experiments on a wide range of visual dense perception tasks, including monocular depth estimation, surface normal estimation, image segmentation, and matting to demonstrate the effectiveness and generalization capability of our method.

In conclusion, our contributions can be summarized as follows: **1)** We systematically analyze the design space of fine-tuning protocols, considering both model architecture and dataset selection, through comprehensive ablation studies. **2)** Based on these insights, we propose GenPercept, a simple paradigm that harnesses the power of the pre-trained UNet from diffusion models for generalizable dense visual perception tasks.

## 2  PRELIMINARY

We take the latent diffusion model as an example. To model the data distribution, the idea of the diffusion model (Rombach et al., 2022; Chen et al., 2023; Song et al., 2020; Ho et al., 2020) is to randomly sample a noise $\mathbf{z}_T^{(\mathbf{y})} \sim \mathcal{N}(\mathbf{0}, \mathbf{I})$ and sequentially denoise it into a $\mathbf{z}_0^{(\mathbf{y})}$, which is distributed according to the data. In the forward diffusion process, $\mathbf{z}_t^{(\mathbf{y})}$ is sampled by $\mathbf{z}_t^{(\mathbf{y})} = \sqrt{\bar{\alpha}_t}\mathbf{z}^{(\mathbf{y})} + \sqrt{1 - \bar{\alpha}_t}\boldsymbol{\epsilon}$, where $\boldsymbol{\epsilon} \sim \mathcal{N}(\mathbf{0}, \mathbf{I})$, and $\bar{\alpha}_t = \prod_{s=1}^{t}(1 - \beta_s)$. The variance schedule $\{\beta_t \in (0, 1)\}_{t=1}^{T}$ is interpolated between $\beta_{start}$ and $\beta_{end}$ with $T$ steps, where larger values of $(\beta_{start}, \beta_{end})$ correspond to smaller $\bar{\alpha}_t$ values, *i.e.*, smaller proportion of noise. In the reverse process, Salimans & Ho (2021)

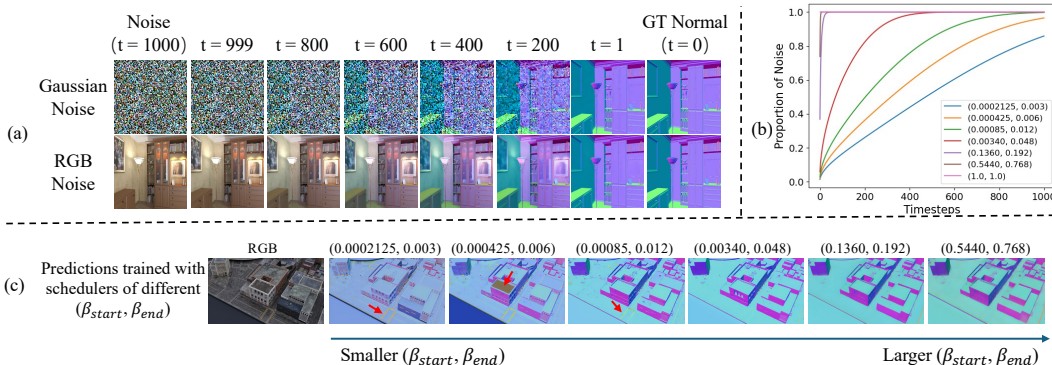

Figure 2: Illustration of different noise forms and proportions in the forward diffusion process. (a) Visualization of interpolating ground-truth labels with Gaussian noise and RGB noise. (b) The relationship between the noise proportion $\sqrt{\bar{\alpha}_t}$ and the $(\beta_{start}, \beta_{end})$ hyperparameters. (c) Small $(\beta_{start}, \beta_{end})$ values during the training of deterministic multi-step generation tend to lead to unclean estimation, which contains some RGB information. Enlarging them may alleviate this issue.

use a "v-prediction" objective, where a denoiser $\boldsymbol{v}_\theta$ minimizes the following:

$$\mathcal{L} = \mathbb{E}_{\mathbf{z}^{(\mathbf{y})}, \boldsymbol{\epsilon} \sim \mathcal{N}(0,I), t \sim \mathcal{U}(T)} \left\| (\sqrt{\bar{\alpha}_t} \boldsymbol{\epsilon} - \sqrt{1 - \bar{\alpha}_t} \mathbf{z}^{(\mathbf{y})}) - \boldsymbol{v}_\theta(\mathbf{z}_t, t) \right\|_2^2. \tag{1}$$

To fully leverage the pre-trained prior of the diffusion models for dense prediction tasks, previous works have reformulated these tasks as a multi-step denoising process, especially on monocular depth estimation. Given a data pair $(\mathbf{z}^{(\mathbf{x})}, \mathbf{z}^{(\mathbf{y})})$ where $\mathbf{z}^{(\mathbf{x})}$ is the observation and $\mathbf{z}^{(\mathbf{y})}$ is the prediction target, *stochastic multi-step generation* methods (Ke et al., 2024; Fu et al., 2024b; Gui et al., 2024) such as Marigold (Ke et al., 2024) add $\mathbf{z}^{(\mathbf{x})}$ as an additional input to the denoiser $\boldsymbol{v}_\theta$, and use $\boldsymbol{v}_\theta(\mathbf{z}_t^{(\mathbf{y})}, \mathbf{z}^{(\mathbf{x})}, t)$ to predict $\mathbf{z}^{(\mathbf{y})}$. By contrast, *deterministic multi-step generation* methods such as DMP (Lee et al., 2024) take the observation $\mathbf{z}^{(\mathbf{x})}$ as a deterministic noise and compose $\mathbf{z}_t^{(\mathbf{y})}$ as a blend between $\mathbf{z}^{(\mathbf{x})}$ and $\mathbf{z}^{(\mathbf{y})}$:

$$\mathbf{z}_t = \mathbf{z}_t^{(\mathbf{y})} = \sqrt{\bar{\alpha}_t} \mathbf{z}^{(\mathbf{y})} + \sqrt{1 - \bar{\alpha}_t} \mathbf{z}^{(\mathbf{x})}, \quad t = [1, ..., T], \tag{2}$$

The denoising process and forward diffusion process of these two categories are illustrated in Fig. 1 and Fig. 2 (a). We offer detailed formulations in the supplementary material.

# 3 DIFFUSION MODELS FOR VISUAL PERCEPTION TASKS

In this section, we explore the necessity and highlight the findings of the multi-step diffusion mechanism, the architectural components, training strategy, and fine-tuning data quality. We select the stochastic method Marigold (Ke et al., 2024) and the deterministic method DMP (Lee et al., 2024) as our baseline methods. The default experimental setting here is similar to Ke et al. (2024) and can be found in the supplementary material.

## 3.1 THE FORM AND PROPORTION OF NOISE IN THE FORWARD DIFFUSION PROCESS

For the training process of Marigold and DMP, the timestep $t$ is sampled to control the proportion of noise added to the ground truth, and the network is trained to recover a clean ground truth from a noisy latent. For smaller timesteps such as "$t = 200$", as illustrated in Fig. 2(a), the input to the network retains significant ground truth information (*e.g.*, the purple color of the normal). Therefore, we hypothesize that *a certain level of ground-truth label information being part of the input* makes the network comparatively be easier to recover the clean ground truth latent than the case of the absence of any ground truth information during training. This can limit the network's capacity to learn comprehensive knowledge and lead to unsatisfactory performance, as it is known that networks can become lazy that tend to exploit "shortcuts"—the ground truth labels in the input in our case. On the other hand, for inference there is no such ground truth available, causing disparity of input

Table 1: Comprehensive quantitative comparisons about the impact of noise forms and proportions in the forward diffusion process on monocular depth estimation. Visualizations of different noise forms and the effect of $(\beta_{start}, \beta_{end})$ values are shown in Fig. 2. The performance of DMP improves steadily, while Marigold shows initial improvements followed by a decline. When $\beta_{start}$ and $\beta_{end}$ are equal to 1, the inference process can be reduced to one step without compromising performance. "Rank" means the average rank of ten evaluation performance (smaller is better).

| Type | Noise Form | Multi-res Noise | Steps | $(\beta_{start}, \beta_{end})$ | KITTI AbsRel↓ | KITTI δ₁↑ | NYU AbsRel↓ | NYU δ₁↑ | ScanNet AbsRel↓ | ScanNet δ₁↑ | DIODE AbsRel↓ | DIODE δ₁↑ | ETH3D AbsRel↓ | ETH3D δ₁↑ | Rank↓ |
|---|---|---|---|---|---|---|---|---|---|---|---|---|---|---|---|
| Marigold | Gaussian | ✓ | 10 | (0.0002125, 0.003) | 0.358 | 0.462 | 0.297 | 0.555 | 0.246 | 0.625 | 0.494 | 0.565 | 0.267 | 0.640 | 7.0 |
| Marigold | Gaussian | ✓ | 10 | (0.000425, 0.006) | 0.122 | 0.854 | 0.106 | 0.887 | 0.136 | 0.829 | 0.345 | 0.716 | 0.086 | 0.927 | 5.8 |
| baseline Marigold | Gaussian | ✓ | 10 | (0.00085, 0.012) | 0.099 | 0.909 | 0.063 | 0.956 | 0.075 | 0.937 | 0.316 | 0.764 | 0.075 | 0.947 | 3.6 |
| Marigold | Gaussian | ✓ | 10 | (0.0034, 0.048) | 0.100 | 0.906 | 0.057 | 0.963 | 0.063 | 0.957 | 0.308 | 0.768 | 0.074 | 0.948 | 2.3 |
| Marigold | Gaussian | ✓ | 10 | (0.1360, 0.192) | 0.119 | 0.861 | 0.058 | 0.963 | 0.061 | 0.961 | 0.315 | 0.760 | 0.073 | 0.950 | 2.8 |
| Marigold | Gaussian | ✓ | 10 | (0.5440, 0.768) | 0.124 | 0.852 | 0.060 | 0.961 | 0.064 | 0.958 | 0.322 | 0.749 | 0.079 | 0.943 | 4.7 |
| Marigold | Gaussian | ✓ | 10 | (1.0, 1.0) | 0.104 | 0.897 | 0.055 | 0.965 | 0.059 | 0.962 | 0.312 | 0.762 | 0.069 | 0.955 | 1.7 |
| Marigold | Gaussian | ✓ | 1 | (1.0, 1.0) | 0.104 | 0.897 | 0.055 | 0.965 | 0.059 | 0.962 | 0.312 | 0.762 | 0.069 | 0.955 | - |
| Marigold | Gaussian | ✗ | 10 | (0.0002125, 0.003) | 0.587 | 0.255 | 0.337 | 0.490 | 0.257 | 0.604 | 0.600 | 0.469 | 0.372 | 0.503 | 7.0 |
| Marigold | Gaussian | ✗ | 10 | (0.000425, 0.006) | 0.536 | 0.289 | 0.313 | 0.527 | 0.248 | 0.621 | 0.565 | 0.499 | 0.328 | 0.575 | 6.0 |
| baseline Marigold | Gaussian | ✗ | 10 | (0.00085, 0.012) | 0.153 | 0.807 | 0.162 | 0.802 | 0.187 | 0.737 | 0.411 | 0.641 | 0.157 | 0.826 | 5.0 |
| Marigold | Gaussian | ✗ | 10 | (0.0034, 0.048) | 0.101 | 0.907 | 0.058 | 0.963 | 0.066 | 0.954 | 0.309 | 0.765 | 0.074 | 0.950 | 2.4 |
| Marigold | Gaussian | ✗ | 10 | (0.1360, 0.192) | 0.115 | 0.870 | 0.056 | 0.965 | 0.060 | 0.961 | 0.313 | 0.763 | 0.072 | 0.953 | 2.3 |
| Marigold | Gaussian | ✗ | 10 | (0.5440, 0.768) | 0.124 | 0.848 | 0.059 | 0.963 | 0.063 | 0.958 | 0.318 | 0.752 | 0.077 | 0.946 | 3.7 |
| Marigold | Gaussian | ✗ | 10 | (1.0, 1.0) | 0.102 | 0.901 | 0.054 | 0.966 | 0.059 | 0.962 | 0.312 | 0.762 | 0.071 | 0.955 | 1.5 |
| Marigold | Gaussian | ✗ | 1 | (1.0, 1.0) | 0.102 | 0.901 | 0.054 | 0.966 | 0.059 | 0.962 | 0.312 | 0.762 | 0.071 | 0.955 | - |
| DMP | RGB | ✗ | 10 | (0.0002125, 0.003) | 0.476 | 0.336 | 0.267 | 0.601 | 0.216 | 0.677 | 0.457 | 0.588 | 0.185 | 0.757 | 6.9 |
| DMP | RGB | ✗ | 10 | (0.000425, 0.006) | 0.265 | 0.630 | 0.201 | 0.072 | 0.195 | 0.717 | 0.386 | 0.674 | 0.116 | 0.880 | 6.1 |
| baseline DMP | RGB | ✗ | 10 | (0.00085, 0.012) | 0.134 | 0.837 | 0.117 | 0.871 | 0.147 | 0.808 | 0.353 | 0.721 | 0.093 | 0.919 | 5.0 |
| DMP | RGB | ✗ | 10 | (0.0034, 0.048) | 0.107 | 0.890 | 0.077 | 0.939 | 0.087 | 0.923 | 0.318 | 0.766 | 0.078 | 0.940 | 3.8 |
| DMP | RGB | ✗ | 10 | (0.1360, 0.192) | 0.107 | 0.890 | 0.063 | 0.959 | 0.068 | 0.955 | 0.305 | 0.773 | 0.073 | 0.948 | 2.2 |
| DMP | RGB | ✗ | 10 | (0.5440, 0.768) | 0.106 | 0.897 | 0.061 | 0.959 | 0.066 | 0.952 | 0.309 | 0.768 | 0.075 | 0.945 | 2.3 |
| DMP | RGB | ✗ | 10 | (1.0, 1.0) | 0.100 | 0.902 | 0.053 | 0.966 | 0.059 | 0.961 | 0.309 | 0.768 | 0.068 | 0.956 | 1.2 |
| Our baseline | RGB | ✗ | 1 | (1.0, 1.0) | 0.100 | 0.902 | 0.053 | 0.966 | 0.059 | 0.961 | 0.309 | 0.768 | 0.068 | 0.956 | - |

signals between inference and training. We note that T2I tasks may suffer less due to their stochastic nature, namely, converting a text prompt to a generated image is one-to-many mapping process.

To alleviate this issue, we attempt to control the blending proportion by changing the $(\beta_{start}, \beta_{end})$ values of the diffusion model's DDPM scheduler. As shown in Fig. 2(b) and Fig. 2 of the supplementary, training with a $(\beta_{start}, \beta_{end})$ value of (1.0, 1.0) achieves the best rank performance for both Gaussian noise and RGB noise, which is demonstrated in Table 1. Rather than achieving consistent performance improvement while increasing the noise proportion, we observed that Marigold's performance begins to be slightly unstable when the $(\beta_{start}, \beta_{end})$ values are sufficiently high. Experiments of varying the random seed during both the training and inference process are conducted to rule out the influence of randomness. The results show that we may not be able to exactly find a set of unique values for $(\beta_{start}, \beta_{end})$ to achieve the best accuracy, as the final accuracy can be affected by a few other factors besides the aforementioned one.

Additionally, when $(\beta_{start}, \beta_{end})$ are equal to 1, the noise proportion $\sqrt{\bar{\alpha}_t}$ is equal to 0, and the formulation of DMP can be derived from Eq. (1) and Eq. (2) as follows.

$$\mathbf{z}_t = \mathbf{z}_t^{(\mathbf{y})} = \sqrt{\bar{\alpha}_t}\mathbf{z}^{(\mathbf{y})} + \sqrt{1 - \bar{\alpha}_t}\mathbf{z}^{(\mathbf{x})} = \mathbf{z}^{(\mathbf{x})}, \ \ t = [1, ..., T],$$
$$\mathcal{L} = \mathbb{E}_{\mathbf{z}^{(\mathbf{y})}, \boldsymbol{\epsilon} \sim \mathcal{N}(0,I), t \sim \mathcal{U}(T)} \left\| -\mathbf{z}^{(\mathbf{y})} - \boldsymbol{v}_\theta(\mathbf{z}_t, t) \right\|_2^2. \tag{3}$$

In this case, the output of the denoiser $\boldsymbol{v}_\theta(\cdot, \cdot)$ is enforced to learn the negative value of the ground truth latent for each step, and the multi-step denoising is equivalent to the single-step denoising. We propose to reduce the DDIM steps of DMP to one and call it "deterministic one-step perception". The resulting inference can be significantly faster, with performance remaining almost unchanged. We name this as "our baseline" for the subsequent analysis.

> **Finding 1.** By setting the $(\beta_{start}, \beta_{end})$ values to 1, the multi-step generation is simplified to a one-step fine-tuning paradigm without any loss of performance in both stochastic and deterministic methods, *e.g.*, Marigold (Ke et al., 2024) and DMP (Lee et al., 2024) respectively.

## 3.2 WHERE DOES THE RICH VISUAL KNOWLEDGE RESIDE IN DIFFUSION MODELS?

Based on the baseline proposed in §3.1, we conduct detailed ablation studies to thoroughly investigate the necessity of each component of Stable Diffusion. Results are reported in Table 2.

**Denoiser.** We reinitialize the U-Net parameters and train the network from scratch on the same datasets. Without prior knowledge of large data from LAION-5B, the network performs poorly and

Table 2: Explorations on the impact of the Stable Diffusion components on depth estimation. Customized decoders and losses can also enable inference acceleration and performance improvement.

| Setting | Loss | KITTI | | NYU | | ScanNet | | DIODE | | ETH3D | |
|---|---|---|---|---|---|---|---|---|---|---|---|
| | | AbsRel↓ | $\delta_1$↑ | AbsRel↓ | $\delta_1$↑ | AbsRel↓ | $\delta_1$↑ | AbsRel↓ | $\delta_1$↑ | AbsRel↓ | $\delta_1$↑ |
| Our baseline | MSE (Latent) | 0.100 | 0.902 | 0.053 | 0.966 | 0.059 | 0.961 | 0.309 | 0.768 | 0.068 | 0.956 |
| Train U-Net from scratch | MSE (Latent) | 0.219 | 0.650 | 0.186 | 0.736 | 0.183 | 0.729 | 0.426 | 0.614 | 0.185 | 0.741 |
| Train VAE decoder from scratch | MSE (Image) | 0.096 | 0.916 | 0.055 | 0.964 | 0.058 | 0.964 | 0.302 | 0.759 | 0.071 | 0.950 |
| Baseline + Image MSE loss | MSE (Image) | 0.097 | 0.915 | 0.054 | 0.964 | 0.059 | 0.964 | 0.305 | 0.760 | 0.071 | 0.953 |
| Baseline + Image customized loss | MSE + SSI + Grad. (Image) | 0.094 | 0.923 | 0.052 | 0.966 | 0.056 | 0.965 | 0.302 | 0.767 | 0.066 | 0.967 |
| Train DPT decoder from scratch | MSE (Image) | 0.099 | 0.912 | 0.055 | 0.964 | 0.058 | 0.963 | 0.302 | 0.759 | 0.069 | 0.956 |

loses the generalization capability. This indicates that most of the prior knowledge is stored in the U-Net module.

**VAE AutoEncoder.** The VAE encoder's original architecture is kept intact to maintain the consistency of the encoding process. For the VAE decoder, we train it from scratch with image pixel MSE loss. Without pre-trained parameters of the VAE decoder, it still performs well.

**Customized Head and Loss.** The deterministic one-step perception pipeline enables customized heads and loss functions. By utilizing a DPT decoder (Ranftl et al., 2021) and the loss functions of DepthAnythingv2 (Yang et al., 2024b), we can implement a lightweight decoder that supervises pixel-wise information at a higher resolution rather than latent features at $1/8$ resolution. This approach can accelerate inference times and enhance the acquisition of fine-grained details.

> ***Finding 2.*** The primary perceptual prior knowledge of diffusion models is encapsulated within the U-Net of the diffusion model. Customized heads and loss functions offers flexibility and may lead to faster inference speed and improved results.

## 3.3 WHAT ABOUT THE TIMESTEPS AND TEXT PROMPTS?

The timesteps and text prompts are crucial elements in utilizing the Stable Diffusion model to generate diverse images. We conducted ablation studies to investigate their significance. The results reported in Table 3 indicate a negligible difference between various settings. Owing to the inherent certainty associated with visual perception tasks, the diversity typically offered by the textual inputs appears to be unnecessary. Similarly, the utility of timesteps is reduced, as the single-step paradigm does not require progressive denoising.

Table 3: Quantitative comparisons among different timesteps and text prompts on depth estimation.

| Setting | Text Prompt | Train / Infer Timesteps | KITTI | | NYU | | ScanNet | | DIODE | | ETH3D | |
|---|---|---|---|---|---|---|---|---|---|---|---|---|
| | | | AbsRel↓ | $\delta_1$↑ | AbsRel↓ | $\delta_1$↑ | AbsRel↓ | $\delta_1$↑ | AbsRel↓ | $\delta_1$↑ | AbsRel↓ | $\delta_1$↑ |
| Our baseline | "" | Random / 1 | 0.100 | 0.902 | 0.053 | 0.966 | 0.059 | 0.961 | 0.309 | 0.768 | 0.068 | 0.956 |
| Valid text input | "A high quality RGB image" | Random / 1 | 0.101 | 0.900 | 0.053 | 0.967 | 0.058 | 0.964 | 0.312 | 0.762 | 0.070 | 0.954 |
| Random text input | "F3@qV!k2*#Zpˆn%1Lz" | Random / 1 | 0.099 | 0.904 | 0.054 | 0.965 | 0.059 | 0.963 | 0.311 | 0.763 | 0.069 | 0.955 |
| Timestep1 | "" | 1 / 1 | 0.100 | 0.906 | 0.054 | 0.965 | 0.060 | 0.961 | 0.304 | 0.769 | 0.069 | 0.956 |
| Timestep500 | "" | 500 / 500 | 0.102 | 0.897 | 0.053 | 0.966 | 0.059 | 0.961 | 0.307 | 0.765 | 0.068 | 0.956 |
| Timestep900 | "" | 900 / 900 | 0.105 | 0.891 | 0.054 | 0.966 | 0.058 | 0.964 | 0.309 | 0.762 | 0.068 | 0.953 |

> ***Finding 3.*** The timesteps and text prompts of diffusion models are negligible for the performance of visual perception tasks.

## 3.4 HOW TO LEVERAGE THE U-NET'S PRIOR KNOWLEDGE?

The significance of the denoiser cannot be overstated. However, the strategies for its utilization are worth a careful study. Should we freeze the denoiser, utilize its intermediate features, and merely fine-tune the decoder for specific tasks? Alternatively, can we employ LoRA (Hu et al., 2022) instead of extensively fine-tuning the entire denoiser? Unfortunately, the evidence suggests that neither approach is ideal. As illustrated in Table 4, freezing the denoiser significantly compromises performance. Although incorporating LoRA offers some advantages, it may not fully leverage the potential of denoiser, especially with regular LoRA ranks of 4 and 16. This limitation likely stems from the substantial differences between the noise-to-image denoising process and the image-to-perception prediction task.

Table 4: Explorations on the paradigms to leverage U-Net's prior knowledge on depth estimation.

| Setting | LoRA Rank | KITTI AbsRel↓ | KITTI $\delta_1$↑ | NYU AbsRel↓ | NYU $\delta_1$↑ | ScanNet AbsRel↓ | ScanNet $\delta_1$↑ | DIODE AbsRel↓ | DIODE $\delta_1$↑ | ETH3D AbsRel↓ | ETH3D $\delta_1$↑ |
|---|---|---|---|---|---|---|---|---|---|---|---|
| Our baseline | - | 0.100 | 0.902 | 0.053 | 0.966 | 0.059 | 0.961 | 0.309 | 0.768 | 0.068 | 0.956 |
| Freeze U-Net + Train DPT decoder | - | 0.144 | 0.803 | 0.086 | 0.931 | 0.097 | 0.911 | 0.309 | 0.768 | 0.068 | 0.956 |
| Train U-Net with LoRA | 4 | 0.211 | 0.644 | 0.095 | 0.914 | 0.100 | 0.902 | 0.372 | 0.689 | 0.121 | 0.864 |
| Train U-Net with LoRA | 16 | 0.166 | 0.746 | 0.085 | 0.931 | 0.087 | 0.927 | 0.352 | 0.712 | 0.104 | 0.901 |
| Train U-Net with LoRA | 64 | 0.138 | 0.817 | 0.077 | 0.944 | 0.079 | 0.940 | 0.336 | 0.734 | 0.089 | 0.930 |
| Train U-Net with LoRA | 256 | 0.133 | 0.827 | 0.069 | 0.952 | 0.073 | 0.947 | 0.325 | 0.745 | 0.088 | 0.933 |
| Train U-Net with LoRA | 1024 | 0.125 | 0.849 | 0.067 | 0.955 | 0.074 | 0.947 | 0.324 | 0.747 | 0.084 | 0.939 |

Table 5: Investigations into the impact of training data quality on depth estimation.

| Data Quality | Datasets | KITTI AbsRel↓ | KITTI $\delta_1$↑ | NYU AbsRel↓ | NYU $\delta_1$↑ | ScanNet AbsRel↓ | ScanNet $\delta_1$↑ | DIODE AbsRel↓ | DIODE $\delta_1$↑ | ETH3D AbsRel↓ | ETH3D $\delta_1$↑ |
|---|---|---|---|---|---|---|---|---|---|---|---|
| Synthetic Data | Hypersim (50K) + Virtual KITTI (40K) | 0.100 | 0.902 | 0.053 | 0.966 | 0.059 | 0.961 | 0.309 | 0.768 | 0.068 | 0.956 |
| Real Data | Taskonomy (50K) + Cityscapes (40K) | 0.123 | 0.857 | 0.055 | 0.966 | 0.062 | 0.958 | 0.293 | 0.762 | 0.074 | 0.947 |

> ***Finding 4.*** Fine-tuning the denoiser appears to be preferable for achieving better results, compared to either merely utilizing its intermediate features or training a LoRA.

## 3.5 IS THE TRAINING DATA QUALITY ESSENTIAL?

The quality of annotations in real datasets is often lower compared to synthetic datasets, where data is precisely rendered via simulators. In Table 5, we explore the impact of data quality on the fine-tuning process. We sample the same distribution of real data, consisting of 90% from approximately 50K indoor images from the Taskonomy dataset (Zamir et al., 2018) and 10% from about 40K outdoor images from the Cityscapes dataset (Cordts et al., 2016). With lower annotation quality, the model achieves slightly worse performance. Also, the visualization in the supplementary material indicates that noisy data significantly influences detailed predictions in visual perception tasks.

> ***Finding 5.*** Data quality affects the fine-grained details of dense predictions significantly.

## 3.6 SUMMARY OF THE OBERVATIONS

Based on the preceding analysis, an effective approach to leveraging the prior knowledge of diffusion models is to use them as single-step deterministic perception estimators. This can be done with either a VAE decoder or a customized lightweight decoder. Additionally, employing pixel-specific customized losses can further enhance detail and overall performance. We compare our deterministic single-step perception method with previous multi-step paradigms in Fig. 1. In the following section, we extend these findings to a broader set of visual perception tasks, including surface normal estimation, semantic image segmentation, dichotomous image segmentation, and image matting.

## 4 EXPERIMENTS ON VARIOUS DENSE VISUAL PERCEPTUAL TASKS

In this section, we empirically show the robust transfer ability of our GenPercept on diverse visual tasks. Unless specified otherwise, we freeze the VAE AutoEncoder and fine-tune the U-Net of Stable Diffusion v2.1 to estimate the ground-truth label latent for 30000 iterations, with a resolution of (768, 768), a batch size of 32, and a learning rate of $3e-5$. Different customized loss functions are utilized to improve the performance further on dense visual perception tasks.

## 4.1 GEOMETRIC ESTIMATION

For geometry evaluation, the ensemble size, inference resolution, valid evaluation depth range (specific for depth estimation), and evaluation average paradigm (average by pixels or average by the number of images) can be different for each method. To compare these approaches fairly, we follow the open-source evaluation code of Marigold (Ke et al., 2024) for depth and DSINE (Bae & Davison, 2024) for surface normal, and evaluate the performance of partial existing SOTA methods with their officially released model weights. They are labeled with [†] in the Table.

Table 6: Quantitative comparison of affine-invariant depth estimation on five zero-shot datasets. Results marked with † are evaluated following the evaluation protocol of Marigold by ourselves.

| Method | Training Samples | KITTI AbsRel↓ | KITTI $\delta_1$↑ | NYU AbsRel↓ | NYU $\delta_1$↑ | ScanNet AbsRel↓ | ScanNet $\delta_1$↑ | DIODE AbsRel↓ | DIODE $\delta_1$↑ | ETH3D AbsRel↓ | ETH3D $\delta_1$↑ |
|---|---|---|---|---|---|---|---|---|---|---|---|
| MiDaS (Ranftl et al., 2020) | 2M | 0.236 | 0.630 | 0.111 | 0.885 | 0.121 | 0.846 | 0.332 | 0.715 | 0.184 | 0.752 |
| Omnidata (Eftekhar et al., 2021) | 12.2M | 0.149 | 0.835 | 0.074 | 0.945 | 0.075 | 0.936 | 0.339 | 0.742 | 0.166 | 0.778 |
| DPT-large (Ranftl et al., 2021) | 1.4M | 0.100 | 0.901 | 0.098 | 0.903 | 0.082 | 0.934 | 0.182 | 0.758 | 0.078 | 0.946 |
| DepthAnything† (Yang et al., 2024a) | 63.5M | 0.080 | 0.946 | 0.043 | 0.980 | 0.043 | 0.981 | 0.261 | 0.759 | 0.058 | 0.984 |
| DepthAnything v2† (Yang et al., 2024b) | 62.6M | 0.080 | 0.943 | 0.043 | 0.979 | 0.042 | 0.979 | 0.321 | 0.758 | 0.066 | 0.983 |
| Metric3D v2† (Hu et al., 2024) | 16M | **0.052** | **0.979** | **0.039** | **0.979** | **0.023** | **0.989** | 0.147 | **0.892** | **0.040** | **0.983** |
| DiverseDepth (Yin et al., 2020) | 320K | 0.190 | 0.704 | 0.117 | 0.875 | 0.109 | 0.882 | 0.376 | 0.631 | 0.228 | 0.694 |
| LeReS (Yin et al., 2021) | 354K | 0.149 | 0.784 | 0.090 | 0.916 | 0.091 | 0.917 | 0.271 | 0.766 | 0.171 | 0.777 |
| HDN (Zhang et al., 2022) | 300K | 0.115 | 0.867 | 0.069 | 0.948 | 0.080 | 0.939 | 0.246 | **0.780** | 0.121 | 0.833 |
| GeoWizard (Fu et al., 2024b) | 280K | 0.097 | 0.921 | **0.052** | 0.966 | 0.061 | 0.953 | 0.297 | 0.792 | **0.064** | **0.961** |
| DepthFM (Gui et al., 2024) | 63K | 0.083 | 0.934 | 0.065 | 0.956 | - | - | **0.225** | **0.800** | - | - |
| Marigold† (Ke et al., 2024) | 74K | 0.099 | 0.916 | 0.055 | 0.964 | 0.064 | 0.951 | 0.308 | 0.773 | 0.065 | 0.960 |
| DMP Official† (Lee et al., 2024) | - | 0.240 | 0.622 | 0.109 | 0.891 | 0.146 | 0.814 | 0.361 | 0.706 | 0.128 | 0.857 |
| GeoWizard† (Fu et al., 2024b) | 280K | 0.129 | 0.851 | 0.059 | 0.959 | 0.066 | 0.953 | 0.328 | 0.753 | 0.077 | 0.940 |
| DepthFM† (Gui et al., 2024) | 63K | 0.174 | 0.718 | 0.082 | 0.932 | 0.095 | 0.903 | 0.334 | 0.729 | 0.101 | 0.902 |
| Our GenPercept (Depth) | 90K | 0.094 | 0.923 | **0.052** | 0.966 | **0.056** | **0.965** | 0.302 | 0.767 | 0.066 | 0.957 |
| Our GenPercept (Disparity) | 90K | 0.080 | 0.934 | 0.058 | **0.969** | 0.063 | 0.960 | 0.226 | 0.741 | 0.096 | 0.959 |
| Our GenPercept (Disparity + DPT head) | 90K | **0.078** | **0.935** | 0.059 | 0.967 | 0.064 | 0.961 | 0.228 | 0.740 | 0.094 | **0.961** |

Table 7: Quantitative comparison of surface normal estimation on three zero-shot datasets. We evaluate mean error↓, median error ↓ (med.), and the percentages of pixels ↑ with five thresholds. Part of the reported results (†) are evaluated following the evaluation protocol of DSINE by ourselves.

| Method | Training Samples | NYU v2 mean | med. | 5.0° | 7.5° | 11.25° | 22.5° | 30° | ScanNet mean | med. | 5.0° | 7.5° | 11.25° | 22.5° | 30° | Sintel mean | med. | 5.0° | 7.5° | 11.25° | 22.5° | 30° |
|---|---|---|---|---|---|---|---|---|---|---|---|---|---|---|---|---|---|---|---|---|---|---|
| Omnidata v1 (Eftekhar et al., 2021) | 12.2M | 23.1 | 12.9 | 21.6 | 33.4 | 45.8 | 66.3 | 73.6 | 22.9 | 12.3 | 21.5 | 34.5 | 47.4 | 66.1 | 73.2 | 41.5 | 35.7 | 3.0 | 5.8 | 11.4 | 30.4 | 42.0 |
| Ominidata v2 (Kar et al., 2022) | 12.2M | 17.2 | 9.7 | 25.3 | 40.2 | 55.5 | 76.5 | 83.0 | 16.2 | 8.5 | 29.1 | 44.9 | 60.2 | 79.5 | 84.7 | 40.5 | 35.1 | 4.6 | 7.9 | 14.7 | 33.0 | 43.5 |
| Metric3D v2† (Hu et al., 2024) | 8.8M | 13.5 | 6.7 | 40.1 | 53.5 | 65.9 | 82.6 | 87.1 | 11.8 | 5.5 | 46.6 | 60.7 | 71.6 | 85.4 | 89.7 | 22.8 | 14.2 | 18.4 | 28.5 | 41.6 | 66.7 | 75.8 |
| Geowizard (Fu et al., 2024b) | 280K | 17.0 | - | - | - | - | 56.5 | - | 15.4 | - | - | - | - | 61.6 | - | - | - | - | - | - | - | - |
| DINSE† (Bae & Davison, 2024) | 160K | 16.4 | 8.4 | 32.8 | 46.3 | 59.6 | 77.7 | 83.5 | 16.2 | 8.3 | 29.8 | 45.9 | 61.0 | 78.7 | 84.4 | 34.9 | 28.1 | **8.9** | **14.1** | **21.5** | 41.5 | 52.7 |
| Geowizard† (Fu et al., 2024b) | 280K | 19.8 | 11.2 | 18.0 | 32.7 | 50.2 | 73.0 | 79.9 | 21.1 | 11.9 | 15.9 | 29.7 | 47.4 | 70.7 | 77.8 | 36.1 | 28.4 | 4.1 | 8.6 | 16.9 | 39.8 | 52.5 |
| Our GenPercept (Latent MSE loss) | 90K | 17.4 | 9.5 | 23.3 | 40.0 | 56.3 | 76.8 | 83.0 | 16.3 | 8.9 | 25.8 | 42.7 | 59.6 | 79.4 | 84.8 | 44.4 | 31.6 | 3.4 | 7.5 | 15.0 | 37.0 | 48.0 |
| Our GenPercept (Image angular loss) | 90K | **16.4** | **8.0** | **33.3** | **47.8** | **60.9** | **78.3** | **83.7** | **15.2** | **7.4** | **33.9** | **50.7** | **65.0** | **80.9** | **85.7** | **34.6** | **26.2** | 5.2 | 9.8 | 18.4 | **43.8** | **55.8** |

**Monocular Depth Estimation.** The monocular depth estimation aims to predict the vertical distance between the observed object and the camera from an RGB image. The estimated depth is formulated as affine-invariant depth (Yin et al., 2021; Ranftl et al., 2020; 2021), and should be recovered by performing least square regression with the ground truth. The evaluation is performed on five zero-shoft datasets including KITTI (Geiger et al., 2013), NYU (Silberman et al., 2012), ScanNet (Dai et al., 2017), DIODE (Vasiljevic et al., 2019), and ETH3D (Schops et al., 2017). We compute the absolute relative error (AbsRel↓) and percentage of accurate valid depth pixels ($\delta_1$↑). Invalid regions are filtered out and the metrics are averaged on all images.

**Surface Normal Estimation.** The surface normal estimation aims to predict a vector perpendicular to tangent plane of the surface at each point P, which represents the orientation of the object's surface. For evaluation, we compute the angular error on three zero-shot datasets including NYU (Silberman et al., 2012), ScanNet (Dai et al., 2017), and Sintel (Butler et al., 2012). The mean ↓, median ↓, and the percentages of pixels ↑ with error below thresholds [5°, 7.5°, 11.25°, 22.5°, 30°] are reported. Invalid regions are filtered out and the metrics are averaged on all images.

**Quantitative Evaluation.** Quantitative results on monocular depth estimation and surface normal estimation are shown in Table 6 and Table 7, respectively. Even trained on limited synthetic datasets only, our GenPercept shows much robustness and achieves promising performance on diverse unseen scenes. For monocular depth models, we train them with pixel-wise MSE loss, scale-shift-invariant loss (Ranftl et al., 2020), and gradient loss (Ranftl et al., 2020). Furthermore, our disparity model (inverse of the depth) shows much better performance on datasets with outdoor scenes, such as KITTI and DIODE, but less performance on indoor datasets. Therefore, we suggest adopting the depth model for indoor scenes and the disparity model for outdoor scenes experimentally. By replacing the VAE decoder with a lightweight DPT head (Ranftl et al., 2021), GenPercept can infer faster without bearing the performance penalty. For surface normal estimation, the image angular loss brings significant performance improvement thanks to our one-step estimation paradigm.

**Qualitative Results.** Qualitative visualizations are shown in Fig. 3. We observe excellent generalization of our models in that they can estimate accurate geometric information and promising details not only on diverse real and synthetic scenes, but also on comics, color drafts, and even sketches.

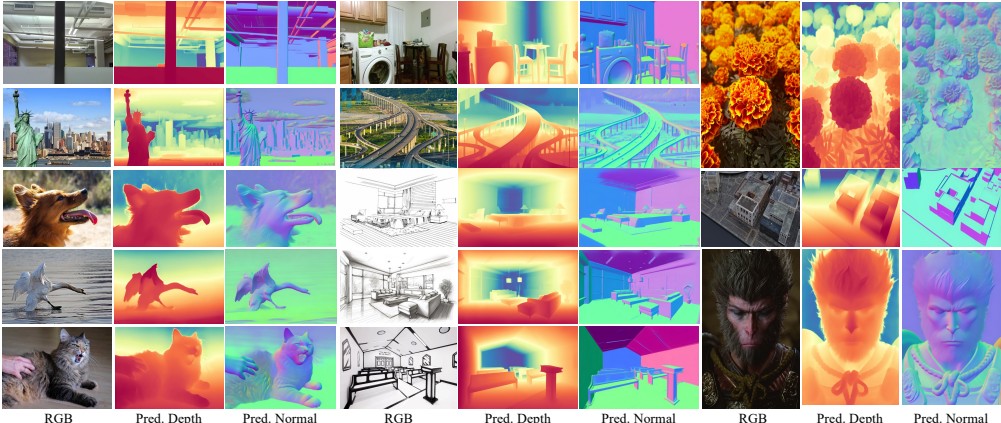

| RGB | Pred. Depth | Pred. Normal | RGB | Pred. Depth | Pred. Normal | RGB | Pred. Depth | Pred. Normal |

Figure 3: Qualitative results of monocular depth and surface normal estimation. The model works surprisingly well on *out-of-domain images (sketch and cartoon images).*

## 4.2 Image Segmentation

**Dichotomous Image Segmentation**. This is a category-agnostic, high-quality object segmentation task that accurately separates the object from the background in an image. Consistent with previous methods, we use the six evaluation metrics specified in the DIS task, which include maximal F-measure ($maxF_\beta \uparrow$) (Achanta et al., 2009), weighted F-measure ($F_\beta^w \uparrow$) (Margolin et al., 2014), mean absolute error ($M \downarrow$) (Perazzi et al., 2012), structural measure ($S_\alpha \uparrow$) (Fan et al., 2017), mean enhanced alignment measure ($E_\phi^m \uparrow$) (Fan et al., 2018; 2021b) and human correction efforts ($HCE_\gamma \downarrow$) (Qin et al., 2022). We choose DIS5K (Qin et al., 2022) as the training and testing dataset. We utilize DIS-TR for training and evaluate our model on DIS-VD and DIS-TE subsets. The pixelwise MSE loss is utilized during training.

Quantitative results of dichotomous image segmentation are shown in Table 8. We only show partial results due to paper page limitations, full comparisons are accessible in the supplementary material. GenPercept outperforms methods like HySM (Nirkin et al., 2021) and IS-Net (Qin et al., 2022) on this challenging dataset across most evaluation metrics, but there exists room for further improvement compared to SoTA methods like MVANet (Yu et al., 2024). As shown in Fig. 5, our approach provides a detailed foreground mask. For thin lines and meticulous objects that are difficult for previous methods to process, our method can also output accurate segmentation results.

**Semantic Image Segmentation.** This is a fundamental computer vision task that involves classifying each pixel in an image into a specific category or class (Zhao et al., 2024; Ji et al., 2024). For training, we utilized the indoor synthetic dataset, HyperSim (Roberts et al., 2021), which comprises 40 semantic segmentation class labels. We encode different classes into 3-channel colormaps, treat the task as a regression problem, and fine-tune the original Stable Diffusion with the pixel-wise MSE loss. As demonstrated in Fig. 4, the model generalizes well to classes within the HyperSim annotations, such as chairs and desks, but struggles with unrecognized categories such as cats and cars.

Another choice involves using a customized segmentation head. We incorporate a custom segmentation head, namely UperNet (Xiao et al., 2018), onto the multi-level features extracted by UNet. For the UperNet segmentation head, we follow the traditional semantic segmentation format to use n-channel output, where n is the number of categories. The quantitative results are presented in Table 9, we test the model's performance on Hypersim (Roberts et al., 2021) and zero-shot ability on a subset of the ADE20k (Zhou et al., 2017) validation set, which contains overlapping classes.

Table 8: Quantitative results of dichotomous image segmentation on DIS5K validation and testing sets. Additional cross-dataset evaluation is provided in the supplementary material.

| Dataset | DIS-VD | | | | | | DIS-TE4 | | | | | | Overall DIS-TE (1-4) | | | | | |
|---|---|---|---|---|---|---|---|---|---|---|---|---|---|---|---|---|---|---|
| Metric | $maxF_\beta \uparrow$ | $F_\beta^w \uparrow$ | $M \downarrow$ | $S_\alpha \uparrow$ | $E_\phi^m \uparrow$ | $HCE_\gamma \downarrow$ | $maxF_\beta \uparrow$ | $F_\beta^w \uparrow$ | $M \downarrow$ | $S_\alpha \uparrow$ | $E_\phi^m \uparrow$ | $HCE_\gamma \downarrow$ | $maxF_\beta \uparrow$ | $F_\beta^w \uparrow$ | $M \downarrow$ | $S_\alpha \uparrow$ | $E_\phi^m \uparrow$ | $HCE_\gamma \downarrow$ |
| U²Net (Qin et al., 2020) | 0.748 | 0.656 | 0.090 | 0.781 | 0.823 | 1413 | 0.795 | 0.705 | 0.087 | 0.807 | 0.847 | 3653 | 0.761 | 0.670 | 0.083 | 0.791 | 0.835 | 1333 |
| SINetV2 (Fan et al., 2021a) | 0.665 | 0.584 | 0.110 | 0.727 | 0.798 | 1568 | 0.699 | 0.616 | 0.113 | 0.744 | 0.824 | 3683 | 0.693 | 0.608 | 0.101 | 0.747 | 0.822 | 1411 |
| HySM (Nirkin et al., 2021) | 0.734 | 0.640 | 0.096 | 0.773 | 0.814 | 1324 | 0.782 | 0.693 | 0.091 | 0.802 | 0.842 | 3331 | 0.757 | 0.665 | 0.084 | 0.792 | 0.834 | 1218 |
| IS-Net (Qin et al., 2022) | 0.791 | 0.717 | 0.074 | 0.813 | 0.856 | 1116 | 0.827 | 0.753 | 0.072 | 0.830 | 0.870 | 2888 | 0.799 | 0.726 | 0.070 | 0.819 | 0.858 | 1016 |
| MVANet (Yu et al., 2024) | 0.904 | 0.861 | 0.035 | 0.909 | 0.937 | 878 | 0.911 | 0.857 | 0.041 | 0.903 | 0.944 | 2301 | 0.916 | 0.855 | 0.035 | 0.905 | 0.938 | 790 |
| Our Genpercept | 0.857 | 0.835 | 0.04 | 0.87 | 0.934 | 1511 | 0.848 | 0.829 | 0.049 | 0.854 | 0.938 | 3799 | 0.863 | 0.839 | 0.039 | 0.872 | 0.936 | 1364 |
| Our Genpercept (infer. at 1024px) | 0.877 | 0.859 | 0.035 | 0.887 | 0.941 | 1262 | 0.874 | 0.858 | 0.041 | 0.874 | 0.947 | 3321 | 0.875 | 0.856 | 0.036 | 0.885 | 0.939 | 1176 |

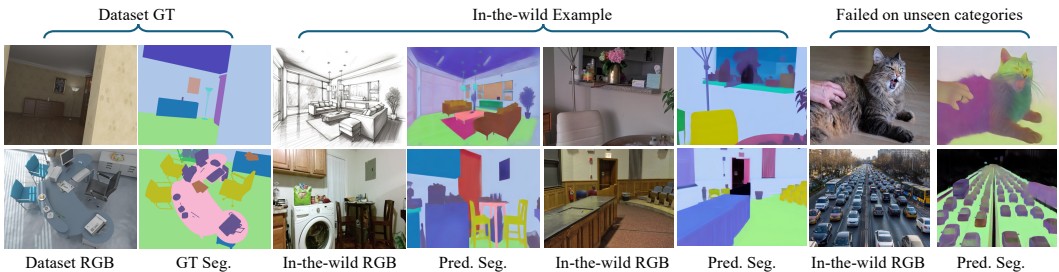

Figure 4: Quanlitative results of semantic segmentation in the wild. Trained on the synthetic indoor Hypersim dataset, GenPercept shows much robustness on the trained categories of complex in-the-wild images, *e.g.*, yellow chairs, green floor, and light blue wall. Due to the limited annotation categories and little negative label of "unknown category", it sometimes fails in outdoor scenes and unseen categories such as cats and cars.

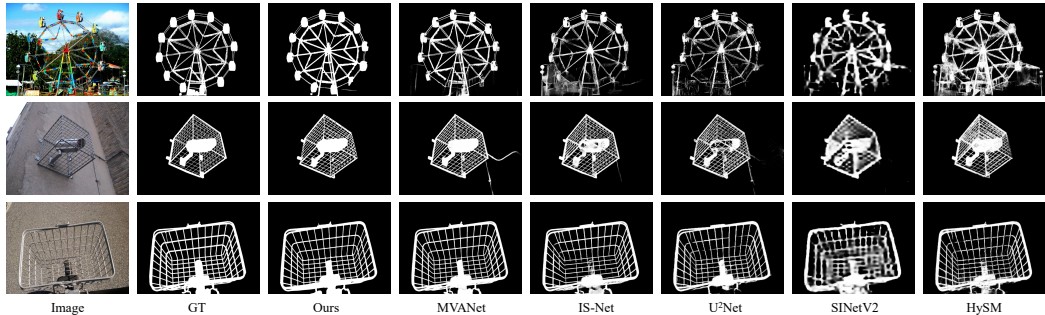

Figure 5: Qualitative comparison of dichotomous image segmentation.

Besides, we compare with Mask2Former (Cheng et al., 2022) by training on ADE20K. GenPercept outperforms ResNet50 (He et al., 2016) and Swin-T (Liu et al., 2021) of Mask2Former but achieves lower performance than Swin-L (Liu et al., 2021).

## 4.3 IMAGE MATTING

**Task Definition.** Image matting aims to extract the foreground, background, and alpha mask from an image. Traditional approaches depend on supplementary inputs that delineate foreground, background, and ambiguous areas to reduce uncertainty. Automatic image matting seeks to remove this dependency by directly estimating these components from the image alone. The implementation details can be found in the supplementary material.

**Quantitative and Qualitative Results.** We evaluate metrics including the sum of absolute differences (SAD), mean squared error (MSE), mean absolute difference (MAD), gradient (Grad.), and Connectivity (Conn.) on the P3M-500-NP test set. SAD and MAD measure the mean L1 distance between predictions and ground truth labels. MSE and CONN focus on L2 distance and connectivity that better reflects human intuition. As shown in Table 10, our GenPercept is less accurate compared with the state-of-the-art methods. However, when transferring the human image matting ability to general image matting tasks, GenPercept achieves much better performance. It proves the robustness brought by the prior knowledge of diffusion models pre-trained on the LAION dataset. Quantitative results of image matting are shown in the supplementary.

Table 9: Quantitative results of semantic segmentation on Hypersim and ADE20k.

| Method | Training Dataset | mIoU↑ (Hypersim) | mIoU↑ (ADE20K) |
|---|---|---|---|
| GenPercept (Train UperNet) | Hypersim | 46.0 | 34.1 |
| GenPercept (Train U-Net + UperNet) | Hypersim | 52.9 | 38.3 |
| Mask2Former R50 | ADE20K | - | 47.2 |
| Mask2Former Swin-T | ADE20K | - | 47.7 |
| Mask2Former Swin-L | ADE20K | - | 56.4 |
| GenPercept (Train U-Net + UperNet) | ADE20K | - | 50.2 |

Table 10: Quantitative comparisons of image matting on the P3M-500-NP and AIM500.

| Method | Test Dataset | SAD ↓ | MAD ↓ | MSE ↓ | CONN ↓ |
|---|---|---|---|---|---|
| HATT (Qiao et al., 2020) | P3M-500-NP | 30.35 | 0.0176 | 0.0072 | 27.42 |
| SHM (Chen et al., 2018) | P3M-500-NP | 20.77 | 0.0122 | 0.0093 | 17.09 |
| MODNet (Ke et al., 2022) | P3M-500-NP | 16.70 | 0.0097 | 0.0051 | 13.81 |
| P3M-Net (Li et al., 2021) | P3M-500-NP | 11.23 | 0.0065 | 0.0035 | 12.51 |
| ViTAE-S (Ma et al., 2023) | P3M-500-NP | 7.59 | 0.0044 | 0.0019 | 6.96 |
| Our GenPercept | P3M-500-NP | 12.77 | 0.0074 | 0.0027 | 10.46 |
| ViTAE-S (Ma et al., 2023) | AIM500 (Zero-shot) | 112.52 | 0.0608 | 0.0602 | 43.18 |
| Our GenPercept | AIM500 (Zero-shot) | 75.5 | 0.0444 | 0.0242 | 36.74 |

## 5 RELATED WORK

**Vision Pre-Training.** Models pretrained on large-scale datasets possess powerful feature extraction capabilities, enabling them to be effectively transferred to a wide range of visual tasks. For instance, the ResNet (He et al., 2016) model pretrained on ImageNet (Russakovsky et al., 2015) can be fine-tuned and applied to perception tasks. By means of contrastive learning, MoCo (He et al., 2020) and CLIP (Radford et al., 2021) acquire rich visual and semantic representations, leveraging their advantages in joint visual and semantic modeling to enhance the performance of multimodal tasks. DINO (Caron et al., 2021), through self-distillation, endows Vision Transformer and convolutional networks with comparable visual representation quality and demonstrates that self-supervised ViT representations contain explicit semantic segmentation information. DINOv2 (Oquab et al., 2024) leverages self-supervised learning on a large curated dataset and exhibits remarkable zero-shot generalization capabilities across computer vision tasks at both image and pixel levels, including classification, semantic segmentation, and depth estimation. In our work, we leverage Stable Diffusion (Rombach et al., 2022) as a prior for scene understanding and transfer it to various perception tasks.

**Diffusion Priors for Dense Prediction.** Several works explore to use the priors of generative models for perceptual tasks. Some works (Bhattad et al., 2024; Du et al., 2023) demonstrate that generative models encode property maps of the scene. By finding latent variable offsets, using LoRA (Hu et al., 2022), etc., generative models can directly produce intrinsic images like surface normals, depth, albedo, etc. LDMSeg (Van Gansbeke & De Brabandere, 2024) devises an image-conditioned sampling process, enabling diffusion models to directly output panoptic segmentation. UniGS (Qi et al., 2023) proposes location-aware color encoding and decoding strategies, allowing diffusion models to support referring segmentation and entity segmentation. Marigold (Ke et al., 2024) fine-tunes diffusion model on limited synthetic data, enabling it to support affine-invariant monocular depth estimation and exhibit strong generalization performance. However, Marigold is time-consuming due to the need for multiple iterations of denoising. Additionally, the Gaussian noise leads to inconsistent results across inferences, requiring aggregation over multiple inferences. Xiang et al. (2023) train a denoising auto-encoder for image classification. The difference of their method compared with traditional denoising auto-encoder is that input images are encoded into a latent code and denoising is performed in the latent space rather than the pixel space. They show good results on very small-scale datasets (CIFAR and ImageNet-tiny) to prove the concept and no results were reported on larger datasets. Furthermore, GeoWizard (Fu et al., 2024a) extends the generative capabilities of Marigold, achieving better performance in joint depth and normal estimation, which enhances applications like 3D reconstruction and novel view synthesis. Moreover, DepthFM (Gui et al., 2024) addresses the speed challenge of Marigold by employing flow matching, offering a fast and efficient monocular depth estimation model.

## 6 CONCLUSION

In this work, we introduce GenPercept, an embarrassingly straightforward yet powerful approach to re-use the off-the-shelf UNet trained using diffusion processes. GenPercept demonstrates the capability to effectively leverage pre-trained diffusion models across a range of downstream dense perception tasks. We contend that our proposed methodology provides an efficient and potent paradigm for harnessing the capabilities of pre-trained diffusion models in dense visual perception tasks.

For future research, we suggest investigating the impact of scaling up the volume of fine-tuning data and exploring the key components of pre-training by applying alternative self-supervised pre-training methods on the LAION dataset, such as Masked Autoencoders (MAE) or Contrastive Language-Image Pretraining (CLIP). It will be helpful to clarify whether the highly detailed visual predictions produced by existing diffusion models are primarily driven by the extensive LAION dataset or the diffusion pretraining paradigm itself.

ACKNOWLEDGMENTS

This work is partially supported by the National Key R&D Program of China (NO.2022ZD0160160101), Ningbo Science and Technology Bureau (Grant Number 2024Z291) and the National Natural Science Foundation of China (No. 62206244). Y. Ge was with The University of Adelaide and his contribution was made when visiting Zhejiang University. C. Shen is the corresponding author.

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
