# Supplementary Materials of
# What Matters When Repurposing Diffusion Models for General Dense Perception Tasks?

**Guangkai Xu**   **Yongtao Ge**   **Mingyu Liu**   **Chengxiang Fan**
**Kangyang Xie**   **Zhiyue Zhao**   **Hao Chen**   **Chunhua Shen**

## 1 The formulation of Three Different Pipelines

### 1.1 Stochastic Multi-Step Generation

For the training process, the RGB image $\mathbf{x}$ and ground-truth label $\mathbf{y}$ are encoded into the latent space with the VAE encoder $\mathbf{z}^{(\mathbf{x})} = \mathcal{E}(\mathbf{x})$, $\mathbf{z}^{(\mathbf{y})} = \mathcal{E}(\mathbf{y})$. The Gaussian noise $\boldsymbol{\epsilon}$ is added to the ground-truth label latent $\mathbf{z}^{(\mathbf{y})}$, and the noisy label latent $\mathbf{z}_t^{(\mathbf{y})}$ is concatenated with the clean image latent $\mathbf{z}^{(\mathbf{x})}$ as U-Net input $\mathbf{z}_t$ for each timestep:

$$\mathbf{z}_t^{(\mathbf{y})} = \sqrt{\bar{\alpha}_t}\mathbf{z}^{(\mathbf{y})} + \sqrt{1 - \bar{\alpha}_t}\boldsymbol{\epsilon}, \;\; t = [1, ..., T],$$
$$\mathbf{z}_t = \text{concat}(\mathbf{z}_t^{(\mathbf{y})}, \mathbf{z}^{(\mathbf{x})}), \tag{1}$$

where $\bar{\alpha}_t = \prod_{s=1}^{t}(1 - \beta_s)$, and $\beta_s$ is sampled from a variance scheduler $\{\beta_t \in (0, 1)\}_{t=1}^{T}$. The scheduler is parameters with two hyperparameters $\beta_{start}$ and $\beta_{end}$, which defines the $\beta_t$ values of t=0 and t=1000, respectively. For a casual timestep $s$, $\beta_s$ is computed by linearly interpolating between $\sqrt{\beta_{start}}$ and $\sqrt{\beta_{end}}$, then squaring each interpolated value. The formulation is as follows.

$$\beta_s = (\sqrt{\beta_{start}} + \frac{s}{T}(\sqrt{\beta_{end}} - \sqrt{\beta_{start}}))^2, \;\; s = [1, ..., T] \tag{2}$$

Then, the denoiser $\boldsymbol{v}_\theta(\cdot, \cdot)$ is enforced to learn the "v-prediction" (Salimans & Ho, 2021) from a timestep $t$ and the corresponding input $\mathbf{z}_t$. During training, the parameters of VAE are frozen, and only the denoiser $\boldsymbol{v}_\theta$ is fine-tuned. The timestep is uniformly sampled from 1 to $T$.

$$\mathcal{L} = \mathbb{E}_{\mathbf{z}^{(\mathbf{y})}, \boldsymbol{\epsilon} \sim \mathcal{N}(0, I), t \sim \mathcal{U}(T)} \left\| (\sqrt{\bar{\alpha}_t}\boldsymbol{\epsilon} - \sqrt{1 - \bar{\alpha}_t}\mathbf{z}^{(\mathbf{y})}) - \boldsymbol{v}_\theta(\mathbf{z}_t, t) \right\|_2^2. \tag{3}$$

For inference, a Gaussian noise $\boldsymbol{\epsilon}_t$ is randomly sampled and denoised step by step with the denoiser $\boldsymbol{v}_\theta(\cdot, \cdot)$.

$$\hat{\mathbf{z}}_T^{(\mathbf{y})} = \boldsymbol{\epsilon}, \;\; \mathbf{z}_t = \text{concat}(\hat{\mathbf{z}}_t^{(\mathbf{y})}, \mathbf{z}^{(\mathbf{x})}), \;\; t = [T, ..., 1],$$
$$\hat{\mathbf{z}}_{t \to 0}^{(\mathbf{y})} = \sqrt{\bar{\alpha}_t} \cdot \hat{\mathbf{z}}_t^{(\mathbf{y})} - \sqrt{1 - \bar{\alpha}_t} \cdot \boldsymbol{v}_\theta(\mathbf{z}_t, t), \;\; \hat{\boldsymbol{\epsilon}}_t = \sqrt{\bar{\alpha}_t} \cdot \boldsymbol{v}_\theta(\mathbf{z}_t, t) - \sqrt{1 - \bar{\alpha}_t} \cdot \hat{\mathbf{z}}_t^{(\mathbf{y})}, \tag{4}$$
$$\hat{\mathbf{z}}_{t-1}^{(\mathbf{y})} = \sqrt{\bar{\alpha}_{t-1}} \cdot \hat{\mathbf{z}}_{t \to 0}^{(\mathbf{y})} + \sqrt{1 - \bar{\alpha}_{t-1}} \cdot \hat{\boldsymbol{\epsilon}}_t, \;\; \hat{\mathbf{y}} = \mathcal{D}(\hat{\mathbf{z}}_{1 \to 0}^{(\mathbf{y})}).$$

where the denoising process first computes the estimated noise $\hat{\boldsymbol{\epsilon}}_t$ and the predicted clean latent code $\hat{\mathbf{z}}_{t \to 0}^{(\mathbf{y})}$ of timestep $t$ from the current latent code $\hat{\mathbf{z}}_t^{(\mathbf{y})}$ and the predicted velocity $\boldsymbol{v}_\theta(\mathbf{z}_t, t)$. Then, it adds the computed noise $\hat{\boldsymbol{\epsilon}}_t$ back to $\hat{\mathbf{z}}_{t \to 0}^{(\mathbf{y})}$ to get the latent code of timestep $t - 1$. After repeating it for $T$ times, the predicted clean latent code $\hat{\mathbf{z}}_{1 \to 0}^{(\mathbf{y})}$ is computed and sent to the VAE decoder $\mathcal{D}$ and estimate the target label $\hat{\mathbf{y}}$. During inference, $m$ randomly sampled noises are introduced to estimate $m$ different predictions, and they are averaged with an ensemble process (Ke et al., 2024) to reduce the randomness of prediction and improve performance.

### 1.2 Deterministic Multi-Step Generation

The inherent random nature of diffusion models makes them challenging to apply to perceptual tasks, which typically aim for accurate results. As a result, existing works have replaced the original

Table 1: Runtime comparison of three diffusion for perception pipelines on an RTX 4090 GPU.

| Experimental Setting | Ensemble | Denoise Steps | Inference Time | GPU Memory |
|---|---|---|---|---|
| Stochastic Multi-step Generation (w. ensemble) | 10 | 10 | ~5.74s | 16GB |
| Stochastic Multi-step Generation (w/o ensemble) | 1 | 10 | ~0.79s | 6.95GB |
| Deterministic Multi-step Generation | 1 | 10 | ~0.79s | 6.95GB |
| Deterministic One-step Inference (Ours) | 1 | 1 | ~0.34s | 6.95GB |
| Deterministic One-step Inference + DPT head (Ours) | 1 | 1 | ~0.24s | 6.32GB |
| Metric3Dv2 (Hu et al., 2024) | 1 | 1 | ~0.25s | 2.63GB |
| DepthAnythingv2 (Yang et al., 2024) | 1 | 1 | ~0.07s | 2.82GB |
| DSINE (Bae & Davison, 2024a) | 1 | 1 | ~0.18s | 2.23GB |
| Marigold (Ke et al., 2024) | 1 | 10 | ~0.79s | 6.95GB |
| GeoWizard (Fu et al., 2024) | 1 | 1 | ~1.32s | 6.81GB |
| DepthFM (Gui et al., 2024) | 1 | 2 | ~0.41s | 6.97GB |
| Our GenPercept (DPT head) | 1 | 1 | ~0.24s | 6.32GB |

Table 2: The impact of training data volume on affine-invariant monocular depth estimation.

| Amount of Data | KITTI | | NYU | | ScanNet | | DIODE | | ETH3D | |
|---|---|---|---|---|---|---|---|---|---|---|
| | AbsRel↓ | $\delta_1$↑ | AbsRel↓ | $\delta_1$↑ | AbsRel↓ | $\delta_1$↑ | AbsRel↓ | $\delta_1$↑ | AbsRel↓ | $\delta_1$↑ |
| 90K (1/1) | 0.100 | 0.902 | 0.053 | 0.966 | 0.059 | 0.961 | 0.309 | 0.768 | 0.068 | 0.956 |
| 45K (1/2) | 0.101 | 0.902 | 0.056 | 0.964 | 0.058 | 0.963 | 0.311 | 0.764 | 0.070 | 0.955 |
| 22.5K (1/4) | 0.109 | 0.884 | 0.056 | 0.963 | 0.059 | 0.963 | 0.322 | 0.754 | 0.073 | 0.950 |
| 11.2K (1/8) | 0.117 | 0.866 | 0.060 | 0.962 | 0.065 | 0.957 | 0.324 | 0.753 | 0.076 | 0.943 |
| 5.6K (1/16) | 0.117 | 0.868 | 0.063 | 0.958 | 0.068 | 0.952 | 0.331 | 0.744 | 0.084 | 0.932 |

Gaussian noise with the target image as RGB noise (Bansal et al., 2024; Lee et al., 2024). Technically, rather than introducing the random Gaussian noise $\epsilon$, we blend the ground-truth label latent $\mathbf{z}^{(\mathbf{y})} = \mathcal{E}(\mathbf{y})$ with the RGB image latent $\mathbf{z}^{(\mathbf{x})} = \mathcal{E}(\mathbf{x})$, which is formulated as:

$$\mathbf{z}_t = \mathbf{z}_t^{(\mathbf{y})} = \sqrt{\bar{\alpha}_t}\mathbf{z}^{(\mathbf{y})} + \sqrt{1 - \bar{\alpha}_t}\mathbf{z}^{(\mathbf{x})}, \quad t = [1, ..., T], \tag{5}$$

Furthermore, the input latent has been modified to the latent code of input image $\mathbf{z}^{(\mathbf{x})}$ instead of random Gaussian. Consequently, the learning objective function and the inference process are reformulated as follows.

$$\mathcal{L} = \mathbb{E}_{(\mathbf{z}^{(\mathbf{x})}, \mathbf{z}^{(\mathbf{y})}), t \sim \mathcal{U}(T)} \left\| (\sqrt{\bar{\alpha}_t}\mathbf{z}^{(\mathbf{x})} - \sqrt{1 - \bar{\alpha}_t}\mathbf{z}^{(\mathbf{y})}) - \boldsymbol{v}_\theta(\mathbf{z}_t, t) \right\|_2^2. \tag{6}$$

$$\hat{\mathbf{z}}_T^{(\mathbf{y})} = \mathbf{z}^{(\mathbf{x})}, \quad t = [T, ..., 1],$$
$$\hat{\mathbf{z}}_{t \to 0}^{(\mathbf{y})} = \sqrt{\bar{\alpha}_t} \cdot \hat{\mathbf{z}}_t^{(\mathbf{y})} - \sqrt{1 - \bar{\alpha}_t} \cdot \boldsymbol{v}_\theta(\hat{\mathbf{z}}_t^{(\mathbf{y})}, t), \hat{\mathbf{z}}_t^{(\mathbf{x})} = \sqrt{\bar{\alpha}_t} \cdot \boldsymbol{v}_\theta(\hat{\mathbf{z}}_t^{(\mathbf{y})}, t) - \sqrt{1 - \bar{\alpha}_t} \cdot \hat{\mathbf{z}}_t^{(\mathbf{y})}, \tag{7}$$
$$\hat{\mathbf{z}}_{t-1}^{(\mathbf{y})} = \sqrt{\bar{\alpha}_{t-1}} \cdot \hat{\mathbf{z}}_{t \to 0}^{(\mathbf{y})} + \sqrt{1 - \bar{\alpha}_{t-1}} \cdot \hat{\mathbf{z}}_t^{(\mathbf{x})}, \quad \hat{\mathbf{y}} = \mathcal{D}(\hat{\mathbf{z}}_{1 \to 0}^{(\mathbf{y})}).$$

where $\hat{\mathbf{z}}_t^{(\mathbf{x})}$ denotes the predicted RGB noise of timestep $t$.

## 1.3 GENPERCEPT: DETERMINISTIC ONE-STEP PERCEPTION

In the main paper, We set the $(\beta_{start}, \beta_{end})$ values to 1, and $\bar{\alpha}_t = \prod_{s=1}^{t}(1 - \beta_s) = 0$, the formulation of Eq. (5) to Eq. (7) can be greatly simplified as follows.

$$\mathbf{z}_t = \mathbf{z}_t^{(\mathbf{y})} = \mathbf{z}^{(\mathbf{x})}, \quad \mathcal{L} = \mathbb{E}_{(\mathbf{z}^{(\mathbf{x})}, \mathbf{z}^{(\mathbf{y})}), t \sim \mathcal{U}(T)} \left\| -\mathbf{z}^{(\mathbf{y})} - \boldsymbol{v}_\theta(\mathbf{z}_t, t) \right\|_2^2,$$
$$\hat{\mathbf{z}}_T^{(\mathbf{y})} = \mathbf{z}^{(\mathbf{x})}, \quad \hat{\mathbf{z}}_{t-1}^{(\mathbf{y})} = -\hat{\mathbf{z}}_t^{(\mathbf{y})}, \quad \hat{\mathbf{z}}_{1 \to 0}^{(\mathbf{y})} = -\boldsymbol{v}_\theta(\hat{\mathbf{z}}_1^{(\mathbf{y})}, t = 1), \quad \hat{\mathbf{y}} = \mathcal{D}(\hat{\mathbf{z}}_{1 \to 0}^{(\mathbf{y})}). \tag{8}$$

**One-step prediction.** The input of the U-Net is an RGB latent code, and the output becomes the negative value of the ground-truth latent code, with no relationship to the timestep $t$. Therefore, we set the number of timesteps $T$ to 1 while preserving the same performance.

$$\mathcal{L} = \mathbb{E}_{(\mathbf{z}^{(\mathbf{x})}, \mathbf{z}^{(\mathbf{y})})} \left\| -\mathbf{z}^{(\mathbf{y})} - \boldsymbol{v}_\theta(\mathbf{z}^{(\mathbf{x})}, t = 1) \right\|_2^2,$$
$$\hat{\mathbf{z}}_{1 \to 0}^{(\mathbf{y})} = -\boldsymbol{v}_\theta(\mathbf{z}^{(\mathbf{x})}, t = 1), \quad \hat{\mathbf{y}} = \mathcal{D}(\hat{\mathbf{z}}_{1 \to 0}^{(\mathbf{y})}). \tag{9}$$

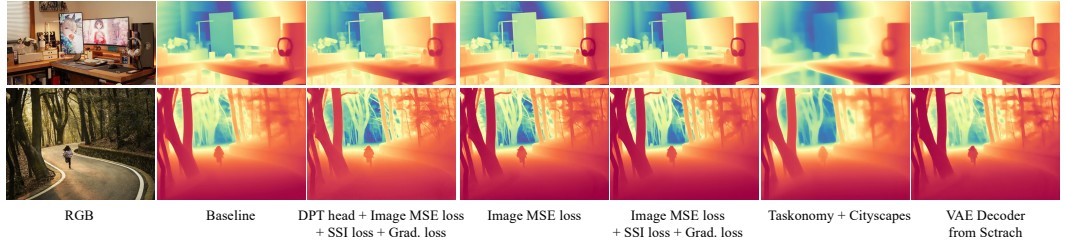

| RGB | Baseline | DPT head + Image MSE loss + SSI loss + Grad. loss | Image MSE loss | Image MSE loss + SSI loss + Grad. loss | Taskonomy + Cityscapes | VAE Decoder from Sctrach |

Figure 1: Qualitative visualization of ablation study.

## 2 A DIFFERENCE BETWEEN IMAGE GENERATION AND VISUAL PERCEPTION

In text-guided image generation, a single textual input can correspond to an immense variety of potential images. This inherent uncertainty makes generating a high-quality image directly from random noise in a single step extremely challenging. Therefore, the multi-step generation enables the model to incrementally remove noise, progressively refining details and textures at each stage, which effectively simplifies the task. However, visual perception tasks conditioned on an RGB image are deterministic without any randomness, and such an easy injective mapping can be estimated with a one-step inference process, as most of the traditional visual perception methods do.

While Marigold series algorithms aim to leverage diffusion models' ability of generating highly detailed images to enhance visual perception with precise details, reformulating straightforward deterministic tasks as a denoising process can further simplify this problem, enforcing the network to exploit "shortcuts", as described in Section 3.1 of the main paper and illustrated in Fig. 2.

## 3 RUNTIME ANALYSIS

In this section, we quantitatively analyze the inference times of the three aforementioned pipelines, as summarized in Table 1. The runtime is evaluated by averaging the inference times over 100 images with a resolution of $768 \times 768$, using an RTX 4090 GPU. For "Stochastic Multi-step Generation" methods (Ke et al., 2024; Fu et al., 2024; Gui et al., 2024), such as Marigold (Ke et al., 2024), they rely on an ensemble process where multiple inferences are performed with varying random noise inputs to mitigate uncertainties introduced by Gaussian noise. Consequently, this approach is computationally expensive. On the other hand, "Deterministic Multi-step Generation" methods (Lee et al., 2024) involve multiple denoising steps, which significantly reduce inference efficiency.

In contrast, our proposed one-step inference paradigm demonstrates a runtime that is 94% and 57% less than those of multi-step methods with ensemble and without ensemble, respectively. Furthermore, by incorporating a customized head, such as the DPT head (Ranftl et al., 2021), both runtime and GPU memory requirements are further reduced by 27% without compromising performance, maintaining a competitive level of accuracy and robustness.

Compared to existing state-of-the-art diffusion-based methods, our proposed GenPercept achieves a notable improvement in inference speed, attributed to the innovative one-step inference paradigm and the customized head. While our method demonstrates inference speeds comparable to Metric3Dv2 (Hu et al., 2024) and DSINE (Bae & Davison, 2024a), it falls behind DepthAnythingV2 (Yang et al., 2024). Note that the superior performance of DepthAnythingV2 is facilitated by its training on a relatively lightweight model, bolstered by extensive labeled and unlabeled datasets, and supported by substantial computational resources distributed across multiple GPUs.

## 4 THE IMPACT OF DATA VOLUME

With only around 50K Hypersim (Roberts et al., 2021) and 40K Virtual KITTI (Cabon et al., 2020) fine-tuning data, GenPercept can generalize well to diverse tasks and datasets. How much data is needed for transferring at least? We gradually reduce the amount of training data. As shown in

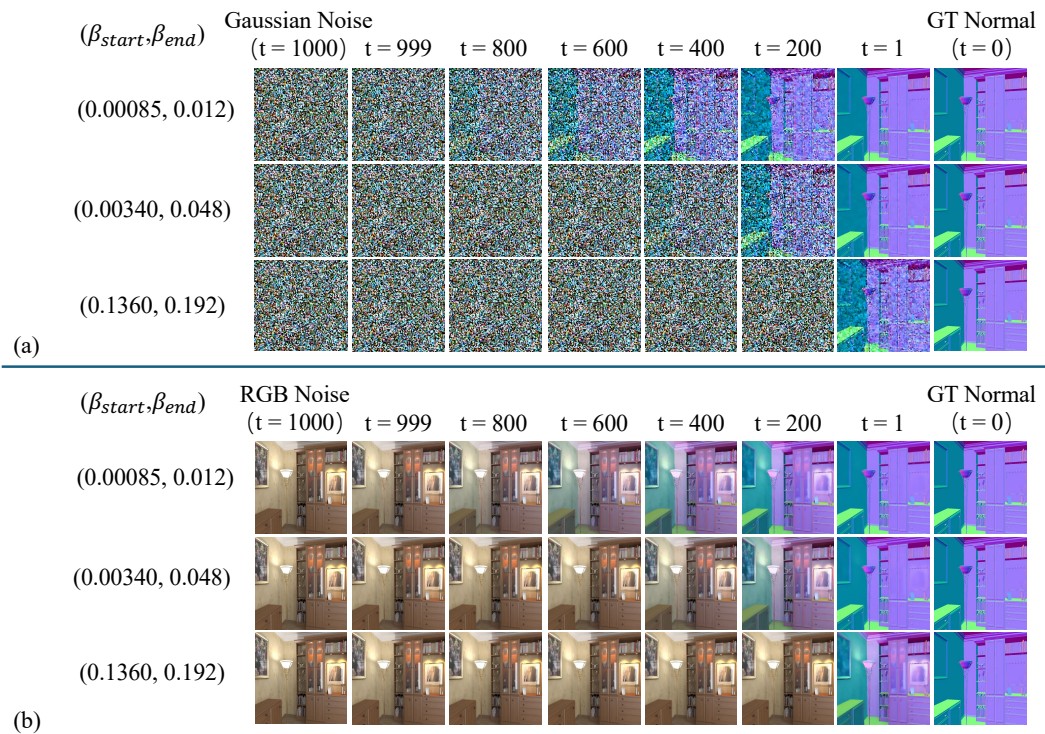

Figure 2: Visualization of different noise forms and proportions in the forward diffusion process.

Table 2, less training data results in slightly worse performance, but GenPercept still shows much robustness to the data volume.

## 5    MORE VISUALIZATION ANALYSIS

**Quantitative Comparisons of Ablations.** Visualization of the ablation study experiments in the main paper §2 is shown in Fig. 1. The estimated depth detail remains comparable with a customized "DPT head" (Ranftl et al., 2021). Models trained with lower-quality data like "Taskonomy + Cityscapes" or without the pre-trained VAE decoder parameters suffer from a decline in the ability to predict details. We attribute them to the low-quality annotation and the oversized decoder, respectively.

**Forward Diffusion Process.** More detailed visualization of different noise forms and proportions in the forward diffusion process is shown in Fig. 2. With larger $(\beta_{start}, \beta_{end})$ values of the DDIM scheduler, the proportion of noise will be much higher during the forward diffusion process. When $(\beta_{start}, \beta_{end})$ reaches (1.0, 1.0), the noisy latent will be pure noise latent, which is Gaussian noise and RGB latent for (a) and (b), respectively.

**Quantitative Comparisons of Generalization.** We compare the generalization performance of models trained on synthetic (50K Hypersim (Roberts et al., 2021) + 40K Virtual KITTI (Cabon et al., 2020)) and real data (50K Taskonomy (Zamir et al., 2018) + 40K Cityscapes (Cordts et al., 2016)) for out-of-distribution scenarios. As illustrated in Fig. 3, the robustness of GenPercept trained on real data is comparable to that trained on synthetic data in challenging scenes, such as underwater environments, non-realistic renderings, and evening settings. Notably, models trained on synthetic data demonstrate superior accuracy in estimating transparent objects and capturing geometric details, owing to the high-quality and densely labeled synthetic ground truth.

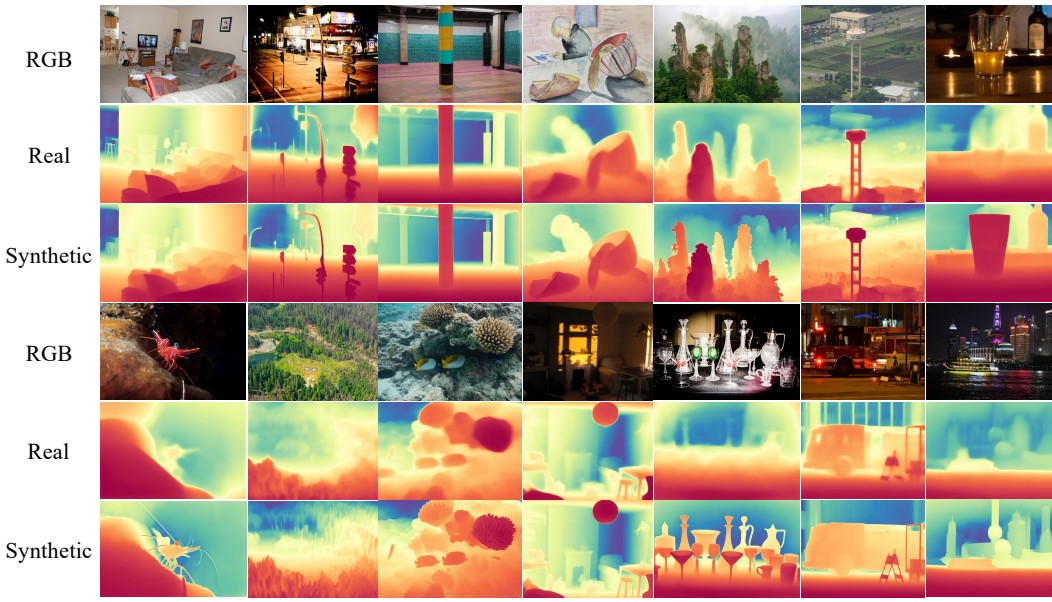

Figure 3: Quantitative comparisons of generalization for models trained on synthetic and real data.

## 6 AN EXTRA ATTEMPT ON HUMAN POSE ESTIMATION

**Task Definition** Human Pose Estimation is a task aimed at determining the spatial configuration of a person or object in a given image or video. This involves identifying and predicting the coordinates of particular keypoints.

**Implementation Details** For human keypoint detection, we use Simple Baseline (Xiao et al., 2018) for person detection and conduct training on the COCO training set with 15K training samples. Performance is evaluated on the COCO (Lin et al., 2014) validation set. As shown in Fig. 4, it is generalizable to unseen objects in the training set. To evaluate the performance on COCO, we use the customized keypoint head of ViTPose (Xu et al., 2022) for decoding the output. The quantitative results compared with a generalist model, Painter(Wang et al., 2023) is shown in Table 3.

Table 3: Pose estimation on COCO.

| Metrics | AP ↑ | AP .5 ↑ | AP .75 ↑ | AP (M) ↑ | AP (L) ↑ |
|---|---|---|---|---|---|
| Painter (Wang et al., 2023) | 0.721 | 0.900 | 0.781 | 0.686 | **0.786** |
| GenPercept | **0.752** | **0.907** | **0.824** | **0.691** | 0.778 |

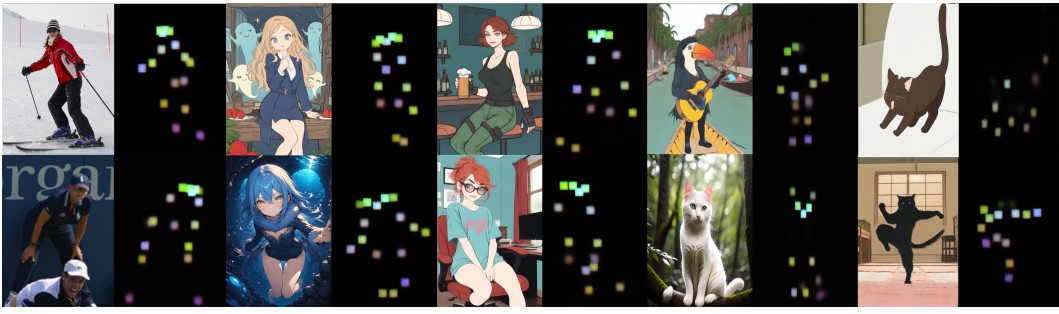

Figure 4: Generalized test results of keypoint detection.

## 7 IMPLEMENTATION DETAILS

**Implementation Details of Exploration Experiments in Section 3 of the Main Paper.** Unless specified otherwise, we follow Marigold's training setting and train for 30000 iterations to estimate the affine-invariant monocular depth. The training dataset contains 50K Hypersim (Roberts et al., 2021) images and 40K Virtual KITTI (Cabon et al., 2020) images, and these images are sampled with a sampling rate of 90% for Hypersim and 10% for Virtual KITTI. We freeze the VAE AutoEncoder and fine-tune the U-Net of Stable Diffusion v2.1 to estimate the ground-truth label latent, with a resolution of (480, 640), a batch size of 32, and a learning rate of 3e-5. The multi-resolution noise (Ke et al., 2024) is employed for Gaussian noise and not used for RGB noise (Lee et al., 2024) by default. For inference, the ensemble size and denoising steps are set to 1 and 10, respectively. Evaluation results of absolute relative error (AbsRel) and $\delta_1$ are reported on five unseen monocular depth datasets, including KITTI (Geiger et al., 2013), NYU (Silberman et al., 2012), ScanNet (Dai et al., 2017), DIODE (Vasiljevic et al., 2019), and ETH3D (Schops et al., 2017).

## 8 MORE QUANTITATIVE AND QUALITIVE EXPERIMENTS

### 8.1 MONOCULAR DEPTH ESTIMATION

**More Qualitative Evaluation.** We show the robustness of the monocular depth estimation model in diverse scenes in Fig. 5. Compared to DPT (Ranftl et al., 2021), our Genpercept performs better on estimating details and shows much better robustness even on some sketches. Compared to Marigold (Ke et al., 2024), our method achieves better relative depth visualization.

### 8.2 SURFACE NORMAL ESTIMATION

**More Qualitative Evaluation.** In Fig. 6, we showcase more qualitative results for the surface normal estimation. DSINE (Bae & Davison, 2024b) is trained on 160K images of 10 datasets, including both real and synthetic data. Our GenPercept is only trained on one synthetic dataset (Hypersim (Roberts et al., 2021)), and can estimate much more detailed surface normal maps.

### 8.3 DICHOTOMOUS IMAGE SEGMENTATION

**More Quantitative Evaluation.** To conduct a comprehensive evaluation, we compare our approach with numerous previous models including models for medical image segmentation (Ronneberger et al., 2015), salient object detection (Qin et al., 2019; Zhao et al., 2020; Wei et al., 2020; Chen et al., 2020; Qin et al., 2020), camouflaged object detection (Fan et al., 2021a; Mei et al., 2021), semantic segmentation (Zhao et al., 2017; Chen et al., 2018; Wang et al., 2020; Yu et al., 2018; Zhao et al., 2018; Howard et al., 2019; Fan et al., 2021b; Nirkin et al., 2021) and models like IS-Net (Qin et al., 2022) and MVANet (Yu et al., 2024) specifically trained for DIS.

As shown in Table 4, our proposed model significantly outperforms methods like IS-Net across most evaluation metrics on this challenging dataset. Compared with MVANet, which inferences with a resolution of $1024 \times 1024$, our GenPercept achieves slightly lower performance, but the results remains competitive and can highlight the effectiveness of our approach for DIS.

**More Qualitative Evaluation.** As shown in Fig. 8, our method yields refined segmentation results, providing cleaner foreground masks. It also can produce precise outputs for intricate lines.

**Cross Dataset Evaluation.** To test the generalization ability of our model, we randomly select some images from other datasets(Agustsson & Timofte, 2017; Lin et al., 2014; Shao et al., 2019) and in-the-wild images for experiments. As shown in Fig. 7, compared to IS-Net and IS-Net-General-Use(Qin et al., 2022), our approach exhibits finer segmentation quality across diverse images, providing cleaner foreground masks. Compared to MVANet (Yu et al., 2024), GenPercept exhibits enhanced robustness when applied to in-the-wild images. This improvement can be attributed to its large-scale pre-training on the LAION dataset and the extensive parameterization of the diffusion model. In contrast, the backbone of MVANet is pre-trained on the ImageNet dataset (Deng et al., 2009), which may limit its performance in more diverse environments.

Table 4: Quantitative results on DIS5K validation and testing sets.

| Dataset | Metric | UNet | BASNet | GateNet | F³Net | GCPANet | U²Net | SINetV2 | PFNet | PSPNet | DLV3+ | HRNet | BSV1 | ICNet | MBV3 | STDC | HySM | IS-Net | MVANet | Ours | Ours (1024px) |
|---|---|---|---|---|---|---|---|---|---|---|---|---|---|---|---|---|---|---|---|---|---|
| DIS-VD | $maxF_\beta\uparrow$ | 0.692 | 0.731 | 0.678 | 0.685 | 0.648 | 0.748 | 0.665 | 0.691 | 0.691 | 0.660 | 0.726 | 0.662 | 0.697 | 0.714 | 0.696 | 0.734 | 0.791 | 0.904 | 0.857 | 0.877 |
| | $F_\beta^w\uparrow$ | 0.586 | 0.641 | 0.574 | 0.595 | 0.542 | 0.656 | 0.584 | 0.604 | 0.603 | 0.568 | 0.641 | 0.548 | 0.609 | 0.642 | 0.613 | 0.640 | 0.717 | 0.861 | 0.835 | 0.859 |
| | $M\downarrow$ | 0.113 | 0.094 | 0.110 | 0.107 | 0.118 | 0.090 | 0.110 | 0.106 | 0.102 | 0.114 | 0.095 | 0.116 | 0.102 | 0.103 | 0.096 | 0.074 | 0.074 | 0.035 | 0.040 | 0.035 |
| | $S_\alpha\uparrow$ | 0.745 | 0.768 | 0.723 | 0.733 | 0.718 | 0.781 | 0.727 | 0.740 | 0.744 | 0.716 | 0.767 | 0.728 | 0.747 | 0.758 | 0.740 | 0.773 | 0.813 | 0.909 | 0.870 | 0.887 |
| | $E_\phi^m\uparrow$ | 0.785 | 0.816 | 0.783 | 0.800 | 0.765 | 0.823 | 0.798 | 0.811 | 0.802 | 0.796 | 0.824 | 0.767 | 0.811 | 0.817 | 0.814 | 0.856 | 0.858 | 0.937 | 0.934 | 0.941 |
| | $HCE_\gamma\downarrow$ | 1337 | 1402 | 1493 | 1567 | 1555 | 1413 | 1568 | 1606 | 1588 | 1520 | 1560 | 1660 | 1503 | 1625 | 1598 | 1324 | 1116 | 878 | 1511 | 1262 |
| DIS-TE1 | $maxF_\beta\uparrow$ | 0.625 | 0.688 | 0.620 | 0.640 | 0.598 | 0.694 | 0.646 | 0.646 | 0.645 | 0.601 | 0.668 | 0.595 | 0.631 | 0.648 | 0.648 | 0.695 | 0.740 | 0.893 | 0.814 | 0.850 |
| | $F_\beta^w\uparrow$ | 0.514 | 0.595 | 0.517 | 0.549 | 0.495 | 0.601 | 0.558 | 0.552 | 0.557 | 0.506 | 0.579 | 0.474 | 0.535 | 0.595 | 0.562 | 0.597 | 0.662 | 0.823 | 0.814 | 0.827 |
| | $M\downarrow$ | 0.106 | 0.084 | 0.099 | 0.095 | 0.103 | 0.083 | 0.094 | 0.094 | 0.089 | 0.102 | 0.088 | 0.108 | 0.095 | 0.083 | 0.090 | 0.082 | 0.074 | 0.037 | 0.036 | 0.036 |
| | $S_\alpha\uparrow$ | 0.716 | 0.754 | 0.701 | 0.721 | 0.705 | 0.760 | 0.727 | 0.722 | 0.725 | 0.694 | 0.742 | 0.695 | 0.716 | 0.740 | 0.723 | 0.761 | 0.787 | 0.879 | 0.868 | 0.878 |
| | $E_\phi^m\uparrow$ | 0.750 | 0.801 | 0.766 | 0.783 | 0.750 | 0.801 | 0.791 | 0.786 | 0.791 | 0.772 | 0.797 | 0.741 | 0.784 | 0.798 | 0.820 | 0.820 | 0.911 | 0.919 | 0.911 | 0.919 |
| | $HCE_\gamma\downarrow$ | 233 | 220 | 230 | 244 | 271 | 224 | 274 | 253 | 267 | 234 | 262 | 288 | 234 | 274 | 249 | 205 | 149 | 103 | 204 | 165 |
| DIS-TE2 | $maxF_\beta\uparrow$ | 0.703 | 0.755 | 0.702 | 0.712 | 0.673 | 0.756 | 0.700 | 0.720 | 0.724 | 0.681 | 0.747 | 0.680 | 0.716 | 0.743 | 0.720 | 0.759 | 0.799 | 0.925 | 0.876 | 0.880 |
| | $F_\beta^w\uparrow$ | 0.597 | 0.668 | 0.598 | 0.620 | 0.570 | 0.668 | 0.618 | 0.633 | 0.636 | 0.587 | 0.664 | 0.564 | 0.627 | 0.672 | 0.636 | 0.667 | 0.728 | 0.874 | 0.852 | 0.859 |
| | $M\downarrow$ | 0.107 | 0.084 | 0.102 | 0.097 | 0.109 | 0.085 | 0.099 | 0.096 | 0.092 | 0.105 | 0.087 | 0.111 | 0.095 | 0.083 | 0.092 | 0.085 | 0.070 | 0.03 | 0.035 | 0.034 |
| | $S_\alpha\uparrow$ | 0.755 | 0.786 | 0.737 | 0.755 | 0.735 | 0.788 | 0.753 | 0.761 | 0.763 | 0.729 | 0.784 | 0.740 | 0.759 | 0.777 | 0.759 | 0.794 | 0.823 | 0.915 | 0.884 | 0.892 |
| | $E_\phi^m\uparrow$ | 0.796 | 0.836 | 0.804 | 0.820 | 0.786 | 0.833 | 0.823 | 0.829 | 0.828 | 0.813 | 0.840 | 0.781 | 0.826 | 0.856 | 0.834 | 0.832 | 0.858 | 0.944 | 0.938 | 0.938 |
| | $HCE_\gamma\downarrow$ | 474 | 480 | 501 | 542 | 574 | 490 | 593 | 567 | 586 | 516 | 555 | 621 | 512 | 600 | 556 | 451 | 340 | 246 | 480 | 410 |
| DIS-TE3 | $maxF_\beta\uparrow$ | 0.748 | 0.785 | 0.726 | 0.743 | 0.699 | 0.798 | 0.730 | 0.751 | 0.747 | 0.717 | 0.784 | 0.710 | 0.752 | 0.772 | 0.745 | 0.792 | 0.830 | 0.936 | 0.885 | 0.898 |
| | $F_\beta^w\uparrow$ | 0.644 | 0.696 | 0.620 | 0.656 | 0.590 | 0.707 | 0.641 | 0.664 | 0.657 | 0.623 | 0.700 | 0.595 | 0.664 | 0.702 | 0.662 | 0.701 | 0.758 | 0.89 | 0.862 | 0.879 |
| | $M\downarrow$ | 0.098 | 0.083 | 0.103 | 0.092 | 0.109 | 0.079 | 0.096 | 0.092 | 0.092 | 0.102 | 0.080 | 0.109 | 0.091 | 0.078 | 0.090 | 0.079 | 0.064 | 0.031 | 0.035 | 0.032 |
| | $S_\alpha\uparrow$ | 0.780 | 0.798 | 0.747 | 0.773 | 0.748 | 0.809 | 0.766 | 0.777 | 0.774 | 0.749 | 0.805 | 0.757 | 0.780 | 0.794 | 0.771 | 0.811 | 0.836 | 0.92 | 0.883 | 0.896 |
| | $E_\phi^m\uparrow$ | 0.827 | 0.856 | 0.815 | 0.848 | 0.801 | 0.858 | 0.849 | 0.854 | 0.843 | 0.833 | 0.869 | 0.801 | 0.852 | 0.880 | 0.855 | 0.857 | 0.883 | 0.954 | 0.951 | 0.954 |
| | $HCE_\gamma\downarrow$ | 883 | 948 | 972 | 1059 | 1058 | 965 | 1096 | 1082 | 1111 | 999 | 1049 | 1146 | 1001 | 1136 | 1081 | 887 | 687 | 512 | 973 | 809 |
| DIS-TE4 | $maxF_\beta\uparrow$ | 0.759 | 0.780 | 0.729 | 0.721 | 0.670 | 0.795 | 0.699 | 0.731 | 0.725 | 0.715 | 0.772 | 0.710 | 0.749 | 0.736 | 0.731 | 0.782 | 0.827 | 0.911 | 0.848 | 0.874 |
| | $F_\beta^w\uparrow$ | 0.659 | 0.693 | 0.625 | 0.633 | 0.559 | 0.705 | 0.616 | 0.647 | 0.630 | 0.621 | 0.687 | 0.598 | 0.663 | 0.664 | 0.652 | 0.693 | 0.753 | 0.857 | 0.829 | 0.858 |
| | $M\downarrow$ | 0.102 | 0.091 | 0.109 | 0.107 | 0.127 | 0.087 | 0.113 | 0.107 | 0.107 | 0.111 | 0.092 | 0.114 | 0.099 | 0.098 | 0.102 | 0.091 | 0.072 | 0.041 | 0.049 | 0.041 |
| | $S_\alpha\uparrow$ | 0.784 | 0.794 | 0.743 | 0.752 | 0.723 | 0.807 | 0.744 | 0.763 | 0.758 | 0.744 | 0.792 | 0.755 | 0.776 | 0.770 | 0.762 | 0.802 | 0.830 | 0.903 | 0.854 | 0.874 |
| | $E_\phi^m\uparrow$ | 0.821 | 0.848 | 0.803 | 0.825 | 0.767 | 0.847 | 0.824 | 0.838 | 0.815 | 0.820 | 0.854 | 0.788 | 0.837 | 0.848 | 0.841 | 0.842 | 0.870 | 0.944 | 0.938 | 0.947 |
| | $HCE_\gamma\downarrow$ | 3218 | 3601 | 3654 | 3760 | 3678 | 3653 | 3683 | 3803 | 3806 | 3709 | 3864 | 3999 | 3690 | 3817 | 3819 | 3331 | 2888 | 2301 | 3799 | 3321 |
| Overall DIS-TE (1-4) | $maxF_\beta\uparrow$ | 0.708 | 0.752 | 0.694 | 0.704 | 0.660 | 0.761 | 0.693 | 0.712 | 0.710 | 0.678 | 0.743 | 0.674 | 0.711 | 0.729 | 0.710 | 0.757 | 0.799 | 0.916 | 0.863 | 0.875 |
| | $F_\beta^w\uparrow$ | 0.603 | 0.663 | 0.590 | 0.614 | 0.554 | 0.670 | 0.608 | 0.624 | 0.620 | 0.584 | 0.658 | 0.558 | 0.622 | 0.658 | 0.628 | 0.665 | 0.726 | 0.855 | 0.839 | 0.856 |
| | $M\downarrow$ | 0.103 | 0.086 | 0.103 | 0.098 | 0.112 | 0.083 | 0.101 | 0.097 | 0.095 | 0.105 | 0.087 | 0.110 | 0.095 | 0.085 | 0.094 | 0.084 | 0.070 | 0.035 | 0.039 | 0.036 |
| | $S_\alpha\uparrow$ | 0.759 | 0.783 | 0.732 | 0.750 | 0.728 | 0.791 | 0.747 | 0.756 | 0.755 | 0.729 | 0.781 | 0.737 | 0.758 | 0.770 | 0.754 | 0.792 | 0.819 | 0.905 | 0.872 | 0.885 |
| | $E_\phi^m\uparrow$ | 0.798 | 0.835 | 0.797 | 0.819 | 0.776 | 0.835 | 0.822 | 0.827 | 0.819 | 0.810 | 0.840 | 0.778 | 0.825 | 0.850 | 0.832 | 0.834 | 0.858 | 0.938 | 0.936 | 0.939 |
| | $HCE_\gamma\downarrow$ | 1202 | 1313 | 1339 | 1401 | 1395 | 1333 | 1411 | 1427 | 1442 | 1365 | 1432 | 1513 | 1359 | 1457 | 1426 | 1218 | 1016 | 790 | 1364 | 1176 |

It is noteworthy that IS-Net-General-Use is fine-tuned on extra datasets to enhance generalization, which indicates that our method has a stronger generalization ability.

## 8.4 IMAGE MATTING

**Implementation Details.** We utilize P3M10K (Li et al., 2021), the largest portrait matting dataset with high-resolution portrait images along with high-quality alpha masks to train our model. The training set contains 9,421 high-quality images and annotations and the test set P3M-500-NP is composed of 500 public images from the Internet. We train our GenPercept with the pixel-wise MSE loss to further improve the final performance.

Fig. 9 shows some results on the P3M test dataset.

**More Qualitative Evaluation.** In Fig. 10, we showcase more qualitative results for the image matting task. It is worth noting that our model works well in various resolutions, light environments, human poses, and human orientations. More importantly, our GenPercept model trained on human matting images shows much more robustness to other objects compared to existing P3M10K (Li et al., 2021) SOTA method ViTAE-S (Ma et al., 2023), as illustrated in Fig. 11. It shows that the ViTAE-S overfits the human matting task, while GenPercept preserves the generalization ability. Besides, we also train GenPercept on a more general image matting task on the Composition-1K (Xu et al., 2017) dataset. As shown in Fig. 12, GenPercept shows robustness on more types of objects such as semi-transparent objects, hollow objects, etc.

## 8.5 IMAGE SEGMENTATION

**More Qualitative Evaluation.** In Fig. 13, we showcase more qualitative results for the image segmentation task. Our method shows much robustness on the trained categories of complex in-the-wild images. Due to the limited annotation categories and little negative label of "unknown category", it sometimes fails in outdoor scenes and unseen categories such as cats and cars. Please zoom in for better visualization and more details.

## 8.6 HUMAN POSE ESTIMATION

**More Qualitative Evaluation.** In Fig. 14, we showcase more qualitative results for the human pose estimation. We conduct experiments on MHP dataset (Li et al., 2017), and we use mmpose (Contributors, 2020) to render the human pose following the setting of (Bai et al., 2023).

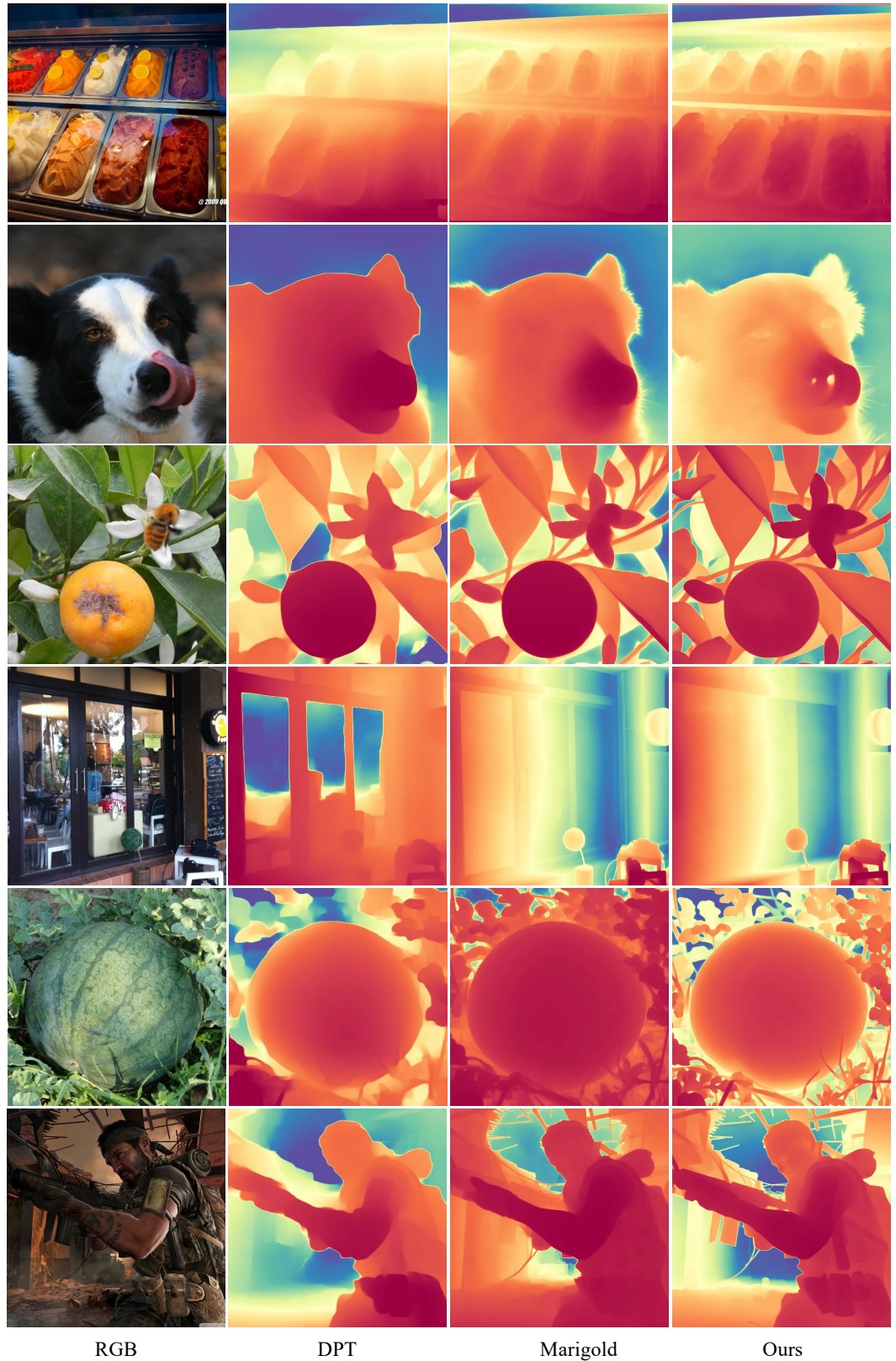

| RGB | DPT | Marigold | Ours |

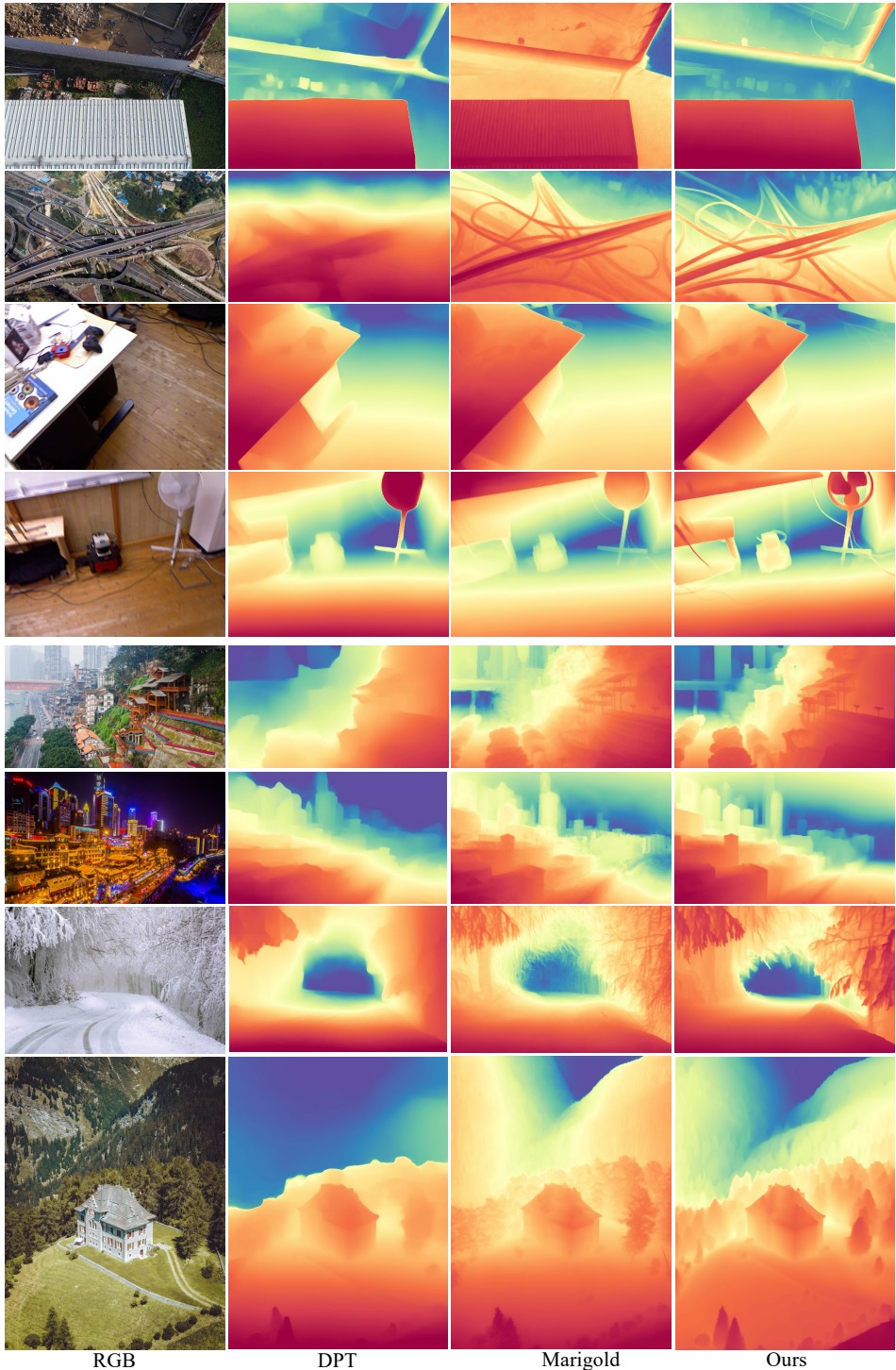

RGB                DPT                Marigold                Ours

Figure 5: More qualitative results for monocular depth estimation.

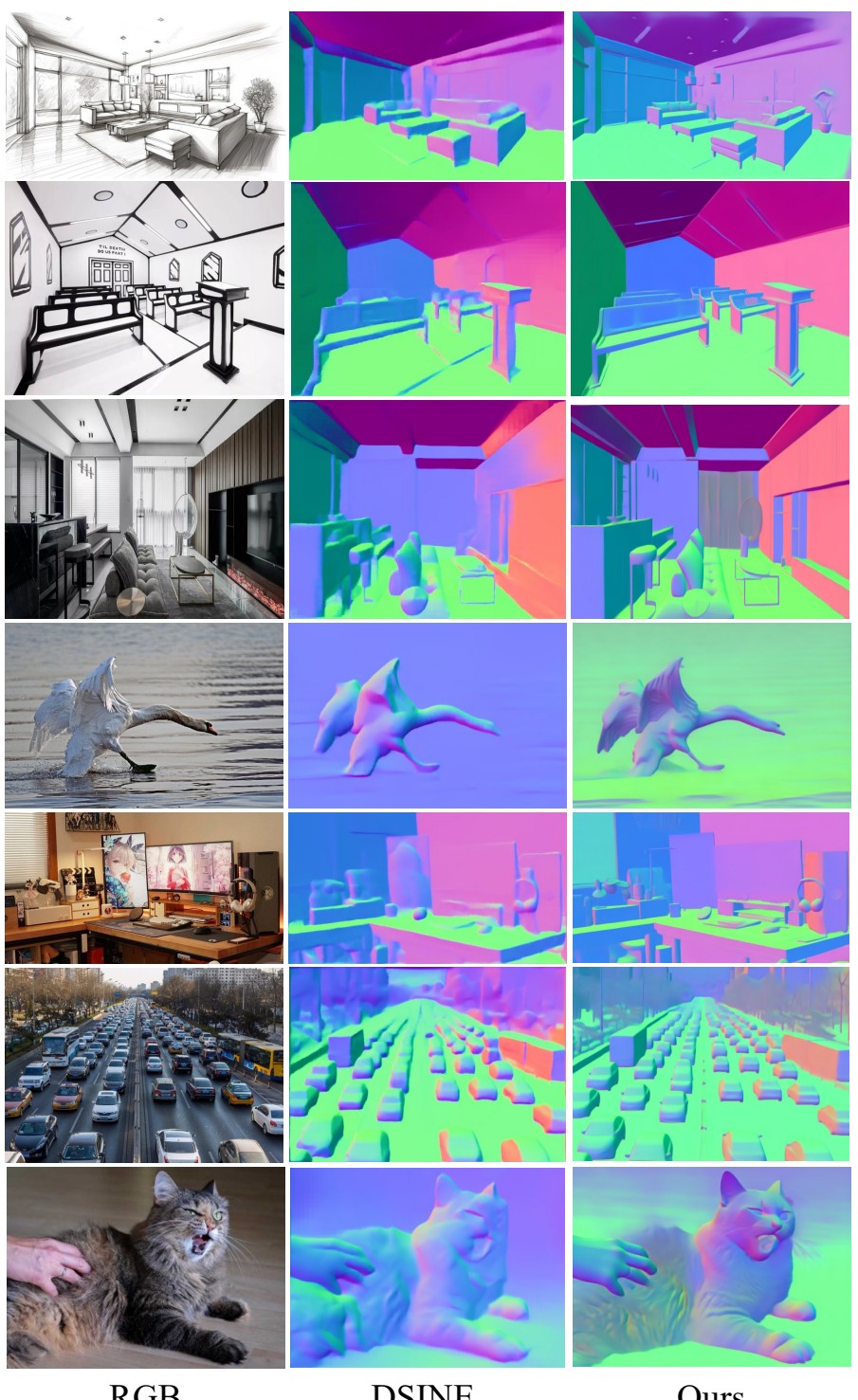

RGB           DSINE           Ours

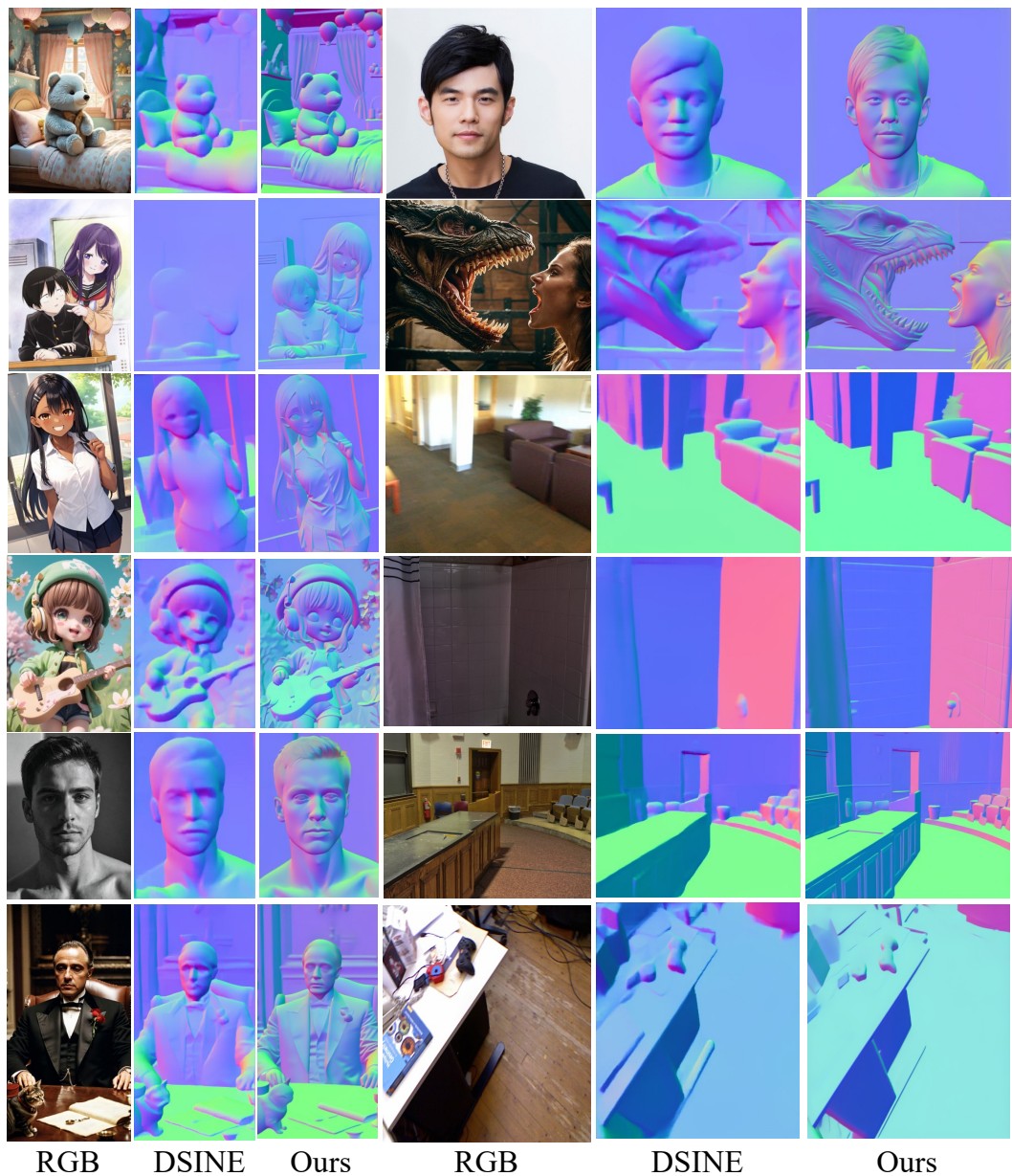

RGB     DSINE     Ours     RGB     DSINE     Ours

Figure 6: More qualitative comparisons of surface normal estimation. Our GenPercept can achieve more detailed results, even compared to the competitive CVPR2024 method DSINE (Bae & Davison, 2024b). Note that these two visualization coordinate systems are slightly different.

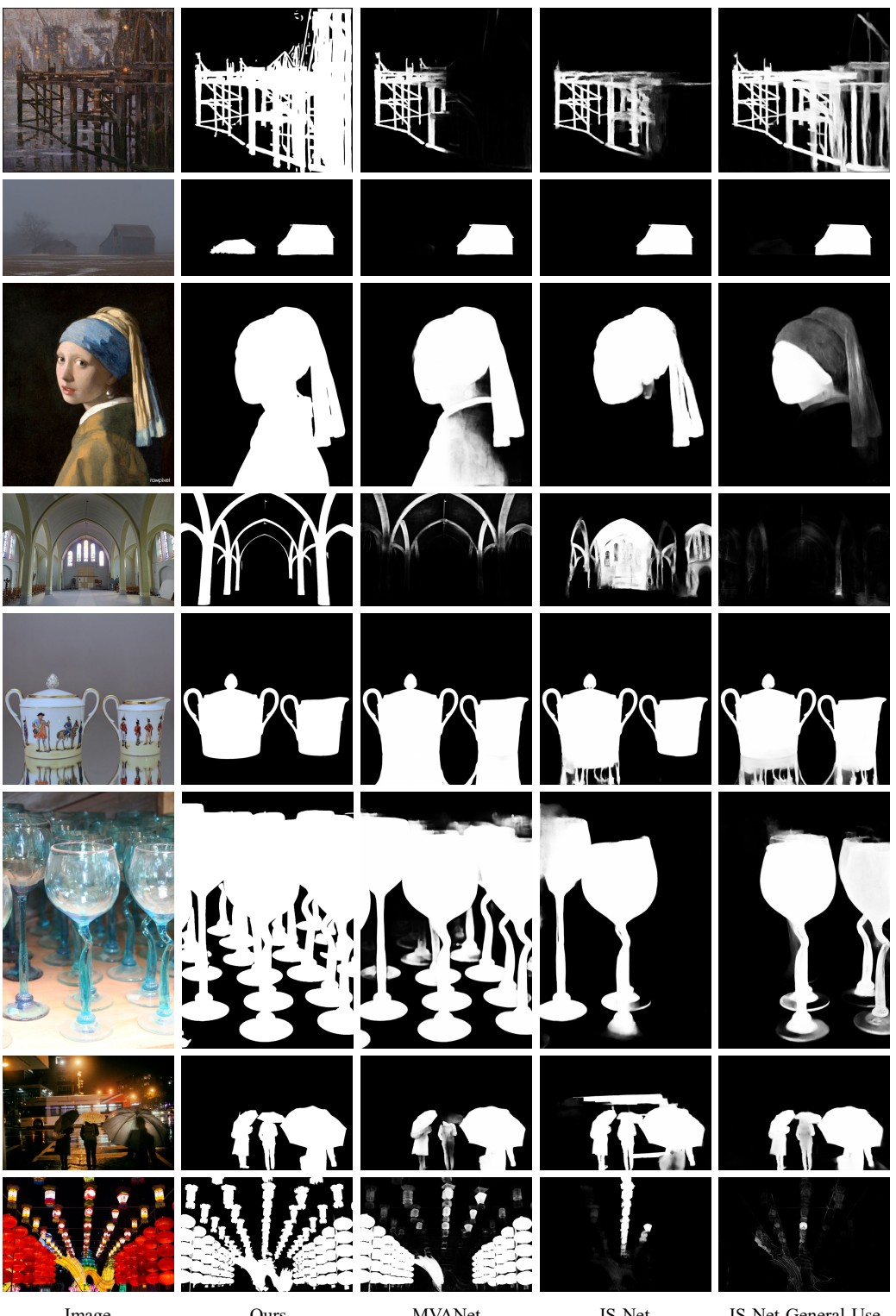

| Image | Ours | MVANet | IS-Net | IS-Net-General-Use |

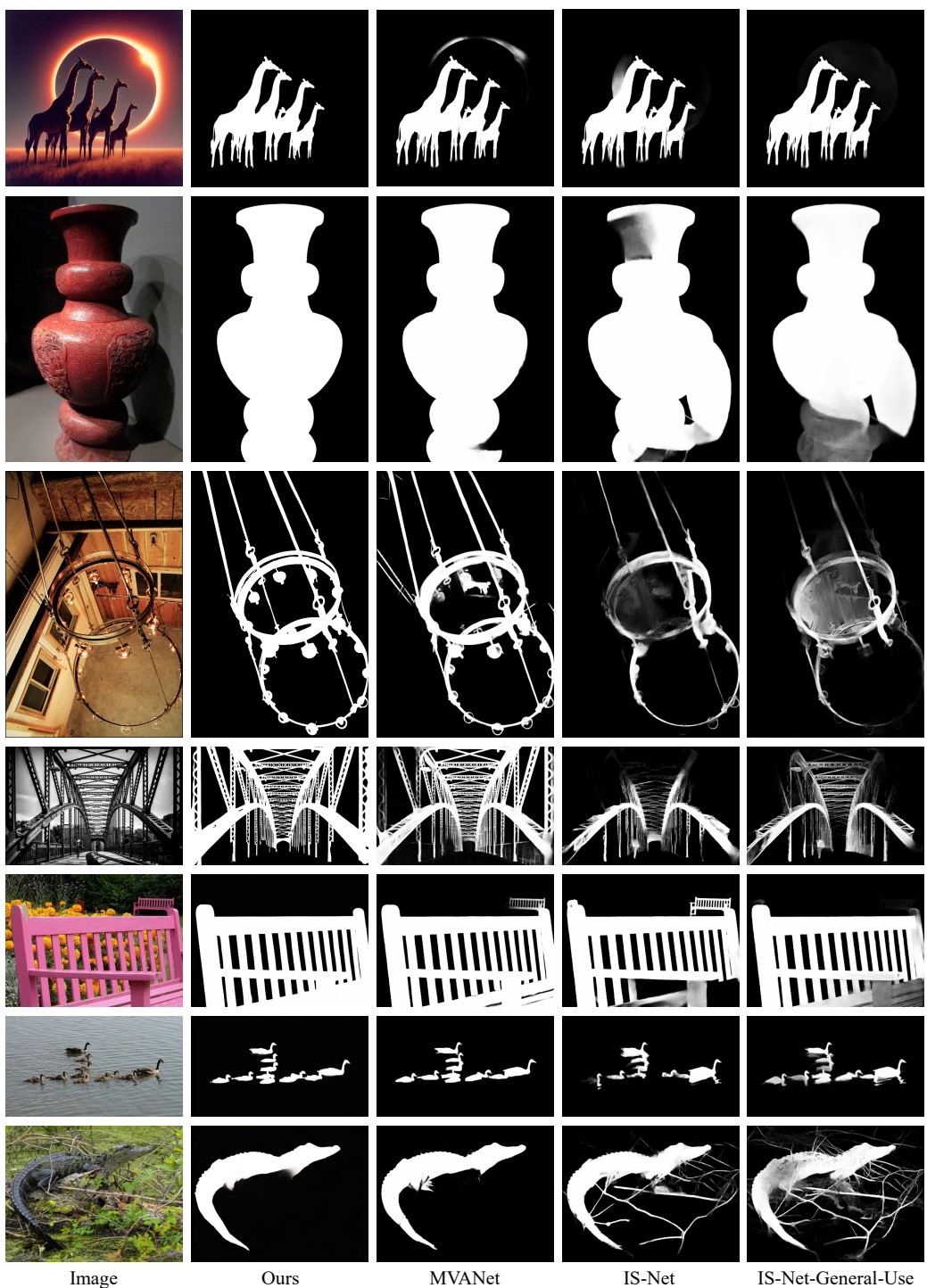

Figure 7: Cross dataset comparison of our model and other models.

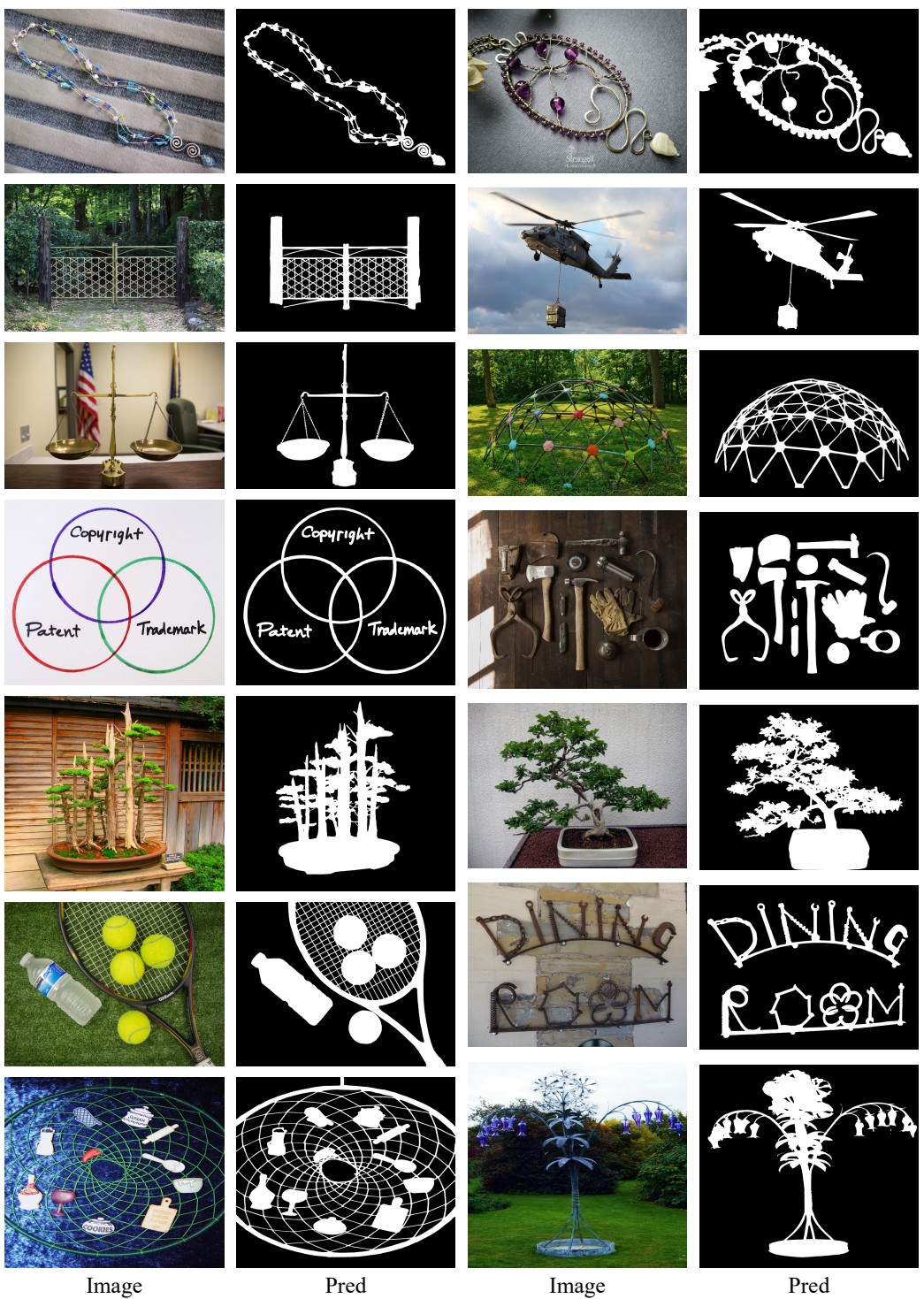

Figure 8: Qualitative results of dichotomous image segmentation.

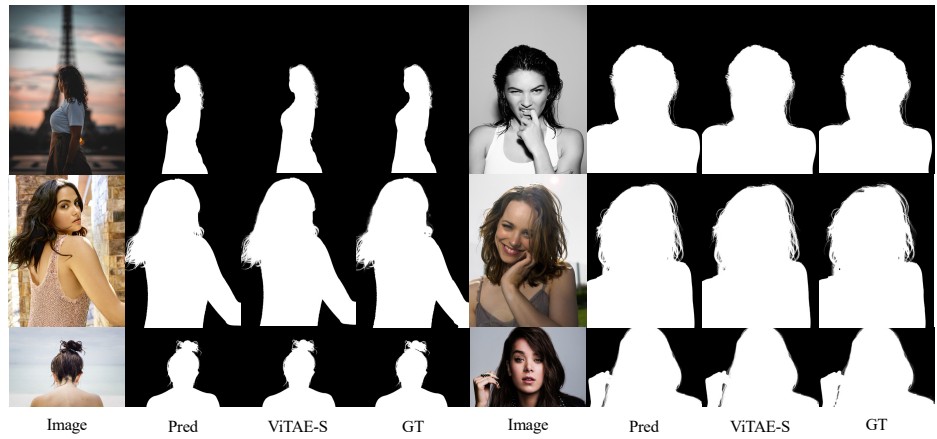

Figure 9: Visualization of image matting on the P3M-500-NP test set.

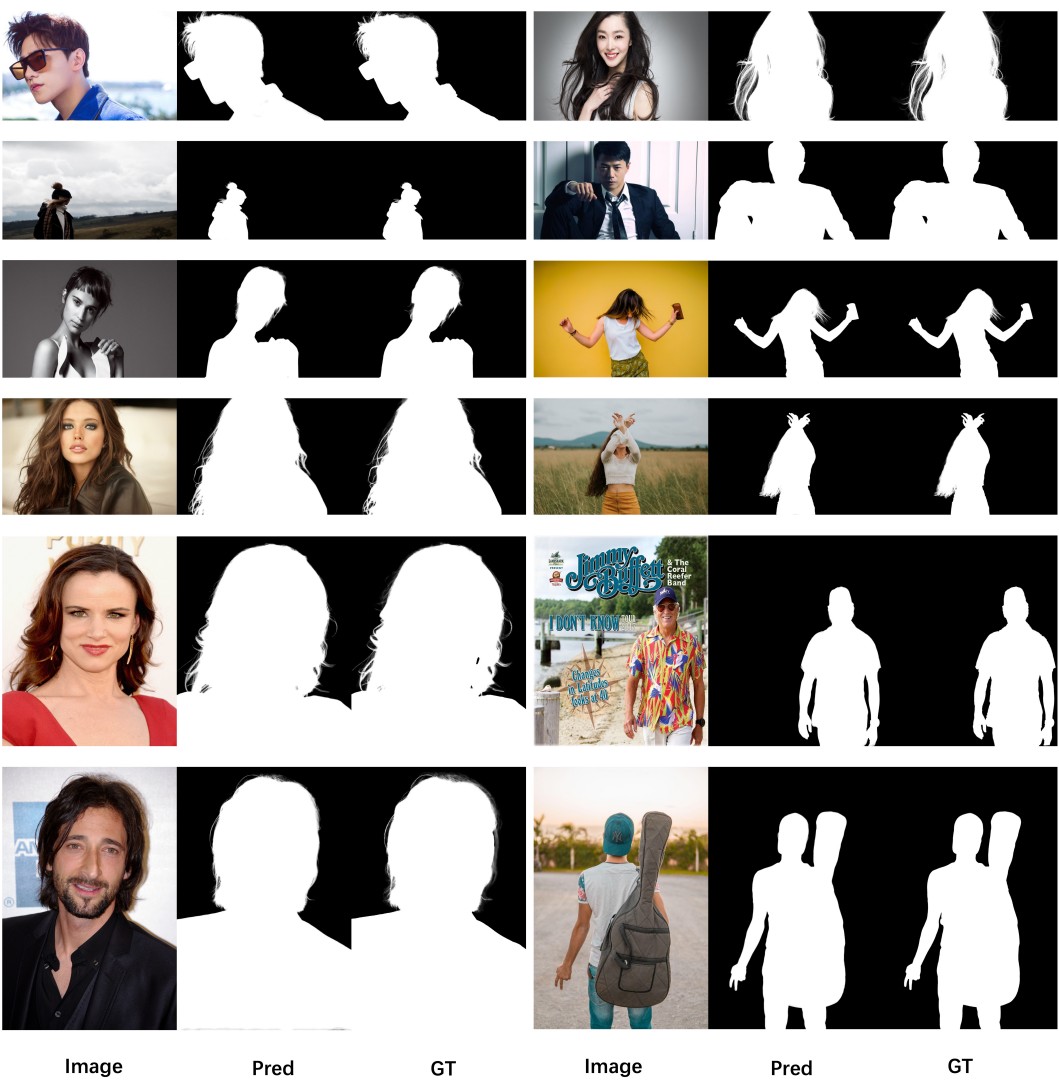

Figure 10: More qualitative results for image matting.

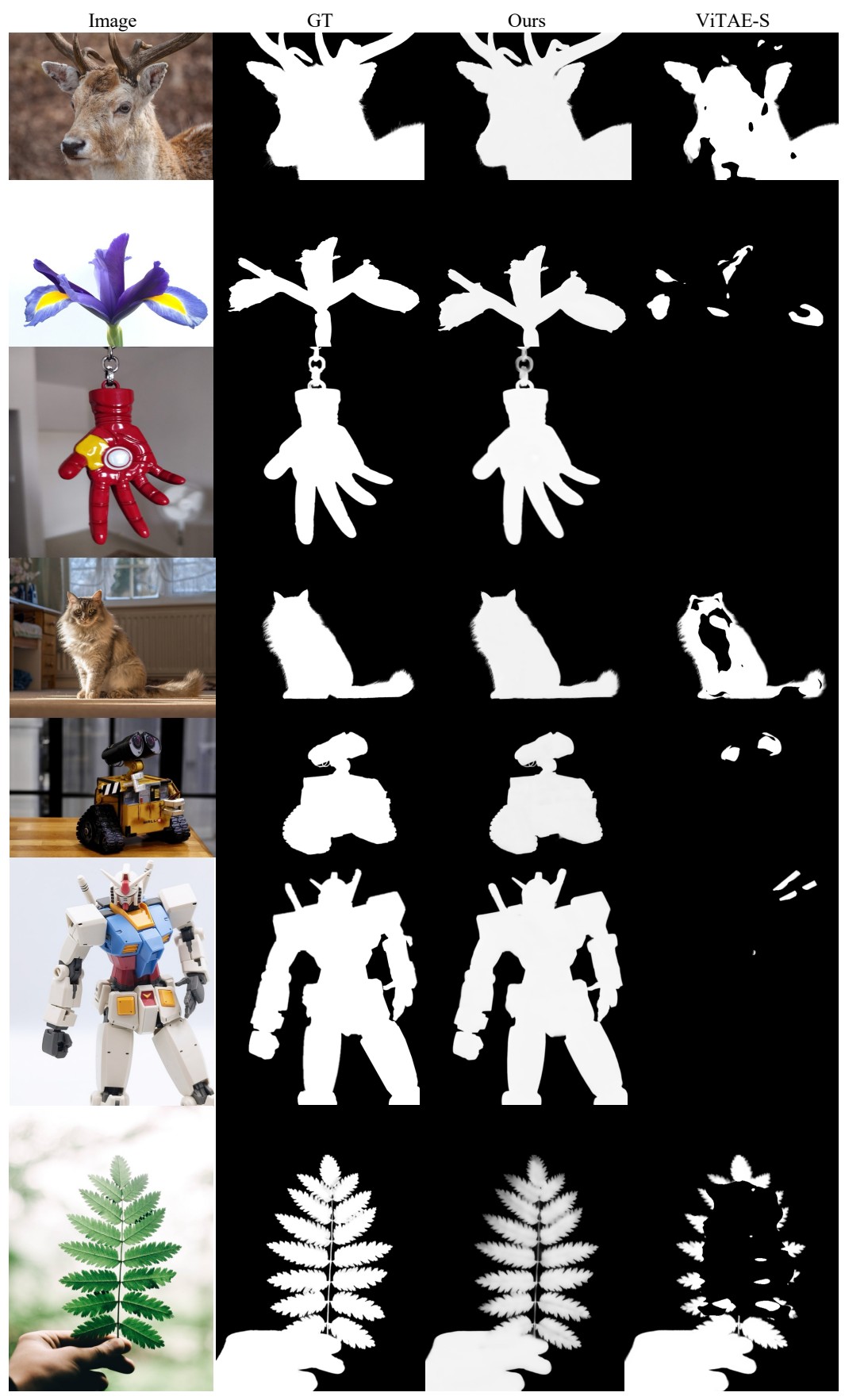

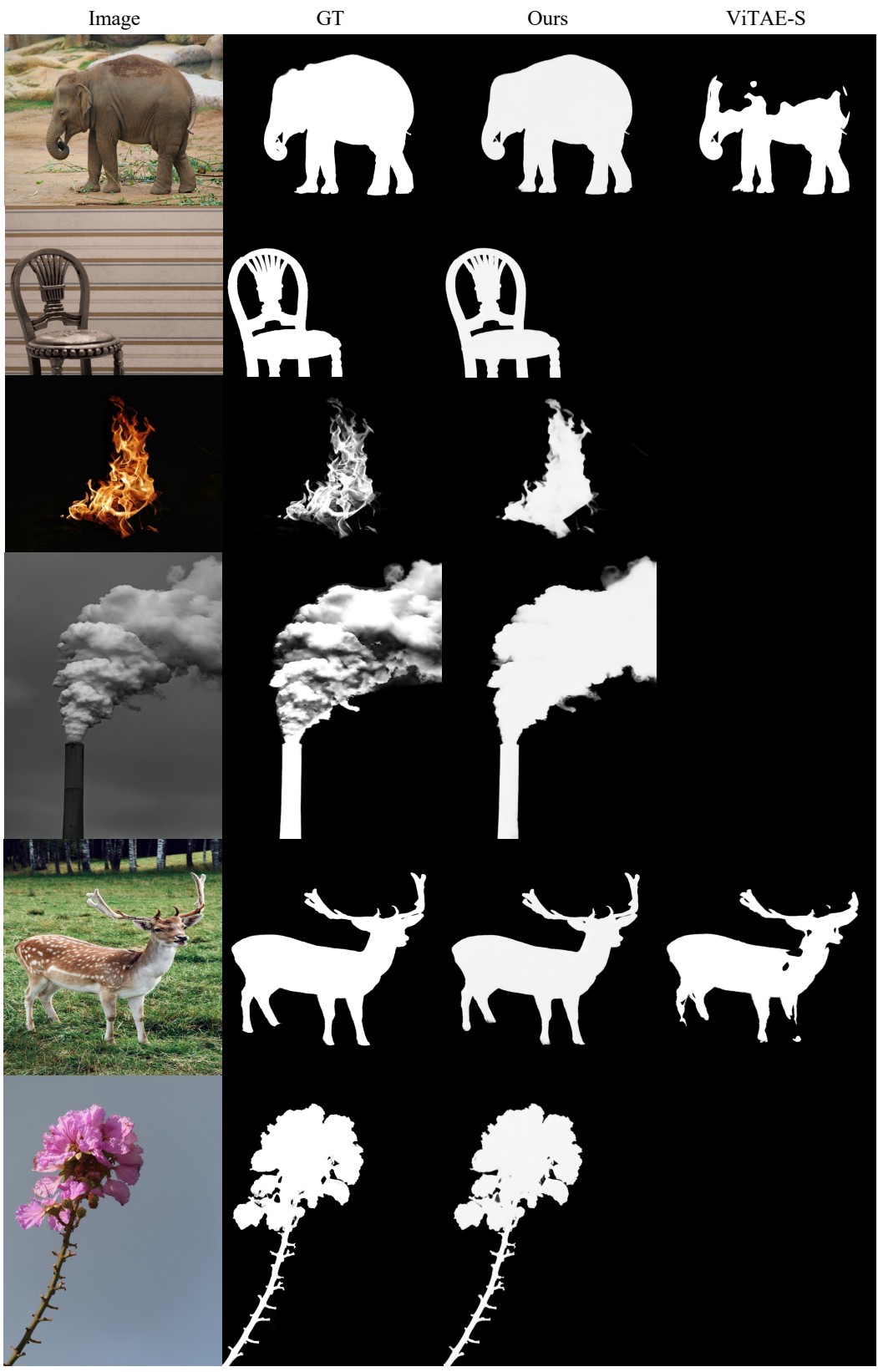

Figure 11: Generalization ability of the human matting model to general image matting images.

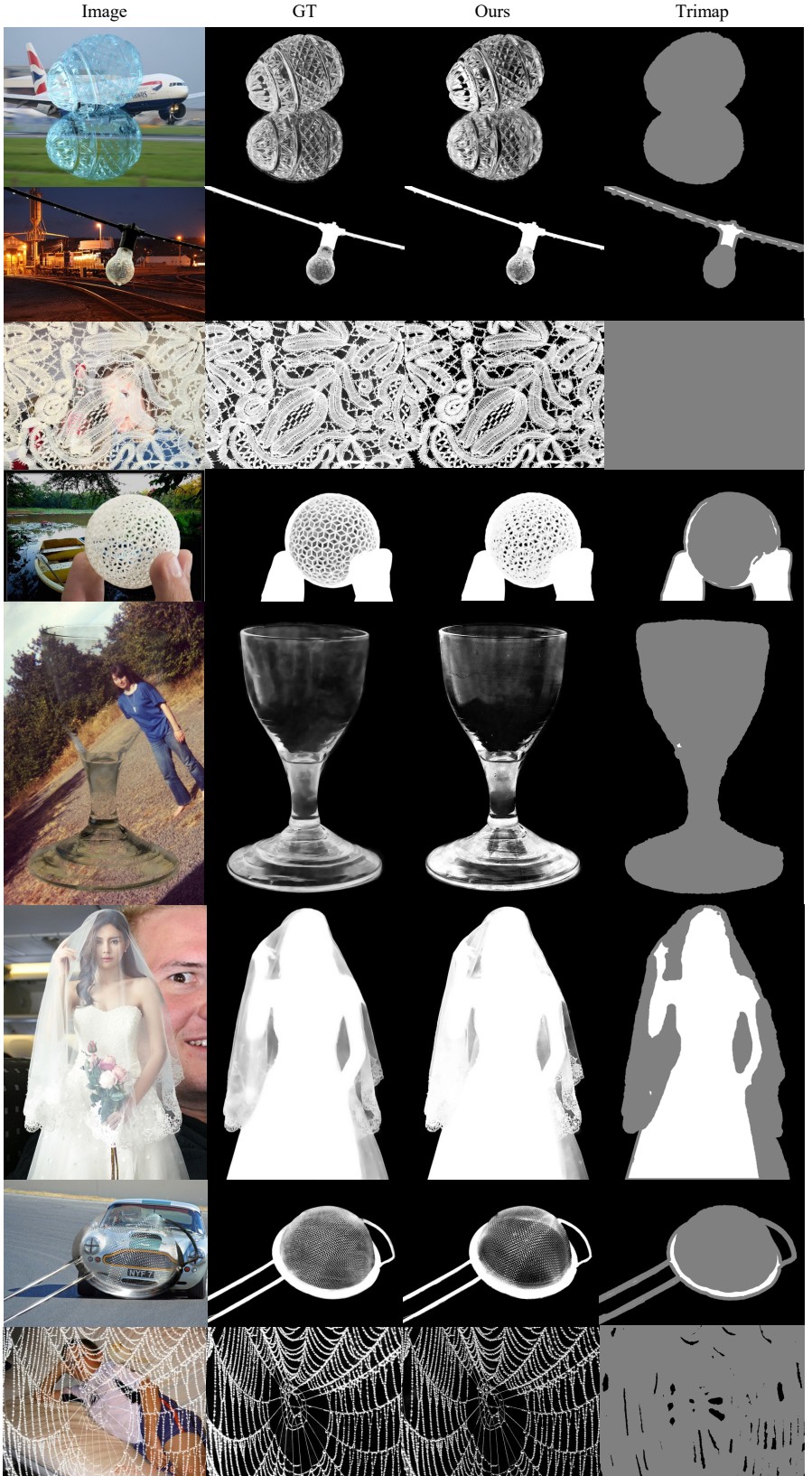

Figure 12: More types of image matting such as semi-transparent objects.

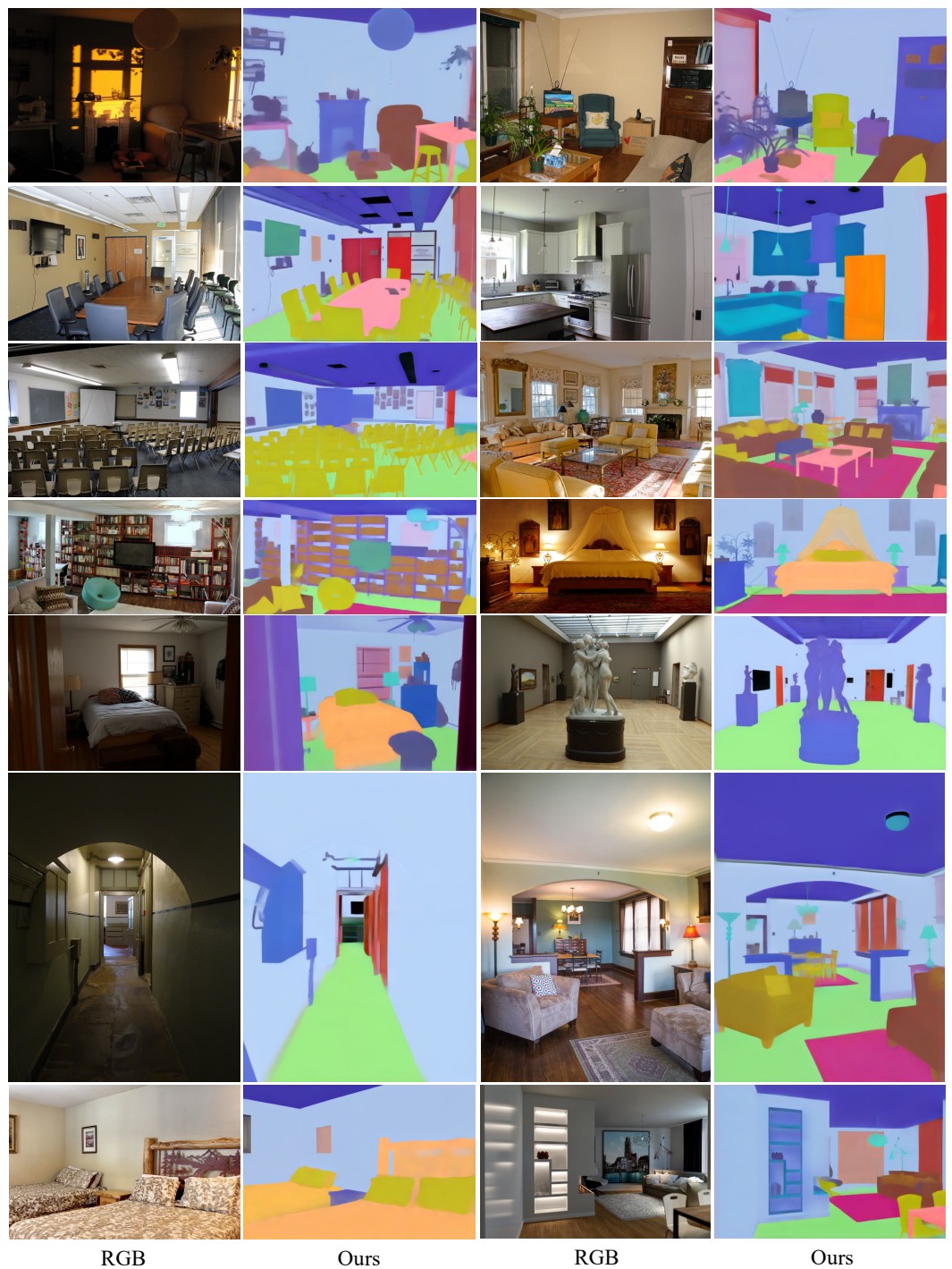

Figure 13: More qualitative results for image segmentation.

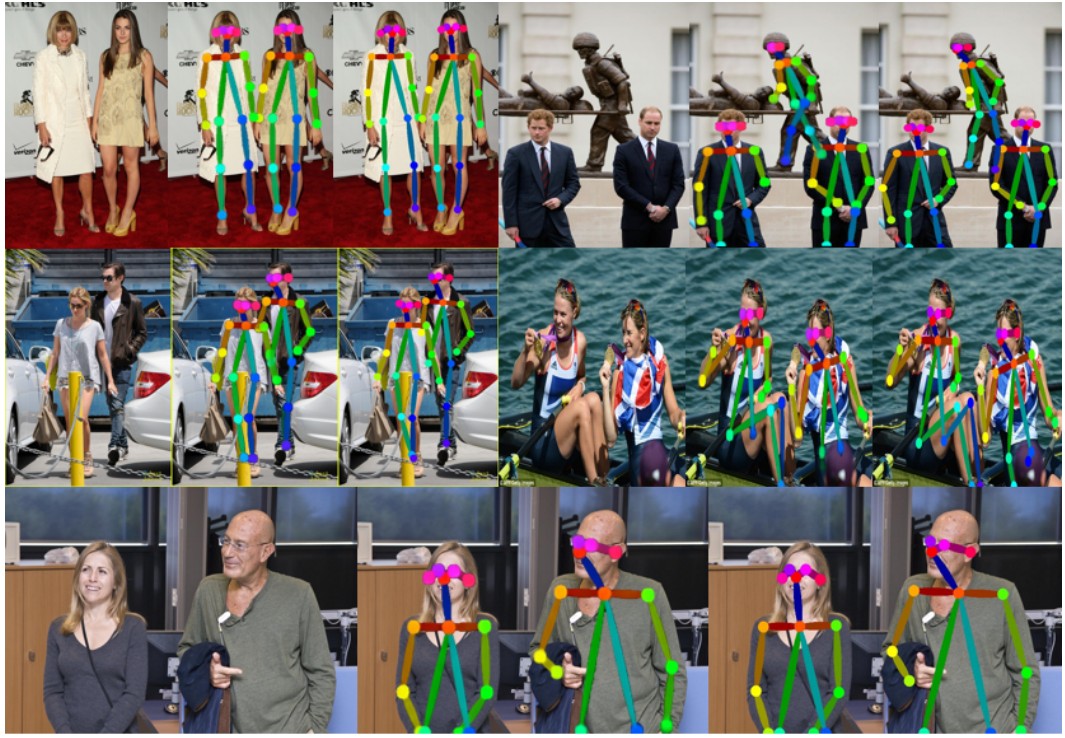

Figure 14: More qualitative results for human pose estimation. (Left: original Image, Mid: prediction, Right: ground truth)