# OpenReview forum: "What Matters When Repurposing Diffusion Models for General Dense Perception Tasks?"
_ICLR.cc/2025/Conference — ICLR 2025 Poster_

### Official Review · Reviewer_QgJw · 2024-10-25

**Soundness:** 3
**Presentation:** 4
**Contribution:** 4
**Rating:** 6
**Confidence:** 4

**Summary:**

This paper presents a novel approach to harnessing pretrained diffusion models for general dense perception tasks, including monocular depth estimation, surface normal estimation, dichotomous image segmentation, semantic segmentation, and human pose estimation. The central idea of this work is to utilize the diffusion model as a robust pretrained backbone, subsequently fine-tuning it for a variety of downstream applications.

**Strengths:**

1. This paper provides systematical analysis of the design space of diffusion model for dense perception task, as shown in the 5 findings in paper. The core design of this paper is to employ a deterministic one step perception.
2. By using the proposed training protocol, the proposed GenPercept harnesses the power of the pre-trained UNet from diffusion models for dense visual perception tasks, including monocular depth estimation, surface normal estimation, dichotomous image segmentation and semantic segmentation.
3. This paper presents a solid analysis accompanied by extensive experiments. The results are not only intuitive but also promising, demonstrating a strong potential for downstream application.

**Weaknesses:**

1. Given that GenPercept is currently trained on a relatively small dataset, it tends to lag behind models that benefit from extensive data training. To enhance its performance, it would be beneficial to scale GenPercept's training with a larger volume of data in the future.
2. Given the robust prior established by training on the extensive LAION dataset, the question arises: what would be the outcome of employing alternative self-supervised methods, such as MAE or CLIP, using the same LAION dataset? A comparative analysis of these approaches against the diffusion pretrain would provide valuable insights into their relative efficacy and potential advantages.

**Questions:**

1. Trained with synthetic data, why does GenPercept not suffer from the sim2real domain gap?
2. In the supplementary materials, line 22 should utilize the citation format 'citep'.
3. Can GenPercept be applied to general perception tasks, such as detection?

---

> ### Author Response · Authors · 2024-11-20
> **Response to reviewer 6WHQ**
>
> We deeply appreciate your acknowledgment of our motivation and approach. The insights you've provided, especially regarding the questions and potential areas for improvement, are immensely helpful. Below, we strive to address these in detail.
>
> > W1: Given that GenPercept is currently trained on a relatively small dataset, it tends to lag behind models that benefit from extensive data training. To enhance its performance, it would be beneficial to scale GenPercept's training with a larger volume of data in the future.
>
> We sincerely appreciate your valuable guidance on expanding the training data volume. This insight holds significant value not only for practical applications but also for advancing academic research on the effectiveness of large-scale data training. We have incorporated this suggestion into the conclusion section and plan to design and conduct experiments to explore this further in future work.
>
>
> > W2:Given the robust prior established by training on the extensive LAION dataset, the question arises: what would be the outcome of employing alternative self-supervised methods, such as MAE or CLIP, using the same LAION dataset? A comparative analysis of these approaches against the diffusion pretrain would provide valuable insights into their relative efficacy and potential advantages.
>
>
> Thank you for your insightful suggestion. It is quite valuable to explore whether the detailed visual perception predictions generated by existing diffusion models primarily benefit from the extensive LAION dataset or the diffusion pretraining paradigm itself. We have incorporated this suggestion into the conclusion section and will continue to investigate the advantages and unique contributions of diffusion models in future research.
>
>
> > Q1: Trained with synthetic data, why does GenPercept not suffer from the sim2real domain gap?
>
> We attribute this phenomenon to two reasons. First, the diffusion models pre-trained on the LAION dataset have the ability to generate both realistic and stylized images, which may decrease the sim2real domain gap, because it can regard the simulated images as a specific style of real image. Second, with the development of simulators, the synthetic datasets tend to be more and more realistic, which decreases the sim2real domain gap on perception tasks a lot.
>
> > Q2: In the supplementary materials, line 22 should utilize the citation format 'citep'.
>
> Thanks so much for your reminder. We have fixed it and updated it in the latest version of our manuscript.
>
> > Q3: Can GenPercept be applied to general perception tasks, such as detection?
>
> We attempted the human pose estimation (keypoint detection) task by reformulating it as a 3-channel dense map estimation in the supplementary, and it shows some robustness to out-of-domain images like cats and animation. An alternative approach would be to replace the VAE decoder with a customized detection head and corresponding loss functions. Similarly, it can be extended to implement tasks like object detection.

---

> > ### Comment · Reviewer_QgJw · 2024-11-23
> >
> > Thanks to the authors for the detailed response. After reading other reviewers’ reviews, I find that I share similar questions with reviewer 7LHd regarding why preventing “ground truth leakage” leads to improved model performance:
> >
> > 1. This claim seems to deviate from the principles of original diffusion models. Preventing “ground truth leakage” in your experiments involves training the diffusion model with almost pure Gaussian noise. However, diffusion training typically requires varying noise levels to help the model learn diverse denoising capabilities for the iterative denoising process. The depth or normal map you aim to predict does not appear to affect the diffusion formulation, as it can be treated as a specific type of "image." Could you clarify why reducing “ground truth leakage” is effective in your approach?
> > 2. As mentioned in Lines 189-190, as the β value increases, the impact of “multi-resolution noise” diminishes, however, the performance is improved. Does this contradict your assumption in Lines 157-158, where you stated that “multi-resolution noise” can enhance accuracy? Additionally, is there any relationship between “multi-resolution noise” and the noise proportion?
> > 3. In Tab. 1, which is the default Marigold configuration, why are these metrics significantly worse than those reported in the Marigold paper?
> >
> > Thanks again for the authors' response. This appears to be a critical issue in your paper, and addressing it could significantly enhance its credibility and soundness.

---

> > > ### Author Response · Authors · 2024-11-23
> > > **Response to reviewer QgJw**
> > >
> > > We sincerely appreciate your fast response and are delighted to have the opportunity to engage in further discussion with you.
> > >
> > > > Q1: This claim seems to deviate from the principles of original diffusion models. Preventing “ground truth leakage” in your experiments involves training the diffusion model with almost pure Gaussian noise. However, diffusion training typically requires varying noise levels to help the model learn diverse denoising capabilities for the iterative denoising process. The depth or normal map you aim to predict does not appear to affect the diffusion formulation, as it can be treated as a specific type of "image." Could you clarify why reducing “ground truth leakage” is effective in your approach?
> > >
> > >
> > > Thank you for raising this insightful question. The reason hinds behind the difference between text-guided image generation and visual perception tasks. In _text-guided image generation_, a single textual input can correspond to an immense variety of potential images. This **inherent uncertainty** makes generating a high-quality image directly from random noise in a single step extremely challenging. Therefore, the _multi-step generation_ enables the model to incrementally remove noise, progressively refining details and textures at each stage, which effectively **simplifies the task**. However, _visual perception tasks_ conditioned on an RGB image are **deterministic without any randomness**, and such an easy _injective mapping_ can be estimated with a _one-step inference process_, as most of the traditional visual perception methods do.
> > >
> > > While Marigold series algorithms aim to leverage diffusion models' ability of generating highly detailed images to enhance visual perception with precise details, reformulating straightforward deterministic tasks as a denoising process can **further simplify this problem**, leading to what is described as "ground truth leakage" in Section 3.1 of the main paper and illustrated in Figure 2 of the supplementary. In summary, our experiments and theoretical analysis can prove the unnecessity of employing the denoising process for visual perception tasks. We have updated this analysis in the supplementary material.
> > >
> > >
> > > > Q2: As mentioned in Lines 189-190, as the β value increases, the impact of “multi-resolution noise” diminishes, however, the performance is improved. Does this contradict your assumption in Lines 157-158, where you stated that “multi-resolution noise” can enhance accuracy? Additionally, is there any relationship between “multi-resolution noise” and the noise proportion?
> > >
> > >
> > > Thanks so much for your valuable guidance. In lines 189–190, the phrase “the impact of multi-resolution noise diminishes” indicates that the performance gap between “Marigold with multi-resolution noise” and “Marigold without multi-resolution noise” gradually narrows. Similarly, in lines 157–158, the statement “multi-resolution noise can enhance accuracy” is supported by a comparison of the overall performance of “Marigold with multi-resolution noise” versus “Marigold without multi-resolution noise.” The former generally demonstrates superior performance, particularly when beta values are small. These two conclusions do not conflict with each other.
> > >
> > > We agree that there does not exist any relationship between “multi-resolution noise” and the noise proportion. To avoid misunderstanding, we have deleted the related description.
> > >
> > >
> > >
> > > > Q3: In Tab. 1, which is the default Marigold configuration, why are these metrics significantly worse than those reported in the Marigold paper?
> > >
> > > The primary difference lies in the dataset composition. Marigold utilizes "54K Hypersim + Virtual KITTI," whereas GenPercept employs "50K Hypersim + 40K Virtual KITTI." For GenPercept, we adopt a stringent filtering policy to exclude invalid Hypersim scenes, and this may result in slightly lower performance. However, all experiments in Section 3 are conducted under fair and consistent experimental settings, ensuring that the conclusions remain unaffected. When comparing with Marigold in Section 4, we use its officially released weights for evaluation.

---

### Official Review · Reviewer_7LHd · 2024-11-01

**Soundness:** 3
**Presentation:** 2
**Contribution:** 3
**Rating:** 6
**Confidence:** 3

**Summary:**

This paper researches the process of fine-tuning pre-trained text-to-image diffusion models for dense perception tasks like monocular depth estimation and semantic segmentation, and analyzes several design choices in that process. In this analysis, it considers the model architecture, training procedure, model initialization, dataset selection, and fine-tuning protocol. Based on this analysis, five findings are made:

1.	Diffusion models can be fine-tuned accurately and efficiently for dense perception tasks with a one-step, deterministic approach.
2.	The most useful prior knowledge of the pre-trained diffusion model is contained in the U-Net denoiser. The VAE decoder can be replaced with other components without problems.
3.	With the one-step model, multi-timestep training is irrelevant, and additional text inputs do not significantly impact performance.
4.	Fine-tuning the U-Net denoiser leads to better results than freezing it or applying low-rank adaptation.
5.	Training data quality affects the prediction quality.

Based on these findings, the paper presents GenPercept, a paradigm for fine-tuning diffusion models for dense prediction tasks. Experiments show that GenPercept is an effective method to fine-tune Stable Diffusion for various dense prediction tasks. On several benchmarks, GenPercept achieves a competitive performance.

**Strengths:**

1.	Most importantly, the analysis of the design choices that play a role when fine-tuning diffusion models for monocular depth estimation, described in Sec. 3, leads to various relevant and useful insights.

    a)	The finding that fine-tuning can be done accurately with a one-step, deterministic approach (Tab. 1) is useful because it allows for significantly more efficient inference.

    b)	Moreover, by formulating the task as one-step estimation, the model can now te trained with task-specific losses, such as the image angular loss for surface normal estimation. With experiments (Tab. 7), this is shown to improve the results.

    c)	The finding that most of the useful prior knowledge is contained in the U-Net of the pre-trained diffusion model, and not in the VAE decoder (Tab. 2), is useful because it means that the VAE decoder can be replaced with task-specific decoders. This is shown to be significantly boost the semantic segmentation performance in Tab. 9 (per L418-L423).

    d)	The finding that the use of multiple timesteps during training is not necessary (Tab. 3) is useful because it allows the training procedure to be simplified to single-timestep training.

    e)	The finding that the U-Net denoiser should be fine-tuned fully, without low-rank adaptation (Tab. 4), is useful because it provides clear guidelines for future methods that aim to fine-tune diffusion models for depth estimation and other dense perception tasks.

2.	The presentation of the main findings of the paper in the text boxes (e.g., L283 and L296) is very helpful and useful. These text boxes clearly summarize the impact of the results that have been discussed thus far, and they help the reader to see the most interesting results of the paper at a glance.

3.	The presented GenPercept model is valuable because it takes advantage of the aforementioned findings of the design-choice analysis. The application and evaluation of this model on multiple tasks and datasets provides the reader with insights into the performance that can be obtained when combining the best individual design choices. Moreover, these results can serve as a baseline for future works that aim to fine-tune diffusion models for multiple downstream perception tasks.

**Weaknesses:**

1.	The scores for DepthFM in Tab. 6 and GeoWizard in both Tab. 6 and Tab. 7 do not correspond to the scores originally reported in the respective papers. The numbers in these tables should be altered to reflect the originally reported numbers, or the text should explain why the numbers differ from these original numbers. Some examples:

    a)	GeoWizard: 9.7 AbsRel and 92.1 $\delta_{1}$ on KITTI in original paper [a], but 12.9 AbsRel and 85.1 $\delta_{1}$ in this submitted manuscript.

    b)	DepthFM: 8.3 AbsRel and 93.4 $\delta_{1}$ on KITTI in original paper [b], but 17.4 AbsRel and 71.8 $\delta_{1}$ in this submitted manuscript.

2.	The results in Tab. 8 (and Tab. 2 of the supp) make it seem like GenPercept achieves state-of-the-art results in dichotomous image segmentation. However, this table does not contain the results of top-performing model MVANet [c]. This model achieves a max$F_{\beta}$ score of 0.916 on Overall DIS-TE (1-4), compared to 0.863 by GenPercept. Even with results that are inferior to MVANet, GenPercept has value, but it should be clear to the reader that GenPercept does not achieve state-of-the-art results, so these results should be added to the tables.
3.	For the image matting task on the P3M-500-NP dataset, GenPercept is only compared to the ResNet-34 variant of P3M [d], not the Swin-T or ViTAE-S variants which achieve much better results, e.g., 7.59 SAD for ViTAE-S compared to 11.23 SAD for ResNet-34. By only reporting the results for the ResNet-34 variant, it seems like GenPercept performs similarly to the state of the art, whereas this is not the case. The results of P3M for ViTAE-S and/or Swin-T should be added to the paper, or the text should clearly explain why such a comparison is not necessary.
4.	One of the main benefits of the one-step inference procedure of GenPercept is the improved efficiency compared to Marigold and other diffusion-based methods. The efficiency is also mentioned as a key characteristic of the method (L539). However, the paper does not explicitly evaluate the efficiency of the model. Therefore, it is not clear what the exact speedup is over existing diffusion-based models, or how the efficiency compares to the other methods reported in Tab. 6, 7, 8, and 10. The paper would be significantly stronger if the efficiency of GenPercept was shown quantitatively, by reporting the inference time of GenPercept and other methods.
5.	L187-L188 states that the results indicate that increasing the ‘training challenges’ leads to improved model performance. However, the paper does not provide any explanation for this. Why does the performance increase when the training task becomes more challenging? This seems counterintuitive. The value of the paper would improve if the authors provided an explanation (or hypothesis) for this phenomenon, as this could provide insights into the way these diffusion models learn dense perception tasks.
6.	The paper does not clearly define $\beta_{start}$ and $\beta_{end}$, even though first experiment and finding (Tab. 1 & L204) are fully focused around changing the values of these parameters. Fig. 2(b) illustrates that they impact the proportion of noise that is added to the latents in different time steps, but an exact definition is not provided. In L107, the paper refers to the supplementary material for more details, but the supplementary material only mentions $\beta_{s}$ and $\beta_{t}$. The clarity of the paper would improve if a clear definition of $\beta_{start}$ and $\beta_{end}$ was provided.
7.	The paper does not evaluate if the multi-class image segmentation performance of GenPercept is competitive with existing image segmentation methods. Currently, the paper only conducts an experiment where GenPercept is trained on HyperSim images with 40 semantic classes, and evaluated on ADE20K images with 40 classes. As this is a newly proposed setting, these results are not comparable with existing segmentation methods. To see if fine-tuning diffusion models is truly valuable for image segmentation, GenPercept should be fine-tuned and evaluated on standard segmentation benchmarks like ADE20K, and compared to existing segmentation models like Mask2Former [e].
8.	A minor weakness of the paper is that there is not a version of GenPercept that clearly works better than the other. As discussed in L357-L359 and shown in Tab. 6, the GenPercept model trained for depth estimation scores better on NYU, ScanNet and ETH3D, while the model trained for disparity estimation scores significantly better on KITTI and DIODE. As a result, two different models are necessary for different situations, and it is not always clear which of the models works best for an unknown application domain.

Some minor weaknesses, which do not significantly impact my rating:

9.	L230: The acronym “DPT” is not defined, nor is there a reference to a work where it is presented. As a result, it’s not clear what a “DPT encoder” is.
10.	The meaning of the “fine-tune” column in Tab. 9 is not clear. From the text in L418-L423, it appears that it refers to the use of the UPerNet decoder. If so, the text “UPerNet” seems more appropriate than “fine-tune”. If it means something else than this, this should be better specified in the paper.
11.	The ordering of the labels in the legend of Fig. 2 (b) is confusing, as there appears to be no logical ordering. The legend, and the figure as whole, would be easier to interpret if the labels were ordered in descending or ascending manner, e.g. starting with (0.0002125, 0.003) and ending with (1.0, 1.0), or vice versa.

12.	There are a few errors in the text:

    a) L024: "tailed for" - Do the authors mean "tailored for"?

    b)	L187-L188: "increasing training challenges lead to" => "increasing training challenges leads to"

[a] Fu et al., “GeoWizard: Unleashing the Diffusion Priors for 3D Geometry Estimation from a Single Image,” ECCV 2024.

[b] Gui et al., “DepthFM: Fast Monocular Depth Estimation with Flow Matching,” arXiv:2403.13788, 2024.

[c] Yu et al., "Multi-view Aggregation Network for Dichotomous Image Segmentation," CVPR 2024.

[d] Ma et al., "Rethinking Portrait Matting with Privacy Preserving," IJCV 2023.

[e] Cheng et al., “Masked-attention Mask Transformer for Universal Image Segmentation,” CVPR 2022.

**Questions:**

The main reason that I currently give a slightly low rating is because of the incorrect numbers of existing models, the missing quantitative comparisons to some other existing models, the missing efficiency experiments, and some missing explanations (see the ‘weaknesses’ section). I would like to ask the authors to carefully address my concerns, answer the questions posed in the ‘weaknesses’ section, and revise the manuscript accordingly.

Additionally, I have some other questions/suggestions:

1.	From the text in L411-L423, it appears like the classes for semantic segmentation are always encoded into 3-channel colormaps, even when UPerNet is used. Is this really the case? If so, why isn’t the ‘regular’ semantic segmentation format used, with one channel for each individual class? If not, please clarify this in the text.
2.	In the related work section, it seems appropriate to also mention DINOv2, because it has shown to be very suitable for downstream visual perception tasks like depth estimation (e.g., with Depth Anything [g]) and semantic segmentation.

[f] Oquab et al., “DINOv2: Learning Robust Visual Features without Supervision,” TMLR 2024.

[g] Yang et al., “Depth Anything: Unleashing the Power of Large-Scale Unlabeled Data,” CVPR 2024.

---

**Update after author discussion.** After reading the different reviews, the authors' response, and the revised manuscript, and having a discussion with the authors, I have decided to upgrade my rating from 5 to 6.

By answering my questions and revising the manuscript based on my review and follow-up questions, the authors have convincingly addressed the majority of my concerns. Importantly, the paper now compares the proposed method more fairly to existing methods, provides insightful inference speed results, better explains the operation of the method, and discusses the cause of the impact of different $\beta$ values in a more nuanced way. As such, the I believe the paper is stronger than the initial submission, and I upgrade my rating.

---

> ### Author Response · Authors · 2024-11-20
> **Part 1 of Response to reviewer 7LHd**
>
> We sincerely thank you for your very careful and detailed review. We are delighted that you found our work to be comprehensive and convincing. The questions and weaknesses you've pointed out are incredibly helpful to us, and we will do our utmost to address them below.
>
> > W1: The scores for DepthFM in Tab. 6 and GeoWizard in both Tab. 6 and Tab. 7 do not correspond to the scores originally reported in the respective papers. The numbers in these tables should be altered to reflect the originally reported numbers, or the text should explain why the numbers differ from these original numbers. Some examples:
> >
> >a) GeoWizard: 9.7 AbsRel and 92.1$\delta_1$ on KITTI in original paper [a], but 12.9 AbsRel and 85.1 $\delta_1$ in this submitted manuscript.
> >
> >b) DepthFM: 8.3 AbsRel and 93.4 $\delta_1$ on KITTI in original paper [b], but 17.4 AbsRel and 71.8 $\delta_1$ in this submitted manuscript.
>
> Both Geowizard and DepthFM reported the final results with little evaluation details, but they didn't release the evaluation code. For geometry evaluation, the ensemble size, inference resolution, valid evaluation depth range (specific for depth estimation), and evaluation average paradigm (average by pixels or average by the number of images) can be different for each method. **To compare these approaches fairly**, we follow the _open-source evaluation code of Marigold for depth and DSINE for surface normal_, and evaluate the performance of existing SOTA methods with their officially released model weights. Therefore, the performance can be different from that reported in their paper. We add an explanation of the performance in red at the beginning of Section 4 of the main paper.
>
> > W2: The results in Tab. 8 (and Tab. 2 of the supp) make it seem like GenPercept achieves state-of-the-art results in dichotomous image segmentation. However, this table does not contain the results of top-performing model MVANet [c]. This model achieves a max
> $F_{\beta}$ score of 0.916 on Overall DIS-TE (1-4), compared to 0.863 by GenPercept. Even with results that are inferior to MVANet, GenPercept has value, but it should be clear to the reader that GenPercept does not achieve state-of-the-art results, so these results should be added to the tables.
> >
> > W3: For the image matting task on the P3M-500-NP dataset, GenPercept is only compared to the ResNet-34 variant of P3M [d], not the Swin-T or ViTAE-S variants which achieve much better results, e.g., 7.59 SAD for ViTAE-S compared to 11.23 SAD for ResNet-34. By only reporting the results for the ResNet-34 variant, it seems like GenPercept performs similarly to the state of the art, whereas this is not the case. The results of P3M for ViTAE-S and/or Swin-T should be added to the paper, or the text should clearly explain why such a comparison is not necessary.
>
>
> More comparisons of dichotomous image segmentation and image matting. Thank you very much for pointing out this issue. We fully agree with your guidance to accurately reflect the correct ranking of GenPercept. The quantitative and qualitative comparisons of these two tasks have been updated in Table 8, Table 10, Figure 5, and Figure 6 of the main paper, and Table 4, Figure 7, and Figure 10 of the supplementary. Although GenPercept performs lower than existing SOTA methods in these two tasks, we find that **GenPercept exhibits much more enhanced robustness when applied to in-the-wild images of both dichotomous image segmentation and image matting** thanks to the robust pre-training knowledge of stable diffusion. Unlike achieving the highest performance on a specific dataset, _GenPercept offers a general network architecture_ that shows much robustness for dense perception tasks and challenging in-the-wild images and possesses its unique value.

---

> ### Author Response · Authors · 2024-11-20
> **Part 2 of Response to reviewer 7LHd**
>
> > W4: One of the main benefits of the one-step inference procedure of GenPercept is the improved efficiency compared to Marigold and other diffusion-based methods. The efficiency is also mentioned as a key characteristic of the method (L539). However, the paper does not explicitly evaluate the efficiency of the model. Therefore, it is not clear what the exact speedup is over existing diffusion-based models, or how the efficiency compares to the other methods reported in Tab. 6, 7, 8, and 10. The paper would be significantly stronger if the efficiency of GenPercept was shown quantitatively, by reporting the inference time of GenPercept and other methods.
>
> We are very grateful for your valuable suggestion. The runtime analysis has been updated and highlighted in the "Runtime Analysis" section of the supplementary. Our proposed one-step inference paradigm demonstrates a runtime that is _94% and 57% less than_ those of multi-step methods with ensemble and without ensemble, respectively. Besides, by incorporating a customized head such as a DPT head, both runtime and GPU memory requirements are _further reduced by 27%_ without compromising performance, maintaining a competitive level of efficiency.
>
>
> > W5: L187-L188 states that the results indicate that increasing the ‘training challenges’ leads to improved model performance. However, the paper does not provide any explanation for this. Why does the performance increase when the training task becomes more challenging? This seems counterintuitive. The value of the paper would improve if the authors provided an explanation (or hypothesis) for this phenomenon, as this could provide insights into the way these diffusion models learn dense perception tasks.
>
> **The "training challenge" here represents "preventing the ground truth leakage"**. For the training process, the input of diffusion models is derived from the forward diffusion process, a linear blending process of noise and ground truth with different timesteps and noise forms, as illustrated in Figure 2(a). For small timesteps like "t=200", _the input blended image still contains part of ground truth information_ (e.g., the purple color of the surface normal) and may lead to a "ground truth leakage" problem.
>
> On the other hand, the blending proportion is controlled by the beta values ($\beta_{start}$, $\beta_{end}$) of the diffusion model scheduler. As shown in Figure 2(b) and Figure 2 of the supplementary, _increasing the beta values can increase the noise proportion and decrease the ground truth proportion_. Training with larger beta values can consistently achieve better performance for both Gaussian noise and RGB noise, which is proved in Table 1 and Figure 2(c). The detailed revisions have been updated and highlighted in red in Section 3.
>
> > W6: The paper does not clearly define $\beta_{start}$ and $\beta_{end}$, even though first experiment and finding (Tab. 1 & L204) are fully focused around changing the values of these parameters. Fig. 2(b) illustrates that they impact the proportion of noise that is added to the latents in different time steps, but an exact definition is not provided. In L107, the paper refers to the supplementary material for more details, but the supplementary material only mentions $\beta_{s}$ and $\beta_{t}$. The clarity of the paper would improve if a clear definition of $\beta_{start}$ and $\beta_{end}$ was provided.
>
> We are very grateful for your guidance. The proportion of Gaussian noise is related to $\alpha_t$, and $\alpha_t$ is computed by cumulative production when the scheduler of $\beta$ values is known. The scheduler is parameters with two hyperparameters $\beta_{start}$ and $\beta_{end}$, which defines the $\beta$ values of t=0 and t=1000, respectively. For a casual timestep $s$, $\beta_s$ is computed by linearly interpolating between $\sqrt{\beta_{start}}$ and $\sqrt{\beta_{end}}$, then squaring each interpolated value. The detailed revisions have been updated and highlighted in red in Section 1 of the supplementary.

---

> ### Author Response · Authors · 2024-11-20
> **Part 3 of Response to reviewer 7LHd**
>
> > W7: The paper does not evaluate if the multi-class image segmentation performance of GenPercept is competitive with existing image segmentation methods. Currently, the paper only conducts an experiment where GenPercept is trained on HyperSim images with 40 semantic classes, and evaluated on ADE20K images with 40 classes. As this is a newly proposed setting, these results are not comparable with existing segmentation methods. To see if fine-tuning diffusion models is truly valuable for image segmentation, GenPercept should be fine-tuned and evaluated on standard segmentation benchmarks like ADE20K, and compared to existing segmentation models like Mask2Former [e].
>
> Thanks so much for the valuable advice. With a similar experimental setting, we train GenPercept with an UpperNet on ADE20K and Mask2Former, and the results are provided in Table 9. GenPercept outperforms ResNet50 and Swin-T of Mask2Former but achieves lower performance than that of Swin-L. Revisions have been updated in red in the main paper.
>
> > W8: A minor weakness of the paper is that there is not a version of GenPercept that clearly works better than the other. As discussed in L357-L359 and shown in Tab. 6, the GenPercept model trained for depth estimation scores better on NYU, ScanNet and ETH3D, while the model trained for disparity estimation scores significantly better on KITTI and DIODE. As a result, two different models are necessary for different situations, and it is not always clear which of the models works best for an unknown application domain.
>
> We believe that the difference between the depth model and the disparity model inherently exists for all the network architectures. Experimentally, we suggest adopting the depth model for indoor scenes and the disparity model for outdoor scenes.
>
> ---
>
> minor weakness:
>
>
> > W9: L230: The acronym “DPT” is not defined, nor is there a reference to a work where it is presented. As a result, it’s not clear what a “DPT encoder” is.
>
> DPT [a] is a classical architecture of vision transformers for dense prediction. We simply leverage its head to realize lightweight monocular depth estimation. We have updated and cited it in the paper.
>
> > W10:The meaning of the “fine-tune” column in Tab. 9 is not clear. From the text in L418-L423, it appears that it refers to the use of the UPerNet decoder. If so, the text “UPerNet” seems more appropriate than “fine-tune”. If it means something else than this, this should be better specified in the paper.
> >
> > W11: The ordering of the labels in the legend of Fig. 2 (b) is confusing, as there appears to be no logical ordering. The legend, and the figure as whole, would be easier to interpret if the labels were ordered in descending or ascending manner, e.g. starting with (0.0002125, 0.003) and ending with (1.0, 1.0), or vice versa.
> >
> > W12:There are a few errors in the text:
> >
> > a) L024: "tailed for" - Do the authors mean "tailored for"?
> >
> > b) L187-L188: "increasing training challenges lead to" => "increasing training challenges leads to"
>
> We appreciate your suggestion. The relevant revisions have been highlighted in red in the latest version of the manuscript.
>
> > Q1: From the text in L411-L423, it appears like the classes for semantic segmentation are always encoded into 3-channel colormaps, even when UPerNet is used. Is this really the case? If so, why isn’t the ‘regular’ semantic segmentation format used, with one channel for each individual class? If not, please clarify this in the text.
>
> For the UperNet segmentation head, we follow the traditional semantic segmentation format to use n-channel output where n is the number of categories. We have updated and clarified it in red in the paper.
>
> > Q2: In the related work section, it seems appropriate to also mention DINOv2, because it has shown to be very suitable for downstream visual perception tasks like depth estimation (e.g., with Depth Anything [g]) and semantic segmentation.
>
> Thanks for your advice, we will cite DINOv2 in the related works.
>
> [a] Ranftl, René, Alexey Bochkovskiy, and Vladlen Koltun. "Vision transformers for dense prediction." Proceedings of the IEEE/CVF international conference on computer vision. 2021.

---

> ### Comment · Reviewer_7LHd · 2024-11-22
> **Response to rebuttal**
>
> I would like to thank the authors for their detailed response to all the reviews. In their response, the authors have adequately addressed the majority of my concerns. However, I have a few remaining concerns and some follow-up questions.
>
> ---
>
> **Regarding W1:** I thank the authors for the explanation. However, it is likely that the default Marigold and DSINE settings are not optimal for DepthFM and GeoWizard, so I’m not convinced that using the Marigold and DSINE evaluation settings to generate the evaluation scores for these methods is completely fair. To present a complete picture to the reader, I believe it is fair to (in addition to the reproduced scores that are already in Tab. 6 and Tab. 7) also present the originally reported scores of DepthFM and GeoWizard, with a note about the difference in evaluation settings.
>
> ---
>
> **Regarding W4:** I thank the authors for including the runtime analysis in the revised paper. This demonstrates the positive impact of the one-step inference procedure on the model’s speed. However, it is not yet clear how the prediction speed compares to that of existing models listed in Tab. 6 and Tab. 7. Could the authors report the runtime for some of these state-of-the-art methods, such that is clear how GenPercept compares in terms of efficiency?
>
> ---
>
> **Regarding W5:** I do not completely follow what the authors mean by “ground truth leakage” as mentioned in the rebuttal and L182 of the revised manuscript. Why is it a bad thing that the blended input image still contains part of the ground truth during the forward diffusion process, e.g., at timestep t=200? Why does this lead to decreased depth estimation performance? This is not clearly explained.
>
> ---
>
> I look forward to reading the authors’ response. Thanks in advance!

---

> > ### Author Response · Authors · 2024-11-23
> > **Response to reviewer 7LHd**
> >
> > Thank you for your fast and valuable feedback. We sincerely appreciate your thoughtful comments and have carefully addressed the concerns raised, as detailed below.
> >
> >
> > > Regarding W1: I thank the authors for the explanation. However, it is likely that the default Marigold and DSINE settings are not optimal for DepthFM and GeoWizard, so I’m not convinced that using the Marigold and DSINE evaluation settings to generate the evaluation scores for these methods is completely fair. To present a complete picture to the reader, I believe it is fair to (in addition to the reproduced scores that are already in Tab. 6 and Tab. 7) also present the originally reported scores of DepthFM and GeoWizard, with a note about the difference in evaluation settings.
> >
> > We fully acknowledge and agree on the importance of reporting the origin scores. We sincerely appreciate the valuable suggestion and have revised the Table 6 and Table 7 accordingly.
> >
> >
> >
> > > Regarding W4: I thank the authors for including the runtime analysis in the revised paper. This demonstrates the positive impact of the one-step inference procedure on the model’s speed. However, it is not yet clear how the prediction speed compares to that of existing models listed in Tab. 6 and Tab. 7. Could the authors report the runtime for some of these state-of-the-art methods, such that is clear how GenPercept compares in terms of efficiency?
> >
> >
> > Thank you once again for your valuable suggestion. We fully agree on the importance of comparing the GenPercept inference speed with existing state-of-the-art methods. Accordingly, we have incorporated this comparison into Table 1 of the supplementary materials. Compared to existing state-of-the-art diffusion-based methods, our proposed GenPercept achieves a notable improvement in inference speed, attributed to the innovative one-step inference paradigm and the customized head. While our method demonstrates inference speeds comparable to Metric3Dv2 and DSINE, it falls behind DepthAnythingV2. Note that the superior performance of DepthAnythingV2 is facilitated by its training on a relatively lightweight model, bolstered by extensive labeled and unlabeled datasets, and supported by substantial computational resources distributed across multiple GPUs.
> >
> >
> >
> > > Regarding W5: I do not completely follow what the authors mean by “ground truth leakage” as mentioned in the rebuttal and L182 of the revised manuscript. Why is it a bad thing that the blended input image still contains part of the ground truth during the forward diffusion process, e.g., at timestep t=200? Why does this lead to decreased depth estimation performance? This is not clearly explained.
> >
> >
> > In each iteration of training diffusion models for visual perception, a timestep t is sampled to control the proportion of noise added to the ground truth latent, and the network is trained to recover a clean ground truth latent from the noisy latent. For smaller timesteps like t=200, as illustrated in Figure 2(a), the input to the network retains significant ground truth information, making it comparatively easier to recover the clean ground truth latent than attempting recovery in the absence of any ground truth information. In contrast, the experimental setting of GenPercept involves inputting purely a noisy latent devoid of ground truth information, presenting a greater challenge. This hypothesis can be proved by the experiments summarized in Table 1, where blending the ground-truth latent with an increasing proportion of noise consistently leads to stable performance improvements during training. We have updated the related modification in Section 3.

---

> > ### Author Response · Authors · 2024-11-23
> > **An extra analysis about the “ground truth leakage”**
> >
> > Reviewer QgJw raised a similar concern regarding W5: specifically, why reducing “ground truth leakage” is effective, and this seems to deviate from the principles of original diffusion models. We copy the question and our response here and hope this clarification will be helpful to you.
> >
> > > Q1 of reviewer QgJw: This claim seems to deviate from the principles of original diffusion models. Preventing “ground truth leakage” in your experiments involves training the diffusion model with almost pure Gaussian noise. However, diffusion training typically requires varying noise levels to help the model learn diverse denoising capabilities for the iterative denoising process. The depth or normal map you aim to predict does not appear to affect the diffusion formulation, as it can be treated as a specific type of "image." Could you clarify why reducing “ground truth leakage” is effective in your approach?
> >
> > In _text-guided image generation_, a single textual input can correspond to an immense variety of potential images. This **inherent uncertainty** makes generating a high-quality image directly from random noise in a single step extremely challenging. Therefore, the _multi-step generation_ enables the model to incrementally remove noise, progressively refining details and textures at each stage, which effectively **simplifies the task**. However, _visual perception tasks_ conditioned on an RGB image are **deterministic without any randomness**, and such an easy _injective mapping_ can be estimated with a _one-step inference process_, as most of the traditional visual perception methods do.
> >
> > While Marigold series algorithms aim to leverage diffusion models' ability of generating highly detailed images to enhance visual perception with precise details, reformulating straightforward deterministic tasks as a denoising process can **further simplify this problem**, leading to what is described as "ground truth leakage" in Section 3.1 of the main paper and illustrated in Figure 2 of the supplementary. In summary, our experiments and theoretical analysis can prove the unnecessity of employing the denoising process for visual perception tasks. We have updated this analysis in the supplementary material.

---

> ### Comment · Reviewer_7LHd · 2024-11-25
> **Response to authors**
>
> Thanks to the authors for providing further clarifications and for further revising the manuscript.
>
> After reading the different comments about the "ground truth leakage", including the comment by reviewer QgJw, I still have some remaining concerns though.
>
> First, I would not refer to this phenomenon as "ground truth leakage". "Ground truth leakage" could suggest that the ground truth is somehow used by the model during inference, which is not the case here.
>
> Second, L186-L187 states the following:
>
> > Our quantitative and qualitative analyses, presented in table 1 and fig. 2(c), indicate that increasing the noise proportion leads to improved model performance.
>
> However, Tab. 1 shows that the performance decreases when increasing the noise proportions ($\beta_{start}$, $\beta_{end}$) to (0.1360, 0.192) and (0.5440, 0.768) in combination with Marigold. While L189 states that the authors consider this to be caused by the randomness of Gaussian noise, the same trend can be observed both with and without multi-resolution noise, so the effect does not appear to be completely random. Moreover, using large noise proportions (1.0, 1.0) does not lead to better results than using (0.0034, 0.048). Therefore, the situation appears to be a bit more nuanced than sketched by the authors. Specifically, the results indicate that there are multiple different somewhat optimal values for $\beta_{start}$ and $\beta_{end}$, i.e., (a) somewhat low values like the baseline Marigold settings and (b) pure noise. This contradicts the claim that reducing "ground truth leakage" improves the performance, as the performance does not consistently improve when "ground truth leakage" is reduced by using larger noise proportions.
>
> Some specific questions related to this:
> - Have the authors conducted multiple different training runs (with different random seeds) with noise proportions  (0.1360, 0.192) and (0.5440, 0.768) and observed large variations in performance between runs, suggesting that it is really due to randomness? Or were the results quite consistent, suggesting that it is not due to randomness?
> - If the results are not due to randomness, how do the authors explain them? In this case, the "ground truth leakage" is still lower for noise proportions (0.5440, 0.768) than for (0.00085, 0.012), but the performance is worse. Do the authors still think the results can be explained by "ground truth leakage"?
> - Similarly, if reducing "ground truth leakage" improves performance, why does the performance not improve when increasing noise proportions from (0.0034, 0.048) to (1.0, 1.0)?
>
> Even if it not possible to find a single, conclusive reason/explanation for the results in Tab. 1, I think the experiment has value as it shows the behavior of models across different settings and identifies that using pure noise during allows for a good performance, which subsequently enables single-step inference. However, if such a reason cannot be found, this should be mentioned honestly and clearly in the paper. Of course, the authors can still provide hypotheses for the results, but in this case it should be clear that these are hypotheses and that it is not certain if they are correct.

---

> > ### Author Response · Authors · 2024-11-27
> > **Response to reviewer 7LHd**
> >
> > Thanks to reviewer 7LHd for the important findings and suggestions. We have revised our clarification to remain neutral and impartial in our analyses.
> >
> > > First, I would not refer to this phenomenon as "ground truth leakage". "Ground truth leakage" could suggest that the ground truth is somehow used by the model during inference, which is not the case here.
> >
> > We appreciate and agree with the feedback regarding the term "ground truth leakage". We have modified our description and now refer to it as a hypothesis of "a certain level of ground-truth label information being part of the input during training." Thank you for your valuable guidance.
> >
> > > While L189 states that the authors consider this to be caused by the randomness of Gaussian noise, the same trend can be observed both with and without multi-resolution noise, so the effect does not appear to be completely random. Moreover, using large noise proportions (1.0, 1.0) does not lead to better results than using (0.0034, 0.048).
> >
> > To investigate whether the observed performance decrease is attributable to the inherent randomness of diffusion models, we followed the reviewer’s suggestion and conducted experiments by varying the random seed during both the training and inference process. However, the results showed only slight variations, which led us to reconsider our original hypothesis. These findings suggest that the performance decline is not due to randomness as initially proposed, and we have changed the related description. For the result using noise proportion (1.0, 1.0), we consider it better than using (0.0034, 0.048) according to the "Rank" performance, which means the average rank of ten evaluation performance.
> >
> > > Specifically, the results indicate that there are multiple different somewhat optimal values for $\beta_{start}$ and $\beta_{end}$, i.e., (a) somewhat low values like the baseline Marigold settings and (b) pure noise. This contradicts the claim that reducing "ground truth leakage" improves the performance, as the performance does not consistently improve when "ground truth leakage" is reduced by using larger noise proportions.
> >
> > We fully agree with the finding that "there are multiple different somewhat optimal beta values". The relevant revisions have been updated in Section 3.1. We rule out the influence of randomness and propose this hypothesis, which indicates there may exist various factors besides the "ground-truth label being part of the input". For the related specific questions, the points mentioned above can help address them.

---

> ### Comment · Reviewer_7LHd · 2024-11-28
> **Response to authors**
>
> Thanks to the authors for their detailed answer, and for updating the manuscript accordingly.
>
> After the extensive discussion, and taking into account the other reviews and responses, I have decided to upgrade my rating from 5 to 6.
>
> By answering my questions and revising the manuscript, the authors have convincingly addressed the majority of my concerns. Importantly, the paper now compares the proposed method more fairly to existing methods, provides insightful inference speed results, better explains the operation of the method, and discusses the cause of the impact of different $\beta$ values in a more nuanced way. As such, the I believe the paper is stronger than the initial submission, and I upgrade my rating.

---

> > ### Author Response · Authors · 2024-12-01
> > **Response to reviewer 7LHd**
> >
> > Thank you for your thorough and constructive feedback. We greatly appreciate the time and effort you invested in reviewing our manuscript. Your positive feedback is encouraging, and the detailed comments and thoughtful suggestions have significantly improved the quality of our work. We are grateful and pleased for your recognition of the revisions we made in response to your questions.

---

### Official Review · Reviewer_6WHQ · 2024-11-03

**Soundness:** 3
**Presentation:** 3
**Contribution:** 3
**Rating:** 6
**Confidence:** 3

**Summary:**

This paper reveals key elements of diffusion models for downstream dense perception tasks. The authors did a full range of validation from the perspectives of model design and training data, unveiled some IMPORTANT factors, and proposed a new model named GenPercept,
The authors have conducted extensive experiments on five dense perception tasks, including monocular depth estimation, surface normal estimation, image segmentation, and matting. This extensive experimentation serves as a testament to the effectiveness and universality of their method.

**Strengths:**

- The authors conduct extensive ablation studies from different aspects and finally figure out the key factors affecting transfer efficiency and performance when using pre-trained diffusion models.
- Starting from the inconsistency of the results of two existing methods, the authors dig deeper into the influencing factors and then propose their GenPercept. This approach can provide some new ideas for doing research.
- These findings are inspiring and can provide constructive insights into model design when adapting pre-trained models for downstream tasks.

**Weaknesses:**

- The biggest weakness is that all conclusions are based on experiments using large synthetic datasets trained on deep estimation tasks. I have two concerns. First, would these conclusions still hold if the datasets were small? Second, would these conclusions still hold if the dataset was completely real?
- It is not appropriate to use the P3M 10K dataset to verify the robustness of GenPercept in image matting. This is because the human matting dataset is extremely similar to the dichotomous image segmentation task if only the boundary regions are considered and the deterministic foreground regions are ignored. Instead, if the authors wish to validate its robustness in the image matting task, more types of objects (e.g., semi-transparent objects) should be included and then its performance should be observed.

**Questions:**

What would be the comparison of generalization for OOD data for models trained using synthetic and real data?

---

> ### Author Response · Authors · 2024-11-20
> **Response to reviewer 6WHQ**
>
> Thank you so much for your thoughtful recognition of our goals and strategy. The questions and critiques you’ve shared are crucial for our progress, and we are committed to addressing them comprehensively in the following sections.
>
> > W1: The biggest weakness is that all conclusions are based on experiments using large synthetic datasets trained on deep estimation tasks. I have two concerns. First, would these conclusions still hold if the datasets were small? Second, would these conclusions still hold if the dataset was completely real?
>
> The training dataset in our study includes 50K Hypersim images and 40K Virtual KITTI images, whose data volume is relatively small compared to traditional methods such as DPT, which utilizes 1.4M images. To address this, we conducted an additional experiment exploring the impact of data volume, with results detailed in the latest version of the supplementary. These experiments demonstrate some robustness in training with respect to data volume.
>
> For realistic datasets, we provide a fair comparison between models trained exclusively on synthetic datasets and those trained on real datasets. Quantitative and qualitative results are presented in Table 5 of the main paper and Figure 3 of the supplementary. Models trained purely on real data perform worse in terms of both quantitative metrics and visualization quality compared to their synthetic-trained counterparts, highlighting the effectiveness of synthetic data.
>
> > W2: It is not appropriate to use the P3M 10K dataset to verify the robustness of GenPercept in image matting. This is because the human matting dataset is extremely similar to the dichotomous image segmentation task if only the boundary regions are considered and the deterministic foreground regions are ignored. Instead, if the authors wish to validate its robustness in the image matting task, more types of objects (e.g., semi-transparent objects) should be included and then its performance should be observed.
>
> We appreciate your valuable suggestion. Besides human matting, we also train GenPercept on a more general image matting task on the Composition-1k[a] dataset, and the qualitative results have been updated in Figure 11 of the supplementary. It shows robustness on more types of objects such as semi-transparent objects, hollow objects, etc. Besides, the human matting GenPercept model shows much more robustness on general objects. Please see Figure 10 of supplementary for details.
>
> > Q1: What would be the comparison of generalization for OOD data for models trained using synthetic and real data?
>
> Thanks so much for your suggestion. We compare the generalization performance of models trained on synthetic and real data for out-of-distribution scenarios, and the quantitative results are shown in Figure 3 of the supplementary. The model trained on synthetic data achieves comparable robustness to that trained on real data, but achieves better performance on transparent objects and geometric details. Generally, our GenPercept can generalize well to diverse scenes unseen during training. For example, the surface normal estimation of animation images in Figure 6 of the supplementary, and the keypoint estimation of animation and cat images in Figure 4 of the supplementary.
>
> [a] Xu, Ning, et al. "Deep image matting." Proceedings of the IEEE conference on computer vision and pattern recognition. 2017.

---

> > ### Comment · Reviewer_6WHQ · 2024-11-25
> >
> > Thanks for the rebuttal. Overall, my concerns are addressed. I will maintain my initial rating.

---

> > > ### Author Response · Authors · 2024-11-25
> > > **Response to reviewer 6WHQ**
> > >
> > > Thank you for your feedback. We appreciate your time and effort in reviewing our work.

---

### Official Review · Reviewer_nfdP · 2024-11-04

**Soundness:** 3
**Presentation:** 4
**Contribution:** 3
**Rating:** 6
**Confidence:** 4

**Summary:**

This paper investigates key factors affecting the transfer performance of pretrained diffusion models repurposed for dense visual perception tasks, emphasizing the importance of fine-tuning data quality, training strategies, and task-specific supervision. It introduces GenPercept, a one-step fine-tuning paradigm that enhances inference speed and improves fine-grained details in predictions across various perception tasks.

**Strengths:**

1. The exploration of key factors influencing the transferability of text-to-image (T2I) diffusion models to dense visual perception tasks is intriguing and relevant.
2. The experiments are thorough and comprehensive, providing strong support for the findings.
3. The proposed deterministic one-step perception approach effectively integrates these findings and demonstrates comparable performance with minimal fine-tuning.

**Weaknesses:**

There are some confusing aspects in the experimental setup and comparisons for the downstream tasks. See questions.

**Questions:**

1. What training set was used for Table 1? Are you following the training data of DMP or Marigold? If so, why is the first row of Table 5 identical to the baseline? If not, why did you begin directly with a synthetic dataset?
2. For monocular depth estimation, why is DMP not included as a baseline?
3. The authors mention that using a customized head and loss could accelerate inference time. Could you provide a comparison to demonstrate this improvement?

---

> ### Author Response · Authors · 2024-11-20
> **Response to reviewer nfdP**
>
> We sincerely appreciate your thoughtful recognition of our approach and motivation. The questions you have raised provide us with valuable insights, which we will address with careful consideration and detailed analysis in the following sections.
>
> > Q1: What training set was used for Table 1? Are you following the training data of DMP or Marigold? If so, why is the first row of Table 5 identical to the baseline? If not, why did you begin directly with a synthetic dataset?
>
> The motivation of GenPercept is to analyze the crucial designs of methods like Marigold, which leverages diffusion models for perception tasks and achieves excellent geometry detail. So we follow Marigold's training setting for Table 1 and all other experiments in Section 3. The training dataset contains 50K Hypersim images and 40K Virtual KITTI images, and these images are sampled with a sampling rate of 90% for Hypersim and 10% for Virtual KITTI. We provide a more detailed explanation in Section 4 of the supplementary. Therefore, the first row of Table 5 is identical to "Our baseline" in Table 1.
>
> > Q2: For monocular depth estimation, why is DMP not included as a baseline?
>
> Thanks again for your suggestion. We download the official code and depth checkpoint of DMP, and compare with it fairly following the zero-shot monocular depth estimation setting in the main paper. The origin DMP method uses ZoeDepth[a] to generate the pseudo ground truth for each image, and therefore performs poorer than our trained DMP presented in Table 1. The relevant revisions have been highlighted in Table 6 in red in the latest version of the manuscript.
>
> > Q3: The authors mention that using a customized head and loss could accelerate inference time. Could you provide a comparison to demonstrate this improvement?
>
> We appreciate the valuable advice. We fully agree on the significance of inference time comparison, and it has been updated and highlighted in red in the "Runtime Analysis" section of the supplementary. With a customized DPT head, we can achieve 0.24s inference time, which is 27% faster than that of 0.33s for the VAE decoder model.
>
> [a] Bhat, Shariq Farooq, et al. "Zoedepth: Zero-shot transfer by combining relative and metric depth." arXiv preprint arXiv:2302.12288 (2023).

---

> > ### Comment · Reviewer_nfdP · 2024-11-25
> >
> > Thanks for the rebuttal. My concerns are mostly addressed. I prefer to maintaining my initial rating.

---

> > > ### Author Response · Authors · 2024-11-27
> > > **Response to reviewer nfdP**
> > >
> > > Thank you for your valuable feedback. We truly appreciate the time and attention you've dedicated to reviewing our work. Your suggestions are vital in guiding our improvements.

---

### Author Response · Authors · 2024-11-20
**General Response and Summary of Updates to Manuscript**

We would like to express our sincere gratitude to the reviewers and editors for their time, effort, and constructive feedback in evaluating our work. Your thoughtful comments and suggestions have been invaluable in enhancing the quality and rigor of our manuscript. We have carefully read all feedback and have provided detailed responses to each of the reviewers' comments. We trust that these responses adequately address the concerns and suggestions raised.

We are greatly encouraged by the reviewers’ recognition of our efforts in investigating key factors influencing transfer performance (**nfdP**, **6WHQ**, **7LHd**, **QgJw**), the effectiveness and efficiency of our novel one-step GenPercept approach (**nfdP**, **6WHQ**, **7LHd**, **QgJw**), and the robustness of our experiments (**nfdP**, **QgJw**). Below, we provide a high-level summary of the revisions made to the manuscript in response to the reviewers' feedback, followed by a restatement of the key contributions of our work.

---

Here is the summary of updates that we've made to the draft, and the relevant revisions have been highlighted in red in the latest version of the manuscript.
- Clarified some experimental settings and analysis. (**nfdP**, **7LHd**)
- Compared with more related approaches in Table 6, Table 8, Table 9, and Table 10 of the main paper, and Table 4 of the supplementary. (**nfdP**, **7LHd**)
- Added the quantitative efficiency improvement experiment in Table 1 of the supplementary. (**nfdP**, **7LHd**)
- Explored the effectiveness of data volume in Table 1 of the supplementary. (**6WHQ**)
- Added more qualitative analysis experiments in Figure 5 and Figure 6 of the main paper and Figure 1, Figure 2, Figure 3, Figure 7, Figure 10, and Figure 11 of the supplementary material, which shows the generalization ability of GenPercept. (**7LHd**)
- Added more types of objects in image matting in Figure 11 of the supplementary. (**6WHQ**)

To end this update, we would like to summarize the primary contributions of our work. GenPercept presents an efficient and effective approach to leveraging the prior knowledge embedded in diffusion models trained on large-scale data for dense perception tasks. Unlike approaches only focused on achieving state-of-the-art performance on a specific dataset, _GenPercept emphasizes robustness, detail, and generalizability_ in dense visual perception tasks while _maintaining a unified network architecture_. This work opens new possibilities for realizing more generalizable dense perception frameworks with the benefits of diffusion model priors.

---

### Meta-Review · Area_Chair_BWeU · 2024-12-18

**Metareview:**

The paper introduces GenPercept, a novel approach leveraging diffusion models to enhance dense visual perception tasks. The work claims improvements in inference speed and detail of predictions, substantiated through a series of experiments across multiple tasks.

The paper stands out for its rigorous experimental validation and the clear demonstration of the versatility and efficiency of diffusion models in dense perception contexts. The design space analysis and the results presented are commendable and represent the paper's key strengths.

Concerns are raised about the heavy reliance on extensive synthetic datasets, leading to questions about the model's performance with smaller or real-world data. The initial comparison with existing methods was not entirely convincing, and the paper lacked a quantitative evaluation of model efficiency. Additionally, some theoretical claims, such as "ground truth leakage," were not sufficiently clear or substantiated.

The decision leans toward acceptance due to the paper's robust experimental work and the novelty of applying diffusion models in this manner. Despite this, the unresolved issues around the real-world applicability and some theoretical ambiguities slightly detract from the paper's impact.

**Additional Comments On Reviewer Discussion:**

Reviewers raised critical points during the discussion:
- Reviewer nfdP was concerned about the sim2real gap.
- Reviewer 6WHQ felt the performance comparison with existing methods was lacking.
- Reviewer 7LHd noted the absence of quantitative model efficiency evaluation.
- Reviewer QgJw pointed out unclear theoretical claims.

In response, the authors:
- Offered comparisons between synthetic and real data to address nfdP's concerns, though questions about robustness remain.
- Updated the manuscript to better compare with state-of-the-art methods, alleviating 6WHQ's concern.
- Included a runtime analysis to satisfy 7LHd's call for model efficiency evaluation.
- Clarified theoretical aspects and revised their explanation of "ground truth leakage," somewhat addressing QgJw's criticism.

In reaching the final decision, I considered the extent to which the authors' responses alleviated the reviewers' concerns. While not all issues were fully resolved, the authors demonstrated a commitment to enhancing the clarity and rigor of their work, contributing to the decision to accept.

---

### Decision · Program_Chairs · 2025-01-22

Accept (Poster)